# TSGym: Automatic Model Design Framework for Deep Multivariate Time-Series Forecasting

## Abstract

Recently, deep learning has driven significant advancements in multivariate time series forecasting (MTSF) tasks. Prevailing paradigm in MTSF research involves proposing models as pre-defined, holistic architectures. Such an approach limits adaptability across diverse data scenarios, and obscures the individual contributions of their core components. To address this, we propose TSGym, a novel framework for automated MTSF model design. The framework begins with ***decoupling*** existing deep MTSF methods into fine-grained components, which enables a large-scale, component-level evaluation that offers crucial insights, and creates a vast space for the automated construction of potentially superior models. Leveraging this space through ***strategic sampling***, a core ***meta-learner*** is trained to learn the mapping between component configurations and performance across multiple traininig datasets. This enables it to perform zero-shot selection of a top-performing model for any new, unseen time series data. Extensive experiments indicate that the model automatically constructed by our proposed TSGym significantly outperforms existing state-of-the-art MTSF methods and AutoML solutions, and exhibit high potential for ***transferability*** across diverse datasets.

## 1 Introduction

Multivariate time series refer to time series data involving multiple interdependent variables, which are widely present in various fields such as finance (Sezer et al., 2020), energy (Alvarez et al., 2010; Deb et al., 2017), traffic (Cirstea et al., 2022; Yin and Shang, 2016), and health (Bui et al., 2018; Kaushik et al., 2020). Among the numerous research topics, multivariate time series forecasting (MTSF) attracts substantial attention from the research community due to its significant practical applications. Traditional approaches to MTSF are largely based on statistical methods (Abraham and Ledolter, 2009; Zhang, 2003) and machine learning techniques (Hartanto et al., 2023; Masini et al., 2023). In recent years, deep learning (DL) has become the most active area of research for MTSF, driven by its ability to handle complex patterns and large-scale datasets effectively (Wang et al., 2024b).

Early academic efforts of deep MTSF methods like RNN-type methods (Yamak et al., 2019) are reported to struggle with capturing long-term temporal dependencies due to their inherent limitations of gradient vanishing or exploding problem (Zhou et al., 2021; 2022b). More recently, Transformer (Vaswani et al., 2017) shows significant potential, largely due to the effectiveness of its attention mechanisms in modeling temporal correlation (Vaswani et al., 2017; Wen et al., 2022). Consequently, attention mechanism has continuously been studied in MTSF, with a focus on adapting them to time series data, for instance, by exploiting sparsity inductive bias (Li et al., 2019; Zhou et al., 2021), transforming time and frequency domains (Zhou et al., 2022b), and fusing multi-scale series (Liu et al., 2022b). While simpler MLP-based structures emerged (Zeng et al., 2023a) offering alternatives to the established Transformer architecture in MTSF, notable modeling strategies like series-patching and channel-independent (Nie et al., 2023), significantly enhanced the performance of Transformer-based methods, thereby sustaining research interest in them. Building upon these developments, large time-series models including large language models (LLMs) (Jin et al., 2024b; Zhou et al., 2023; Jin et al., 2023b) and time series foundation models (TSFMs) (Jin et al., 2023c; Liu et al., 2024b) have recently been introduced, achieving promising results and fostering new research directions for MTSF. Alongside these advancements in model architectures, active research within the deep MTSF community also focuses on other critical topics, such as variable (channels)

dependency modeling (Nie et al., 2023; Liu et al., 2024a; Zhang and Yan, 2023), series normalization methods (Liu et al., 2022c; Fan et al., 2023), and trend-seasonal decomposition (Zeng et al., 2023a; Liu et al., 2023).

As the field of MTSF continues to diversify, existing studies typically address critical concerns about methodological effectiveness, either by conducting large-scale benchmarks (Wang et al., 2024b; Shao et al., 2024; Qiu et al., 2024) or performing model selection via AutoML (Abdallah et al., 2022; Fischer and Saadallah, 2024). However, we identify three main challenges with these prevailing approaches: First, *the granularity of existing studies is insufficient*. Current benchmarking works evaluate or select models as a whole, which hinders a deeper understanding of the mechanisms that drive model performance. In AutoML, this lack of granularity prevents breakthroughs beyond the limits of existing models. Second, *the scope of existing studies is limited*. Current benchmarking and automated selection efforts are often confined to restricted model architectures or hyperparameters, without covering a broad range of data processing methods or feature modeling techniques. Third, *the range of existing studies is narrow*. Existing studies tend to cover only a subset of network architectures and often lack discussions on more diverse models, such as LLMs and TSFMs.

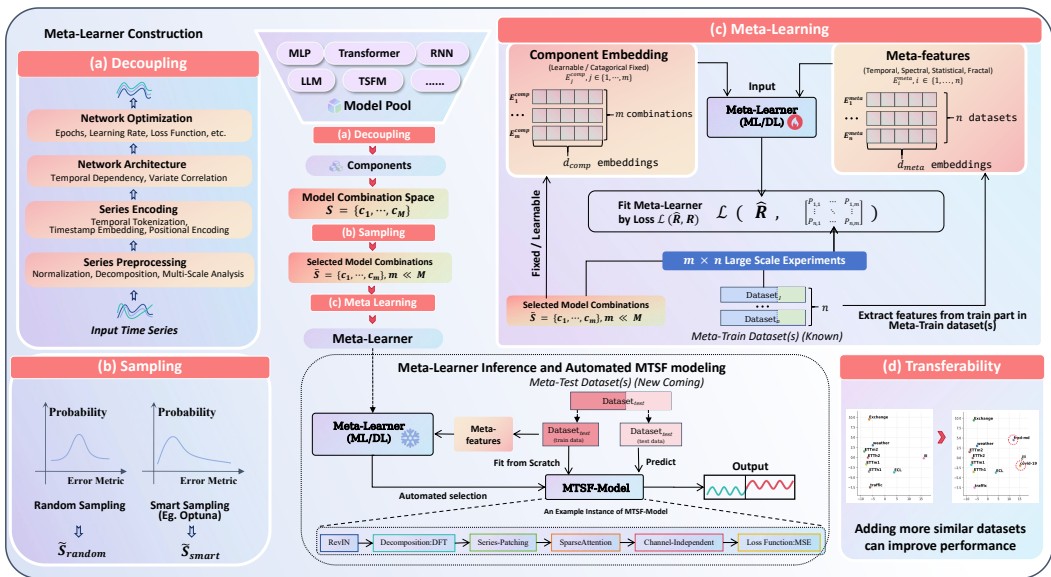

Figure 1: **Framework of the proposed TSGym.** (a) Deep MTSF models are decoupled into fine-grained components organized whithin four stages, creating a space of $M$ combinations; from which $m\,(m \ll M)$ are sampled for evaluation. (b) Sampling can be either *random* or *smart* (e.g., Optuna-guided), with the latter yielding a higher proportion of low-error combinations. (c) A meta-learner is trained on component embeddings and dataset meta-features, supervised by a performance matrix derived from *large-scale experiments* on the $m$ combinations across $n$ datasets. The trained learner then performs zero-shot selection on a new, unseen dataset to identify an optimal structure, which is subsequently trained from scratch. (d) Adding meta-train datasets closer to the target dataset(s) improves transferability, as detailed in Appx.H.3.

To bridge these gaps, we propose TSGym—a framework designed for the **Large-scale Evaluation**, **Component-level Analysis**, and **Automated Model Construction** in deep MTSF tasks. Rather than viewing models as a whole, TSGym systematically decouples popular deep MTSF methods by organizing them into distinct design dimensions involved in the time-series modeling pipeline (see Fig. 1 and Table 1). TSGym conducts fine-grained, isolated evaluations of core components through extensive experiments, thereby identifying key design dimensions/choices and valuable insights from the vast MTSF methods. Moreover, the large-scale experimental analysis in TSGym enable a systematic examination of prevailing claims within the MTSF community, addressing key questions such as the comparison between Transformer and MLP architectures and the adaptability of channel-independent approaches. Moreover, TSGym proposes the first component-level model construction in MTSF tasks, which effectively overcomes limitations in the previous automation methods by enabling more flexible and customized model designs tailored to data characteristics.

Extensive experimental results indicate that the proposed TSGym generally outperforms existing SOTA methods. We summarize the key contributions of TSGym as follows:

**Component-level Evaluation of MTSF Methods**. We propose TSGym, the first large-scale benchmark that systematically decouples deep MTSF methods. By evaluating 16 design dimensions across 10 benchmark datasets, TSGym elucidates contested issues in the current community and offers key insights to inform future development for MTSF.

**Automated MTSF Model Construction**. Leveraging meta-learning, TSGym develops models that outperform current SOTA methods, offering the MTSF community an effective, automated, and data-adaptive solution for model design. The framework is also proven to be *robust*, maintaining strong performance across various sampling strategies and meta-learner architectures, and *flexible*, systematically incorporating novel research findings like LLMs and TSFMs into its component space.

**Discussion on Emerging Large Time-series Models**. TSGym broadens current MTSF scope by applying systematic evaluation and automated combination not only to well-established models like MLP and Transformer, but also to novel large time-series models like LLMs and TSFMs.

## 2 RELATED WORK

### 2.1 DEEP LEARNING-BASED MTSF

MTSF evolves from traditional statistical methods like ARIMA and Gaussian processes to modern deep learning approaches. Recurrent Neural Networks (RNNs) introduce memory mechanisms for sequential data but struggle with long-term dependencies. Temporal Convolutional Networks (TCNs) improve this by capturing multi-scale patterns, though their fixed window sizes limit global context. Transformers, using self-attention, enable long-range forecasting but introduce high computational complexity, leading to efficient variants like sparse attention (Wu et al., 2021) and patch-based models (Nie et al., 2023). Multilayer Perceptrons (MLPs) regain attention as simple yet effective models (Zeng et al., 2023b), with numerous variants offering competitive performance (Chen et al., 2023; Yi et al., 2023; Das et al., 2023; Liu et al., 2023). Leveraging NLP foundation models, LLM adaptation approaches use frozen backbones and prompt engineering (Jin et al., 2024a; Zhou et al., 2023) or fine-tuning (Chang et al., 2023) to transfer pretrained knowledge. Simultaneously, pure TSFMs trained on large datasets achieve zero-shot generalization (Liu et al., 2024b; Goswami et al., 2024), though constrained by Transformers' complexity. Our TSGym framework modularizes six core backbones—RNNs, CNNs, Transformers, MLPs, LLMs, and TSFMs—offering flexible, hybrid integration based on temporal dependencies and resource needs.

In recent advancements in MTSF, we summarize the design paradigm through a unified pipeline (Fig. 1a), consisting of four stages: *Series Preprocessing→Series Encoding→Network Architecture→Network Optimization*. Additionally, several specialized modules are proposed to enhance predictive accuracy by addressing non-stationarity, multi-scale dependencies, and inter-variable interactions. We categorize these developments into 6 specialized modules:

**(1) Normalization** methods like RevIN (Kim et al., 2021) adjust non-stationary data, improving robustness against distribution shifts. **(2) Decomposition** methods, such as Autoformer (Wu et al., 2021)'s trend-seasonality separation, isolate non-stationary components, making the data more predictable by separating trends from seasonality. **(3) Multi-scale analysis** extracts temporal patterns across granularities, as in TimeMixer (Wang et al., 2024a), capturing both high-frequency fluctuations and low-frequency trends through hierarchical resolution modeling. **(4) Temporal tokenization** techniques like PatchTST(Nie et al., 2023)'s subseries-level embedding represent time series hierarchically, improving the capture of complex temporal semantics. **(5) Temporal dependency** modeling through architectures like Transformers leverages self-attention to capture long-range dependencies, effectively modeling both short- and long-term relationships. **(6) Variate correlation learning**, exemplified by DUET (Qiu et al., 2025b), models inter-variable dependencies using frequency-domain metric learning, improving predictions by capturing interactions across variables.

To provide a more detailed categorization and comprehensive technical specifications, please refer to Appx. B. Due to the extensive focus and continuous evolution of these modules in MTSF research, TSGym strives to decouple and modularize these key modules, exploring their real contributions and enabling more flexible model structure selection and configuration.

## 2.2 AutoML for Time Series Forecasting

Current automated approaches for DL-based MTSF can be categorized into ensemble-based (Shchur et al., 2023a) and meta-learning-based (Abdallah et al., 2022; Fischer and Saadallah, 2024) methods. The former fits and integrates various models from a predefined pool with ensemble techniques, which inevitably incurs substantial computational cost. The latter leverages meta-features to characterize datasets and selects optimal models for the given datasets. However, both approaches operate at the model level and struggle to surpass the performance ceiling of existing methods. AutoCTS++ (Wu et al., 2024) achieves automated selection by searching over model architectures and hyperparameters, but its search space is limited in scope. In contrast, TSGym is the first framework to support automated selection over a wide range of fine-grained components for MTSF, extending beyond narrow model structures, hyperparameters, and data processing strategies.

A closely related work is ADGym (Jiang et al., 2023), which is designed for tabular anomaly detection with model decomposition. Differently, TSGym deals with multivariate time series data, which presents more complex data processing design choices, such as series sampling, series normalization, and series decomposition. Second, TSGym considers finer-grained model structures, such as various attention variants in Transformers, and broader network types, including LLMs and TSFMs. Third, TSGym explores the value of Optuna (Akiba et al., 2019), a Bayesian-optimization-driven intelligent search framework, which attains a superior design space at markedly lower cost and thus enhances the efficacy of TSGym. It is worth mentioning that the success of TSGym validates the universality of the model decomposition framework, marking an innovation and progression distinct from ADGym. Further details on the differences between two works can be found in Appx.F.

## 3 TSGym: Automatic Model Design Framework for Deep MTSF

### 3.1 Problem Definition for MTSF

In this paper, we focus on the common MTSF settings for time series data containing $C$ variates. Given historical data $\chi = \{x_1^t, \ldots, x_C^t\}_{t=1}^L$, where $L$ is the look-back sequence length and $x_i^t$ is the $i$-th variate, the forecasting task is to predict $T$-step future sequence $\hat{\chi} = \{\hat{x}_1^t, \ldots, x_C^t\}_{t=L+1}^{L+T}$. To avoid error accumulation ($T > 1$), we directly predict all future steps, following (Zhou et al., 2021).

### 3.2 Decoupling Design Choices from Existing Deep MTSF Models

Under the proposed framework, the primary undertaking involves a systematic disentangling of advanced MTSF methods. By first disentangling existing models, it provides the foundation for a flexible assembly architecture and thereby facilitates a granular analysis to identify the components most responsible for performance gains.

Following the taxonomy of the previous study (Wen et al., 2022; Zeng et al., 2023b), we decouple existing SOTA methods according to the standard process of MTSF modeling, while significantly expanding the diversity of the modeling pipeline. Based on the flow direction from the input to the output sequence, the **Pipeline** includes: *Series Preprocessing→Series Encoding→Network Architecture→Network Optimization*, as is demonstrated in Fig. 1(a). Moreover, we structure each pipeline step according to distinct **Design Dimensions**, where a DL-based time-series forecasting model can be instantiated by specified **Design Choices**, as is shown in Table 1, and we provide a detailed visualization of this step-by-step workflow in Fig. 2 for clarity.

Through the proposed design dimensions and choices, TSGym provides detailed description of time-series modeling pipeline, disentangling key elements within mainstream time-series forecasting methods and facilitating component-level comparison/automated construction. For example, TSGym includes multi-scale mixing module proposed in TimeMixer (Wang et al., 2024a), Inverted Encoding method proposed in iTransformer (Liu et al., 2024a), Channel-independent strategy and Series-Patching encoding used in PatchTST (Nie et al., 2023), various attention mechanism discussed in (Wen et al., 2022), and also LLM and TSFM network type choices that are often integrated without fully considering their interactions with other design dimensions.

### 3.3 Automated MTSF model construction via TSGym

**Overview**. Differing from traditional methods that focus on selecting an off-the-shelf model, TSGym aims to customize models given the downstream MTSF tasks and data descriptions. Given a

| Pipeline | Design Dimensions | Design Choices |
|---|---|---|
| ↓Series Preprocessing | Series Normalization
Series Decomposition
Series Sampling/Mixing | w/o Norm, Stat, RevIN, DishTS
w/o Decomp, MA, MoEMA, DFT
w/ Mixing, w/o Mixing |
| ↓Series Encoding | Channel Independent
Sequence Length
Series Tokenization

Timestamp Embedding | Channel Indepen, Channel Depen
48, 96, 192, 512
Series Patching, Inverted Encoding,
Positional Encoding
w/ Embedding, w/o Embedding |
| ↓Network Architecture | Network Backbone
Series Attention

Feature Attention
Hidden Layer Dimensions
FCN Layer Dimensions
Encoder Layers | MLP, GRU, Transformer, LLM, TSFM
w/o Attn, SelfA, AutoCorr, SparseA,
FrequencyA, DestationaryA
w/o Attn, SelfA, SparseA, FrequencyA
64, 256
256, 1024
2, 3 |
| ↓Network Optimization | Training Epochs
Loss Function
Learning Rate
Learning Rate Strategy | 10, 20, 50
MSE, MAE, HUBER
1e-3, 1e-4
w/o lr Adjust, w/ lr Adjust |

Table 1: TSGym's comprehensive design choices for deep MTSF. "A" represents Attention mechanism.

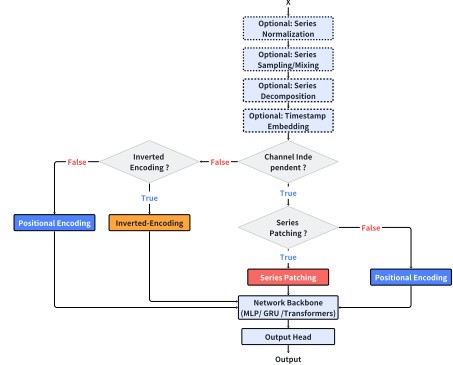

Figure 2: Workflow of TSGym designed model.

pre-defined conflict-free model set $\mathcal{M} = \{M_1, ..., M_m\}$, each model $M_i$ is instantiated by the design choice combinations illustrated in Table 1. TSGym learns the mapping function from these automatically combined models to their associated forecasting performance on the training datasets, and generalize to the test dataset(s) to select the best model based on predicted results.

**Meta-learning for automated MTSF model construction**. Formally speaking, TSGym propose $k$ design dimensions $\mathcal{DD} = \{DD_1, ..., DD_k\}$ for comprehensively describing each step of aforementioned pipeline in deep learning time-series modeling. Each design dimension $DD_i$ represents a set containing elements of different design choices $DC$. By taking the Cartesian product of the sets $\mathcal{DD}$ corresponding to different design dimensions, we obtain the pool of all valid model combinations $\mathcal{M} = DD_1 \times DD_2 \times \cdots \times DD_k = \{(DC_1, DC_2, \ldots, DC_k) \mid DC_i \in DD_i, i = 1, 2, \ldots, k\}$. Considering the potentially large number of combinations and the computational cost, we randomly sampled $\mathcal{M}$ to $\mathcal{M}_s$, where $M_i = (DC_1 = RevIN, DC_1 = DFT, ..., DC_k = Type1) \in \mathcal{M}_s$, for example, which means $M_i$ instantiates RevIN method to normalize input series, then decompose it to the seasonal and trend term. Subsequently, following the Series Encoding and Network Architecture constructing pipeline (as illustrated in Table 1), finally the Type1, i.e., a step decay learning rate strategy is employed to adjust the learning rate for updating the model parameters.

Suppose we have $n$ training datasets $\mathcal{D}_{\text{train}} = \{\mathcal{D}_1, \ldots, \mathcal{D}_n\}$ and the number of sampled model combinations (i.e., the size of the set $\mathcal{M}_s$ is $m$, TSGym conducts extensive experiments on $n$ historical training datasets to evaluate and further collect the forecasting performance of $m$ model combinations. TSGym then acquire the MSE performance matrix $\boldsymbol{P} \in \mathbb{R}^{n \times m}$, where $\boldsymbol{P}_{i,j}$ corresponds to the $j$-th auto-constructed MTSF model's performance on the $i$-th training dataset. Since the difficulty of prediction tasks varies across training datasets, leading to significant differences in the numerical range of performance metrics. Directly using these metrics (e.g., MSE) as training targets of a meta-predictor may result in overfitting on more difficult dataset(s). Therefore, we convert the performance metrics of $\mathcal{M}_s$ into their corresponding normalized ranking, where $\boldsymbol{R}_{i,j} = rank(P_{i,j})/m \in [0, 1]$ and smaller values indicate better performance on the corresponding dataset.

Distinguished from previous model selection approaches (Abdallah et al., 2022; 2025), TSGym decouples more recently MTSF methods (including MLP-Mixer-type, Transformer-based, LLM and TSFM models), and supports fine-grained model construction at the component level, rather than being constrained to a fixed, limited set of existing models, which enables significantly greater flexibility and effectiveness. Specifically, TSGym follows the idea of meta-learning to construct a meta-predictor that learns the mapping function $f(\cdot)$ from training dataset $\mathcal{D}_i$ and model combination $M_j$, to the performance rankings $\boldsymbol{R}_{i,j}$, as is shown in Eq. 1, where the meta-features $\mathbf{E}_i^{meta}$ capture multiple aspects such as statistical, temporal, spectral, and fractal features to fully describe the complex data characteristics of time series datasets. Learnable continuous embeddings $\mathbf{E}_j^{comp}$ are used to represent different model combinations and are updated through the gradient backpropagation of the meta-predictor. This process enables efficient zero-shot inference at test time: for any new dataset, the meta-predictor can identify a top-performing model configuration using only the meta-features extracted from the training data, eliminating the need for any costly experimental trials.

$$f(\mathcal{D}_i, M_j) = \boldsymbol{R}_{i,j}, f : \underbrace{\mathbf{E}_i^{meta}}_{\text{meta features}}, \quad \underbrace{\mathbf{E}_j^{comp}}_{\text{component embed.}} \mapsto \boldsymbol{R}_{i,j}, \quad i \in \{1, \ldots, n\}, j \in \{1, \ldots, m\} \quad (1)$$

We used a simple two-layer MLP as the meta-predictor and trained it through a regression problem, thereby transferring the learned mapping to new test datasets. For a newcoming dataset (i.e., test dataset $\mathbf{X}_{\text{test}}$), we acquire the predicted relative ranking of different components using the trained $f(\cdot)$, and select top-1 ($k$) to construct MTSF model(s). Note this procedure is zero-shot without needing any neural network training on $\mathbf{X}_{\text{test}}$ but only extracting meta-features and pipeline embeddings. We show the effectiveness of the meta-predictor in §4.3.

# 4 EXPERIMENTS

## 4.1 EXPERIMENT SETTINGS

**Datasets**. Following most prior works (Wu et al., 2021; 2023; Jin et al., 2024a), we adopt 9 datasets as experimental data for MTSF tasks, ETT (4 subsets), Traffic, Electricity, Weather, Exchange, ILI. And we utilize the M4 dataset for short-term forecasting tasks. The forecast horizon $L$ for long-term forecasting is $\{96, 192, 336, 720\}$, while for the ILI dataset, it is $\{24, 36, 48, 60\}$. For short-term forecasting, the forecast horizons are $\{6, 8, 13, 14, 18, 48\}$. More details can be seen in Appx. A.

**Baseline**. We compare TSGym against a comprehensive set of baselines, including MTSF and AutoML methods, to demonstrate the superior performance of the pipelines automatically constructed by TSGym. Due to space limitations, the baseline methods presented in this section include the latestapproach DUET (Qiu et al., 2025b), TimeMixer (Wang et al., 2024a), MICN (Wang et al., 2023), SegRNN (Lin et al., 2023), TimesNet (Wu et al., 2023), PatchTST (Nie et al., 2023), Crossformer (Zhang and Yan, 2023), and Autoformer (Wu et al., 2021). We present experiments based on the complete baseline in the Appx. H.

**Evaluation Metrics**. We follow the experimental setup of most prior works, using Mean Squared Error (MSE) and Mean Absolute Error (MAE) as evaluation metrics for long-term forecasting tasks, and using Symmetric Mean Absolute Percentage Error (SMAPE), Mean Absolute Scaled Error (MASE), and Overall Weighted Average (OWA) as metrics for short-term forecasting tasks. The mathematical formulas for these evaluation metrics are provided in the Appx. D.

**Meta-predictor in TSGym**. The meta-predictor is instantiated as a two-layer MLP and trained for 100 epochs with early stopping. The training process utilizes the Adam optimizer (Kingma and Ba, 2014) with a learning rate of 0.001 and batch size of 512. See details in Appx. G.

## 4.2 LARGE-SCALE COMPONENTS-LEVEL ANALYSIS WITH TSGYM

In this work, we perform large evaluations on the decoupled pipelines according to the standard procedure of MTSF methods. Such analysis is often overlooked in previous studies, and we investigate each design dimension of decoupled pipelines by fixing its corresponding design choice (e.g., Self Attention), and randomly sampling other dimensional design choices to construct MTSF pipelines.

In the following sections, we formulate 4 most contentious claims in the MTSF reaserch community and clarify them with our framework. All our conclusions are drawn from 18 experimental settings, spanning nine distinct datasets and two evaluation metrics. These results are presented in Table H10, H11, H12, and H13. Leveraging our open-source framework and the accompanying large-scale experimental results, researchers can explore additional findings of interest beyond those reported in our paper.

**Claim: Transformers are less robust than MLPs. Shao et al. (2024)**

**Yes.** Using the inter-quartile range (IQR) as the robustness metric, Transformers perform worse than MLPs in 13 out of 18 settings, with an average IQR of 0.391—significantly higher than the 0.275 average IQR of MLPs.

**Claim: Transformers exhibit a higher upper bound than MLPs. Shao et al. (2024)**

**No.** Taking the best performance of each MLP and Transformer variant across all pipelines as their respective upper-bound estimate, we observe average upper-bound metrics of 0.406 and 0.408 over the 18 settings, with MLPs attaining the higher bound in 11 of them. This indicates that Transformers do not demonstrate a superior model capacity upper bound.

**Claim: Novel attention mechanisms outperform vanilla self-attention.**

Table 2: Long-term forecasting task. The past sequence length is set as 36 for ILI and 96 for the others. All the results are averaged from 4 different prediction lengths, that is {24, 36, 48, 60} for ILI and {96, 192, 336, 720} for the others. See Table in Appendix for the full results.

| Models | TSGym (Ours) | | DUET (Qiu et al., 2025b) | | TimeMixer (Wang et al., 2024a) | | MICN (Wang et al., 2023) | | TimesNet (Wu et al., 2023) | | PatchTST (Nie et al., 2023) | | DLinear (Zeng et al., 2023b) | | Crossformer (Zhang and Yan, 2023) | | Autoformer (Lin et al., 2023) | | SegRNN (Wu et al., 2021) | |
|---|---|---|---|---|---|---|---|---|---|---|---|---|---|---|---|---|---|---|---|---|
| Metric | MSE | MAE | MSE | MAE | MSE | MAE | MSE | MAE | MSE | MAE | MSE | MAE | MSE | MAE | MSE | MAE | MSE | MAE | MSE | MAE |
| ETTm1 | **0.362** | **0.380** | 0.407 | 0.409 | 0.384 | 0.399 | 0.402 | 0.429 | 0.432 | 0.430 | 0.390 | 0.404 | 0.404 | 0.407 | 0.501 | 0.501 | 0.532 | 0.496 | 0.388 | 0.404 |
| ETTm2 | **0.266** | **0.322** | 0.296 | 0.338 | 0.277 | 0.325 | 0.342 | 0.391 | 0.296 | 0.334 | 0.288 | 0.334 | 0.349 | 0.399 | 1.487 | 0.789 | 0.330 | 0.368 | 0.273 | 0.322 |
| ETTh1 | 0.427 | 0.439 | 0.433 | 0.437 | 0.448 | 0.438 | 0.589 | 0.537 | 0.474 | 0.464 | 0.454 | 0.449 | 0.465 | 0.461 | 0.544 | 0.520 | 0.492 | 0.485 | **0.422** | **0.429** |
| ETTh2 | **0.367** | **0.403** | 0.380 | 0.403 | 0.383 | 0.406 | 0.585 | 0.530 | 0.415 | 0.424 | 0.385 | 0.409 | 0.566 | 0.520 | 1.552 | 0.908 | 0.446 | 0.460 | 0.374 | 0.405 |
| ECL | **0.164** | **0.261** | 0.179 | 0.262 | 0.185 | 0.273 | 0.186 | 0.297 | 0.219 | 0.314 | 0.209 | 0.298 | 0.225 | 0.319 | 0.193 | 0.289 | 0.234 | 0.340 | 0.216 | 0.302 |
| Traffic | **0.433** | **0.301** | 0.797 | 0.427 | 0.496 | 0.313 | 0.544 | 0.320 | 0.645 | 0.348 | 0.497 | 0.321 | 0.673 | 0.419 | 1.458 | 0.782 | 0.637 | 0.397 | 0.807 | 0.411 |
| Weather | **0.240** | 0.276 | 0.252 | 0.277 | 0.244 | **0.274** | 0.264 | 0.316 | 0.261 | 0.287 | 0.256 | 0.279 | 0.265 | 0.317 | 0.253 | 0.312 | 0.339 | 0.379 | 0.251 | 0.298 |
| Exchange | 0.375 | 0.415 | **0.322** | **0.384** | 0.359 | 0.402 | 0.346 | 0.422 | 0.405 | 0.437 | 0.381 | 0.412 | 0.346 | 0.414 | 0.904 | 0.695 | 0.506 | 0.500 | 0.408 | 0.423 |
| ILI | 2.463 | 1.043 | 2.640 | 1.018 | 4.502 | 1.557 | 2.938 | 1.178 | **2.140** | 0.907 | 2.160 | **0.901** | 4.367 | 1.540 | 4.311 | 1.396 | 3.156 | 1.207 | 4.305 | 1.397 |
| 1ˢᵗ Count | 11 | | 2 | | 1 | | 0 | | 1 | | 1 | | 0 | | 0 | | 0 | | 2 | |

Table 3: Short-term forecasting task on M4. The results are averaged from several datasets under different sample intervals. See Table in Appendix for the full results.

| Models | TSGym (ours) | TimeMixer | MICN | TimesNet | PatchTST | DLinear | Crossformer | Autoformer | SegRNN |
|---|---|---|---|---|---|---|---|---|---|
| OWA | **0.856** | 0.884 | 0.984 | 0.907 | 0.965 | 0.922 | 8.856 | 1.273 | 1.007 |
| sMAPE | **11.781** | 11.985 | 13.025 | 12.199 | 12.848 | 12.511 | >30 | 16.392 | 13.509 |
| MASE | **1.551** | 1.615 | 1.839 | 1.662 | 1.738 | 1.693 | >10 | 2.317 | 1.823 |

**Yes.** Among the configurations equipped with attention modules, the vanilla self-attention mechanism ranks first in only one of the 18 experimental settings. Although Auto-Correlation exhibits similarly poor performance, the majority of novel attention mechanisms consistently outperform the vanilla self-attention.

**Claim: Novel sequence encodings outperform the classic series encoding. Chen et al. (2025)**

**Yes.** Across the 18 experimental settings, classic positional encoding never achieves the best performance, recording a mean median error of 0.605. Inverted encoding and series patching achieve 0.558 and 0.549, respectively, with the latter ranking first in 15 settings.

## 4.3 EFFECTIVENESS OF AUTOMATED MODEL CONSTRUCTION VIA TSGYM

Extensive experimental results discussed above indicate that in deep time series modeling, most design choices are determined by data characteristics, meaning one-size-fits-all approaches are seldom effective. This, in turn, emphasizes the necessity of automated model construction.

In this subsection, we compare the MTSF pipeline selected by TSGym with existing SOTA methods. Through large-scale experiments, we found that TSGym outperforms existing SOTA models in both long- and short-term MTSF tasks. Regarding algorithm efficiency, our experiments demonstrate that even when limited to a search pool of lightweight model structures, such as MLP and RNNs, TSGym can still achieve competitive results. We analyze the effectiveness of the pipelines automatically constructed by TSGym through five key questions as follows. Additional details, such as the results based on more metrics and more complex meta-features, can be found in the Appx.H.

**Question 1: Is the model constructed by meta-predictor better than existing SOTA methods?**

Comprehensive forecasting results in Tables 2 and 3 highlight the best performances in red and second-best in blue. Compared to state-of-the-art forecasters, TSGym outperforms others across multiple datasets, achieving the lowest MSE and MAE 11 times, demonstrating strong generalization ability over medium and long forecasting horizons. While models like DUET, TimeMixer, and SegRNN show competitive results on certain datasets, TSGym generally outperforms them, especially in short-term forecasting tasks. As for short-term forecasting tasks, both TSGym and TimeMixer demonstrate competitive performance, with TSGym outperforming on most evaluation metrics.

**Question 2: Is TSGym with lightweight architecture better than existing SOTA methods?**

In the previous section, we compared TSGym using the full component pool with SOTA and found that TSGym outperforms SOTA on several datasets. In this ablation experiment Table 4, we specifically compare the `-Transformer` configuration of TSGym with DUET. Remarkably, even after removing Transformer-related components from the TSGym component pool and retaining only the more computationally efficient MLP- and RNN-based models, TSGym still outperforms DUET on the majority of datasets. This demonstrates the robustness and efficiency of TSGym's architecture and highlights the strong predictive power of the simplified MLP-based design.

## Question 3: Does the training strategies bring significant improvement for TSGym?

Following Table 4, we find that the `+AllPL` configuration, which trains on datasets with varying prediction lengths and transfers this knowledge to a test set with a single prediction length, further improves generalization, with the best performance observed on the ETTm1 dataset. Additionally, removing the Transformer component (`-Transformer`) leads to performance gains on certain datasets, suggesting that a simplified MLP- or RNN-based architecture can be more effective in specific scenarios. These results highlight the flexibility of TSGym's design and the potential benefits of customizing the component pool to suit dataset characteristics.

Table 4: Ablation study evaluates the removal of Transformer-based components and different training strategies, and the final row shows how often TSGym variants outperform DUET.

| Models | TSGym | | -Transformer | | +AllPL | | DUET | |
|---|---|---|---|---|---|---|---|---|
| Metric | MSE | MAE | MSE | MAE | MSE | MAE | MSE | MAE |
| ETTm1 | 0.362 | 0.380 | 0.370 | 0.394 | **0.359** | **0.379** | 0.407 | 0.409 |
| ETTm2 | **0.266** | 0.322 | 0.269 | **0.322** | 0.271 | 0.328 | 0.296 | 0.338 |
| ETTh1 | 0.427 | 0.439 | **0.408** | **0.423** | 0.430 | 0.437 | 0.433 | 0.437 |
| ETTh2 | 0.367 | 0.403 | **0.357** | **0.392** | 0.368 | 0.399 | 0.380 | 0.403 |
| ECL | **0.164** | **0.261** | 0.172 | 0.269 | 0.169 | 0.266 | 0.179 | 0.262 |
| Traffic | **0.433** | 0.301 | 0.456 | 0.313 | 0.446 | **0.298** | 0.797 | 0.427 |
| Weather | 0.240 | 0.276 | 0.238 | 0.271 | **0.236** | 0.276 | 0.252 | 0.277 |
| Exchange | 0.375 | 0.415 | 0.407 | 0.429 | 0.421 | 0.433 | **0.322** | **0.384** |
| ILI | 2.463 | 1.043 | 2.654 | 1.111 | 2.490 | 1.055 | 2.640 | **1.018** |
| Better Count | 13/18 | | 13/18 | | 13/18 | | - | |

## Question 4: Does large time-series models bring significant improvement for TSGym?

Table 5 evaluates the impact of incorporating LLM and TSFM into the base TSGym framework. The introduction of LLM consistently improve forecasting accuracy compared to the baseline TSGym configuration and the addition of TSFM offers some improvements for certain datasets. However, the improvements are not uniform across all datasets, suggesting that further refinement is needed to optimize their impact on MTSF.

Table 5: Ablation study of TSGym incorporating LLM and TSFM in **four datasets**.

| Models | TSGym | | +LLM | | +TSFM | |
|---|---|---|---|---|---|---|
| Metric | MSE | MAE | MSE | MAE | MSE | MAE |
| ETTh1 | 0.439 | 0.453 | **0.431** | **0.441** | 0.476 | 0.465 |
| ETTh2 | **0.356** | 0.396 | 0.362 | **0.395** | 0.399 | 0.418 |
| Exchange | **0.382** | **0.418** | 0.388 | 0.419 | 0.684 | 0.482 |
| ILI | 3.092 | 1.199 | 2.830 | 1.128 | **2.656** | **1.105** |

## Question 5: Does smarter sampling strategy bring improvement for TSGym?

As shown in Table 6, incorporating Optuna, a Bayesian optimization-based sampling method, improves or maintains meta-learner performance on nearly all datasets. Fig.3 illustrates that while random sampling produces a broad distribution dominated by mediocre configurations, Optuna shifts sampling toward low-error regions, increasing the share of high-quality components. By replacing a part of the random pool with these Optuna-sampled configurations while keeping the total pool size fixed, we enhance average quality without losing diversity, leading to the observed performance gains. Further details regarding the Optuna sampling setup are provided in Appx. H.3.

| Sampling Strategies | Random | | +Optuna | |
|---|---|---|---|---|
| Metric | MSE | MAE | MSE | MAE |
| ETTm1 | 0.370 | 0.394 | **0.354** | **0.379** |
| ETTm2 | 0.269 | 0.322 | **0.258** | **0.313** |
| ETTh1 | **0.408** | **0.423** | 0.431 | 0.441 |
| ETTh2 | 0.357 | 0.392 | **0.355** | **0.389** |
| ECL | 0.172 | 0.269 | **0.170** | **0.265** |
| Traffic | 0.456 | 0.313 | **0.427** | **0.295** |
| Weather | **0.238** | **0.271** | 0.240 | 0.280 |
| Exchange | 0.407 | 0.429 | **0.402** | **0.427** |
| ILI | 2.654 | 1.111 | **2.488** | **1.053** |

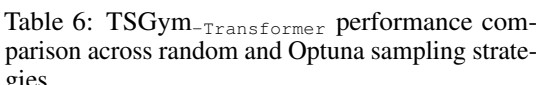

Table 6: TSGym$_{-Transformer}$ performance comparison across random and Optuna sampling strategies.

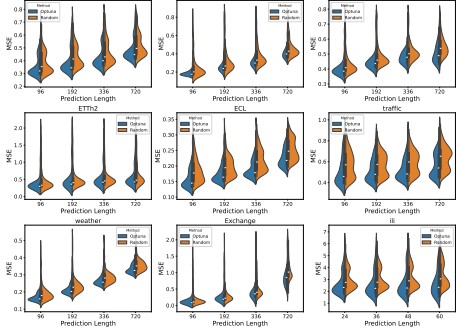

Figure 3: Distribution of model performance selected by two sampling strategies.

### 4.4 COMPARATIVE EXPERIMENTS WITH AUTOML METHODS

Establishing a meaningful benchmark requires selecting comparable frameworks. While many general-purpose AutoML libraries exist (e.g., TPOT (Olson et al., 2016), H2O-3 (H2O.ai, 2022),

Microsoft NNI (Microsoft, 2021), Auto-Keras (Jin et al., 2023a), Auto-Sklearn (Feurer et al., 2022), NASLib (Ruchte et al., 2020)), most are not designed for time-series forecasting. Adapting them for MTSF would be burdensome and potentially leading to an unfair evaluation. Consequently, we benchmarked the two prominent AutoML libraries that explicitly support MTSF on both short-term and long-term forecasting tasks: AutoGluon-TimeSeries (Shchur et al., 2023b) and AutoTS (Catlin, 2020). The dataset partitioning scheme was identical to that used for TSGym. To manage computational demands, AutoGluon was configured with the "high_quality" preset and AutoTS with its "superfast" setting, while all other hyperparameters were maintained at their default values.

For short-term forecasting (Table 7), TSGym demonstrates clear superiority, achieving the best scores across all three metrics: Overall Weighted Average (OWA), Symmetric Mean Absolute Percentage Error (SMAPE), and Mean Absolute Scaled Error (MASE). This indicates a robust and consistently better performance in short-horizon predictions compared to the established AutoML baselines.

Table 7: Short-term forecasting comparison with AutoML methods.

| Model | TSGym (ours) | AutoGluon | AutoTS |
|---|---|---|---|
| **OWA** | **0.856** | 0.950 | 2.002 |
| **SMAPE** | **11.781** | 13.178 | 18.977 |
| **MASE** | **1.551** | 1.775 | 4.981 |

In the more challenging long-term forecasting tasks (Table 8), TSGym continues to show a strong competitive advantage. It secures the lowest (best) Mean Squared Error (MSE) and Mean Absolute Error (MAE) on the majority of datasets, including ETTm1, ETTm2, ETTh1, ETTh2, ECL, and Weather. It is worth noting that AutoGluon achieves better performance on the Exchange dataset and a lower MAE on the ILI dataset, while AutoTS shows a competitive MAE on the Traffic dataset. Nevertheless, TSGym's dominant performance across a wide range of datasets underscores its effectiveness and robustness for long-horizon prediction.

Table 8: Long-term forecasting comparison with AutoML methods.

| Dataset | TSGym (ours) | | AutoGluon | | AutoTS | |
|---|---|---|---|---|---|---|
| | MSE | MAE | MSE | MAE | MSE | MAE |
| ETTm1 | **0.362** | **0.380** | 0.482 | 0.408 | 0.744 | 0.546 |
| ETTm2 | **0.266** | **0.322** | 0.273 | 0.337 | 0.392 | 0.389 |
| ETTh1 | **0.427** | **0.439** | 0.503 | 0.473 | 0.981 | 0.61 |
| ETTh2 | **0.367** | **0.403** | 0.419 | 0.43 | 0.589 | 0.488 |
| ECL | **0.164** | **0.261** | 0.265 | 0.328 | 0.327 | 0.355 |
| Traffic | **0.433** | **0.301** | 0.555 | 0.325 | 0.739 | 0.311 |
| Weather | 0.240 | 0.276 | **0.236** | **0.27** | 0.519 | 0.372 |
| Exchange | 0.375 | 0.415 | **0.33** | **0.393** | 0.588 | 0.494 |
| ILI | 2.463 | 1.043 | **2.271** | **0.979** | 2.533 | 1.049 |

As the results indicate, TSGym consistently and significantly outperforms both AutoGluon and AutoTS on the vast majority of datasets, in both short-term and long-term forecasting tasks. The superiority of TSGym for multivariate time series forecasting lies in its distinct methodology, which automates model construction through fine-grained component decomposition and meta-learning.

## 5 CONCLUSIONS, LIMITATIONS, AND FUTURE DIRECTIONS

To advance beyond holistic evaluations in multivariate time-series forecasting (MTSF), this paper introduced TSGym, a novel framework centered on fine-grained component analysis and the automated construction of specialized forecasting models. By systematically decomposing MTSF pipelines into design dimensions and choices informed by recent studies, TSGym uncovers crucial insights into component-level forecasting performance and leverages meta-learning method for the automated construction of customized models. Extensive experimental results indicate that the MTSF models constructed by the proposed TSGym significantly outperform current MTSF SOTA solutions—demonstrating the advantage of adaptively customizing models according to distinct data characteristics. Our results show that TSGym is highly effective, even without exhaustively covering all SOTA components, and TSGym is made publicly available to benefit the MTSF community.

Future efforts will focus on expanding TSGym's range of forecasting techniques with emerging techniques and refining its meta-learning capabilities by incorporating multi-objective optimization to balance predictive performance against computational costs, especially for large time-series models, while also broadening its applicability across diverse time series analysis tasks.

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

## APPENDIX

For further details, we provide more information in the Appendix, including the evaluated 10 datasets (§A), key modules (§B), compared baselines (§C), metrics mathematical formula (§D), system configuration (§E), ADGym comparison analysis (§F), the details of proposed TSGym (§G), and additional experimental results (§H).

## A  DATASET LIST

We conduct extensive evaluations on nine standard long-term forecasting benchmarks - four ETT variants (ETTh1, ETTh2, ETTm1, ETTm2), Electricity (abbreviated as ECL), Traffic, Weather, Exchange, and ILI, complemented by the M4 dataset for short-term forecasting tasks, with complete dataset specifications provided in Table A1.

Table A1: Data description of the 12 datasets included in TSGym.

| Task | Dataset | Domain | Frequency | Lengths | Dim | Description |
|------|---------|--------|-----------|---------|-----|-------------|
|      | ETTh1 | Electricity | 1 hour | 14,400 | 7 | Power transformer 1, comprising seven indicators such as oil temperature and useful load |
|      | ETTh2 | Electricity | 1 hour | 14,400 | 7 | Power transformer 2, comprising seven indicators such as oil temperature and useful load |
|      | ETTm1 | Electricity | 15 mins | 57,600 | 7 | Power transformer 1, comprising seven indicators such as oil temperature and useful load |
|      | ETTm2 | Electricity | 15 mins | 57,600 | 7 | Power transformer 2, comprising seven indicators such as oil temperature and useful load |
|      | ECL | Electricity | 1 hour | 26,304 | 321 | Electricity records the electricity consumption in kWh every 1 hour from 2012 to 2014 |
| LTF  | Traffic | Traffic | 1 hour | 17,544 | 862 | Road occupancy rates measured by 862 sensors on San Francisco Bay area freeways |
|      | Weather | Environment | 10 mins | 52,696 | 21 | Recorded every for the whole year 2020, which contains 21 meteorological indicators |
|      | Exchange | Economic | 1 day | 7,588 | 8 | ExchangeRate collects the daily exchange rates of eight countries |
|      | ILI | Health | 1 week | 966 | 7 | Recorded indicators of patients data from Centers for Disease Control and Prevention |
|      | Covid-19 | Health | 1 day | 1,392 | 948 | Provide opportunities for researchers to investigate the dynamics of COVID-19 |
|      | FRED-MD | Economic | 1 month | 728 | 107 | Time series showing a set of macroeconomic indicators from the Federal Reserve Bank |
| STF  | M4 | Demographic, Finance, Industry, Macro, Micro and Other | Yearly Quarterly Monthly Weakly Daily Hourly | 19-9933 | 100000 | M4 competition dataset containing 100,000 unaligned time series with varying lengths and time periods |

## B  KEY MODULES

Modern deep learning for MTSF utilizes several specialized modules to tackle non-stationarity, multi-scale dependencies, and inter-variable interactions. In this section, we analyze the design and efficacy of prevalent specialized modules adopted in state-of-the-art models (Fig. 1a).

**Normalization modules** address temporal distribution shifts through adaptive statistical alignment. While z-score normalization employs fixed moments, modern techniques enhance adaptability: RevIN (Kim et al., 2021) introduces learnable affine transforms with reversible normalization/denormalization; Dish-TS (Fan et al., 2023) decouples inter-/intra-series distribution coefficients; Non-Stationary Transformer (Liu et al., 2022d) integrates statistical moments into attention via de-stationary mechanisms. These methods balance stationarized modeling with inherent non-stationary dynamics.

**Decomposition methods**, standard in time series analysis, break down series into components like trend and seasonality to improve predictability and handle distribution shifts. **(1) Time-domain decomposition** utilizes moving average operations to isolate slowly-varying trends from high-frequency fluctuations that represent seasonality (e.g., DLinear (Zeng et al., 2023b), Autoformer, FEDformer). **(2) Frequency-domain decomposition** partitions series via Discrete Fourier Transform (DFT), assigning low-frequency spectra to trends and high-frequency bands to seasonality, which is applied in the Koopa (Liu et al., 2023) model.

**Multi-Scale modeling** addresses the inherent temporal hierarchy in time series data, where patterns manifest differently across various granularities (e.g., minute-level fluctuations vs. daily trends). Pyraformer (Liu et al., 2022b) integrates multi-convolution kernels via pyramidal attention to establish hierarchical temporal dependencies. FEDformer (Zhou et al., 2022b) employs mixed experts to combine trend components from multiple pooling kernels with varying receptive fields, where larger kernels capture macro patterns while smaller ones preserve local details. TimeMixer (Wang et al., 2024a) extends this paradigm through bidirectional mixing operations - upward propagation refines fine-scale seasonal features while downward aggregation consolidates coarse-scale trends. FiLM (Zhou et al., 2022a) dynamically adjusts temporal resolutions through learnable lookback windows, enabling adaptive focus on relevant historical contexts across scales. Crossformer (Zhang and Yan, 2023) implements flexible patchsize configurations, where multi-granular patches independently model short-term fluctuations and long-term cycles through dimension-aware processing.

**Temporal Tokenization strategies**, originating from Transformers (Wang et al., 2024b; Liu et al., 2024a) and now extended to RNNs (Lin et al., 2023), vary by temporal representation granularity: **(1) Point-wise** methods

(e.g., Informer (Zhou et al., 2021), Pyraformer (Liu et al., 2022b)) process individual timestamps as tokens. They offer temporal precision but face quadratic complexity, requiring attention sparsification that may hinder long-range dependency capture. **(2) Patch-wise** strategies (e.g., PatchTST (Nie et al., 2023)) aggregate local temporal segments into patches. Pathformer (Chen et al., 2024) similarly employs patch-based processing via adaptive multi-scale pathways. **(3) Series-wise** approaches (e.g., iTransformer (Liu et al., 2024a)) construct global variate representations, enabling cross-variate modeling but risking temporal misalignment. TimeXer (Wang et al., 2024c) uses hybrid tokenization: patch-level for endogenous variables and series-level for exogenous, bridged by a learnable global token.

**Temporal Dependency Modeling** captures dynamic inter-step dependencies through diverse architectural mechanisms, balancing local interactions and global patterns. Recurrent state transitions (e.g., LSTM) model sequential memory via gated memory cells; temporal convolutions (e.g., TCN (Bai et al., 2018)) construct multi-scale receptive fields using dilated kernels; attention mechanisms (e.g., Transformers) enable direct pairwise interactions across arbitrary time steps. Efficiency-driven innovations include sparse attention (Informer (Zhou et al., 2021)), periodicity-based aggregation (Autoformer (Wu et al., 2021)), and state-space hybrids (Mamba (Gu and Dao, 2024)), achieving tractable long-range dependency modeling while preserving temporal fidelity.

**Variate Correlation**, fundamental to modeling critical correlations in multivariate time series forecasting (MTSF), operates through two primary paradigms (Qiu et al., 2025b): **(1) Channel-Independent (CI) Strategy**: Processes channels independently with shared parameters (e.g., PatchTST (Nie et al., 2023)), ensuring robustness and efficiency but ignoring multivariate dependencies, limiting use with strong inter-channel interactions (Qiu et al., 2025a). **(2) Channel-Dependent (CD) Strategy**: Integrates channel information via methods like channel-wise self-attention (iTransformer (Liu et al., 2024a)) or MLP-based mixing (TSMixer (Chen et al., 2023)). This allows explicit dependency modeling but risks overfitting and struggles with noise in high dimensions.

## C    COMPARED BASELINES

We systematically compare state-of-the-art forecasting models using the 6 architectural modules introduced in Section B. Table C2 presents the configuration of each baseline in terms of these modules. The "Notes" column provides concise annotations of each model's key methodological features, allowing for quick identification of the technical differentiators among the baselines.

## D    METRICS MATHEMATICAL FORMULA

The metrics used in this paper can be calculated as follows(Wu et al., 2023):

$$\text{MSE} = \frac{1}{H}\sum_{i=1}^{H}(\mathbf{X}_i - \widehat{\mathbf{X}}_i)^2, \qquad\qquad \text{MAE} = \frac{1}{H}\sum_{i=1}^{H}|\mathbf{X}_i - \widehat{\mathbf{X}}_i|,$$

$$\text{SMAPE} = \frac{200}{H}\sum_{i=1}^{H}\frac{|\mathbf{X}_i - \widehat{\mathbf{X}}_i|}{|\mathbf{X}_i| + |\widehat{\mathbf{X}}_i|}, \qquad\qquad \text{MAPE} = \frac{100}{H}\sum_{i=1}^{H}\frac{|\mathbf{X}_i - \widehat{\mathbf{X}}_i|}{|\mathbf{X}_i|},$$

$$\text{MASE} = \frac{1}{H}\sum_{i=1}^{H}\frac{|\mathbf{X}_i - \widehat{\mathbf{X}}_i|}{\frac{1}{H-m}\sum_{j=m+1}^{H}|\mathbf{X}_j - \mathbf{X}_{j-m}|}, \qquad \text{OWA} = \frac{1}{2}\left[\frac{\text{SMAPE}}{\text{SMAPE}_{\text{Naïve2}}} + \frac{\text{MASE}}{\text{MASE}_{\text{Naïve2}}}\right],$$

where $m$ is the periodicity of the data. $\mathbf{X}, \widehat{\mathbf{X}} \in \mathbb{R}^{H \times C}$ are the ground truth and prediction results of the future with $H$ time points and $C$ dimensions. $\mathbf{X}_i$ means the $i$-th future time point.

## E    SYSTEM CONFIGURATION

We conducted all experiments in the same experimental environment, which includes four NVIDIA A100 GPUs with 80GB and eight 40GB of memory. We saved overall experimental time by running experiments in parallel.

## F    COMPARED WITH ADGYM

Compared with ADGym (Jiang et al., 2023), TSGym exhibits the following differences and advantages:

**(1) Broader model structure design choices**. ADGym includes only MLP, autoencoder (AE), ResNet, and Transformer architectures, while TSGym provides an in-depth decoupling of different attention mechanisms within Transformers and incorporates two pre-trained large models: LLMs and TSFM. **(2) More diverse data processing design choices**. ADGym focuses solely on data augmentation and two normalization methods,

Table C2: Component Configurations of 27 Baseline Models

| Backbone | Method | Normali-zation | Decom-position | Multi-Scale | Token-izations | Temporal Dependency | Variate Corre-lation | Notes |
|---|---|---|---|---|---|---|---|---|
| RNN | SegRNN(Lin et al., 2023) | SubLast | | | Patch-wise | GRU | CI | Reduces iterations via patch-wise processing and parallel multi-step forecasting. |
| | Mamba(Gu and Dao, 2024) | Stat | | | Point-wise | Selective State Space Model | CD | Efficient model selectively propagating information without attention or MLP blocks. |
| CNN | SCINet(Liu et al., 2022a) | Stat | | TRUE | Point-wise | Conv1d | CD | Recursively downsamples, convolves, and interacts with data to capture complex temporal dynamics. |
| | MICN(Wang et al., 2023) | | MA | TRUE | Point-wise | Conv1d | CD | Combines local features and global correlations using multi-scale convolutions with linear complexity. |
| | TimesNet(Wu et al., 2023) | Stat | | TRUE | Point-wise | Conv2d | CD | Transforms 1D time series into 2D tensors to capture multi-periodicity and temporal variations. |
| MLP | FiLM(Zhou et al., 2022a) | RevIN | | TRUE | Point-wise | Legendre Projection Unit | CD | Preserves historical info and reduces noise with Legendre and Fourier projections. |
| | LightTS(Zhang et al., 2022) | | | | Patch-wise | MLP | CD | Lightweight MLP model for multivariate forecasting, using continuous and interval sampling for efficiency. |
| | DLinear(Zeng et al., 2023b) | | MA | | Point-wise | MLP | CI/CD | Decomposes series into trend and seasonal components, then applies linear layers for improved forecasting. |
| | Koopa(Liu et al., 2023) | Stat | DFT | | Point-wise, Patch-wise | MLP | CD | Uses Koopman theory to model non-stationary dynamics, handling time-variant and time-invariant components. |
| | TSMixer(Chen et al., 2023) | | | | Point-wise | MLP | CD | Simple MLP model efficiently captures both time and feature dependencies for forecasting. |
| | FreTS(Yi et al., 2023) | | | | Point-wise | Frequency-domain MLP | CI/CD | Uses frequency-domain MLPs to capture global dependencies and focus on key frequency components. |
| | TiDE(Das et al., 2023) | Stat | | | Point-wise | MLP | CI | Fast MLP-based model for long-term forecasting, handling covariates and non-linear dependencies. |
| | TimeMixer(Wang et al., 2024a) | RevIN | MA | TRUE | Point-wise | MLP | CI/CD | Fully MLP-based model, disentangles and mixes multi-scale temporal patterns. |
| Transformer | Reformer(Kitaev et al., 2020) | | | | Point-wise | LSHSelf-Attention | CD | Memory-efficient Transformer with locality-sensitive hashing for faster training on long sequences. |
| | Informer(Zhou et al., 2021) | | | | Point-wise | ProbSparse-Attention | CD | Efficient Transformer with ProbSparse-Attention and a generative decoder for faster long-sequence forecasting. |
| | TFT(Lim et al., 2021) | Stat | | | Point-wise | Self-Attention | CD | High-performance, interpretable multi-horizon forecasting model combining recurrent layers for local processing and attention layers for long-term dependencies. |
| | Autoformer(Wu et al., 2021) | | MA | | Point-wise | Auto-Correlation | CD | Uses Auto-Correlation and decomposition for accurate long-term predictions. |
| | PyraFormer(Liu et al., 2022b) | | | TRUE | Point-wise | Pyramid-Attention | CD | Captures temporal dependencies at multiple resolutions with constant signal path length. |
| | NSTransformer(Liu et al., 2022d) | Stat | | | Point-wise | De-stationary Attention | CD | Restores non-stationary information through de-stationary attention for improved forecasting. |
| | ETSformer(Woo et al., 2022) | | DFT | | Point-wise | Exponential-Smoothing-Attention | CD | Integrates exponential smoothing and frequency attention for accuracy, efficiency, and interpretability. |
| | FEDformer(Zhou et al., 2022b) | | MA | TRUE | Point-wise | AutoCorrelation | CD | Combines seasonal-trend decomposition with frequency-enhanced Transformer for efficient forecasting. |
| | Crossformer(Zhang and Yan, 2023) | | | TRUE | Patch-wise | TwoStage-Attention | CD | Captures both temporal and cross-variable dependencies with two-stage attention. |
| | PatchTST(Nie et al., 2023) | Stat | | | Patch-wise | FullAttention | CI | Segments time series into patches and uses channel-independent embeddings. |
| | iTransformer(Liu et al., 2024a) | Stat | | | Series-wise | FullAttention | CD | Redefines token embedding to treat time points as series-wise tokens for better multivariate modeling. |
| | TimeXer(Wang et al., 2024c) | Stat | | | Series-wise | FullAttention | CD | Enhances forecasting by incorporating exogenous variables via patch-wise and variate-wise attention. |
| | PAttn(Tan et al., 2024) | Stat | | | Patch-wise | FullAttention | CI | Similar to PatchTST, uses attention-based patching for efficient forecasting without large language models. |
| | DUET(Qiu et al., 2025b) | RevIN | MA | | Point-wise | FullAttention | CI/CD | Enhances multivariate forecasting by using Mixture of Experts (MOE) for temporal clustering and a frequency-domain similarity mask matrix for channel clustering. |

Table F3: Compared with ADGym, TSGym covers a broader and more in-depth design space, as well as a more structured and extensive automated selection experiment.

| | ADGym | TSGym |
|---|---|---|
| Design Dimensions | 13 | 16 |
| Design Space Size | 195,9552 | 796,2624 |
| Model Architectures | MLP,AE,ResNet,FTTransformer | MLP,RNN, Transformers, LLM, TSFM |
| Max of Data Samples | 3000 | 57,600 |
| Baseline Methods | 7 | 27 |

whereas TSGym encompasses series sampling, series normalization, series decomposition, as well as various series encoding options. **(3) More complex meta-features**. The meta-features in ADGym include statistical metrics for tabular datasets, while TSGym considers multiple sequence characteristics across different channels in multivariate time series, such as distribution drift, sequence autocorrelation, and more. **(4) More standardized automated selection experiments**. Due to time constraints, ADGym limits the sample size to fewer than 3000 samples, whereas TSGym imposes no such restriction, providing a larger-scale experimental design that leads to more solid experimental conclusions.

In summary, compared with ADGym, TSGym makes **significant progress and development in both components benchmarking and automated selection.** More details can be seen in table F3.

## G    META-FEATURES AND META-PREDICTORS

**Details and the selected list of meta-features**. The meta-features in this paper are extracted via TSFEL (Barandas et al., 2020) spanning temporal, statistical, spectral, and fractal domains. In Section 4.2, we present the results of the meta-predictor trained on meta-features derived from these static characteristics. Furthermore, in Fig. G1, we visualize the dimension-reduced meta-features across different datasets. The following categorizes these features with their analytical purposes (see Tables G4–G7 for implementation details):

- **Temporal features** (Table G4): Characterize sequential dynamics through trend detection, entropy analysis, and change-point statistics, preserving sensitivity to temporal ordering.

- **Statistical features** (Table G5): Capture distribution properties via central tendency (mean/median), dispersion (variance/IQR), and shape descriptors (skewness/kurtosis), invariant to observation order.

- **Spectral features** (Table G6): Decompose signals into frequency components using Fourier/wavelet transforms, identifying dominant periodicities and hidden oscillations.

- **Fractal features** (Table G7): Quantify multiscale complexity through fractal dimensions and Hurst exponents, reflecting self-similarity patterns across temporal resolutions.

### Table G4: **Temporal** Meta-feature Specifications

| Feature | Description | Functionality |
|---|---|---|
| Absolute Energy | Computes the absolute energy of the signal. | Measures the total energy of the signal, often used to understand signal power and activity levels. |
| Area Under the Curve | Computes the area under the curve of the signal computed with the trapezoid rule. | Provides a measure of the overall signal amplitude or ""energy"" over time. |
| Autocorrelation | Calculates the first 1/e crossing of the autocorrelation function (ACF). | Measures the correlation of the signal with its own past values, useful for identifying repeating patterns. |
| Average Power | Computes the average power of the signal. | Averages the squared values of the signal, capturing its power over time. |
| Centroid | Computes the centroid along the time axis. | Indicates the ""center"" or ""balance point"" of the signal in time, providing insight into its distribution. |
| Signal Distance | Computes signal traveled distance. | Measures the total path length covered by the signal over time, capturing the extent of signal fluctuations. |
| Negative Turning | Computes number of negative turning points of the signal. | Counts the number of times the signal changes direction from positive to negative. |
| Neighbourhood Peaks | Computes the number of peaks from a defined neighbourhood of the signal. | Identifies the number of peak points within a specified window, useful for pattern detection. |
| Peak-to-Peak Distance | Computes the peak to peak distance. | Measures the time interval between successive peaks, indicating the period of oscillations. |
| Positive Turning | Computes number of positive turning points of the signal. | Counts the number of times the signal changes direction from negative to positive. |
| Root Mean Square | Computes root mean square of the signal. | Calculates the square root of the average squared values of the signal, often used as a measure of signal strength. |
| Slope | Computes the slope of the signal. | Measures the rate of change in the signal's amplitude over time, indicating trends or shifts. |
| Sum of Absolute Differences | Computes sum of absolute differences of the signal. | Measures the total variation in the signal by summing the absolute differences between consecutive values. |
| Zero-Crossing Rate | Computes Zero-crossing rate of the signal. | Counts how many times the signal crosses the zero axis, indicating its frequency and periodicity. |

**Details of the trained meta-predictors**. For each design choice, we first use the LabelEncoder class from scikit-learn to convert it into a numerical class index. This index is then fed into an $nn.Embedding$ layer within our model to obtain a dense vector representation. These learned embeddings, along with other meta-features, subsequently form the input to the meta-predictor. The meta-predictor is optimized using Pearson loss to learn the relative performance ranks of different design choices, thereby emphasizing the linear correlation between predicted and actual rankings.

Moreover, we experimented with different training strategies to guide the meta-predictor in selecting the top-1 design pipelines. We report the results of TSGym with different training strategies in Table 4.

(1) **+Resample**: Constraining the number of combinations from different datasets to be equal when training the meta-predictor.

(2) **+AllPL**: Training on datasets with varying prediction lengths and transfers this knowledge to a test set with a single prediction length.

Table G5: **Statistical** Meta-feature Specifications

| Feature | Description | Functionality |
|---|---|---|
| Maximum Value | Computes the maximum value of the signal. | Identifies the highest amplitude or peak value in the signal, useful for determining extreme values. |
| Mean Value | Computes mean value of the signal. | Calculates the average value of the signal, providing insight into its central tendency. |
| Median | Computes the median of the signal. | Finds the middle value of the signal when sorted, offering robustness to outliers. |
| Minimum Value | Computes the minimum value of the signal. | Identifies the lowest amplitude or trough value in the signal, useful for detecting minima. |
| Standard Deviation | Computes standard deviation (std) of the signal. | Measures the variation or spread of the signal values, indicating how much the signal deviates from the mean. |
| Variance | Computes variance of the signal. | Quantifies the spread of signal values, related to the square of the standard deviation. |
| Empirical Cumulative Distribution Function | Computes the values of ECDF along the time axis. | Provides a cumulative distribution function, representing the probability distribution of the signal values. |
| ECDF Percentile | Computes the percentile value of the ECDF. | Extracts specific percentiles from the cumulative distribution, useful for understanding the signal's quantiles. |
| ECDF Percentile Count | Computes the cumulative sum of samples that are less than the percentile. | Measures the number of samples falling below a given percentile, providing distribution insights. |
| ECDF Slope | Computes the slope of the ECDF between two percentiles. | Measures the steepness or rate of change in the cumulative distribution, indicating distribution sharpness. |
| Histogram Mode | Compute the mode of a histogram using a given number of bins. | Finds the most frequent value in the signal's histogram, representing the peak of the signal's distribution. |
| Interquartile Range | Computes interquartile range of the signal. | Measures the range between the 25th and 75th percentiles, indicating the spread of the central 50% of the signal values. |
| Kurtosis | Computes kurtosis of the signal. | Measures the ""tailedness"" of the signal distribution, indicating the presence of outliers or extreme values. |
| Mean Absolute Deviation | Computes mean absolute deviation of the signal. | Measures the average deviation of the signal values from the mean, providing an indication of signal variability. |
| Mean Absolute Difference | Computes mean absolute differences of the signal. | Calculates the average of absolute differences between successive signal values, reflecting the signal's smoothness. |
| Mean Difference | Computes mean of differences of the signal. | Computes the average of the first-order differences, used to measure overall signal change. |
| Median Absolute Deviation | Computes median absolute deviation of the signal. | Measures the spread of the signal values around the median, offering a robust measure of variability. |
| Median Absolute Difference | Computes median absolute differences of the signal. | Similar to mean absolute difference but based on the median, used to assess signal smoothness. |
| Median Difference | Computes median of differences of the signal. | Calculates the median of first-order differences, providing insights into signal trend stability. |
| Skewness | Computes skewness of the signal. | Measures the asymmetry of the signal's distribution, indicating whether it is skewed towards higher or lower values. |

Table G6: **Spectral** Meta-feature Specifications

| Feature | Description | Functionality |
|---|---|---|
| Entropy | Computes the entropy of the signal using Shannon Entropy. | Quantifies the uncertainty or randomness in the signal, offering insights into its complexity. |
| Fundamental Frequency | Computes the fundamental frequency of the signal. | Identifies the primary frequency at which the signal oscillates, crucial for detecting periodic behaviors. |
| Human Range Energy | Computes the human range energy ratio. | Measures the energy in the human audible range, useful for identifying signals relevant to human hearing. |
| Linear Prediction Cepstral Coefficients | Computes the linear prediction cepstral coefficients. | Extracts features related to the signal's frequency components, commonly used in speech and audio processing. |
| Maximum Frequency | Computes maximum frequency of the signal. | Identifies the highest frequency component of the signal, providing insight into its frequency range. |
| Maximum Power Spectrum | Computes maximum power spectrum density of the signal. | Measures the peak value in the power spectral density, identifying dominant frequencies in the signal. |
| Median Frequency | Computes median frequency of the signal. | Identifies the frequency that divides the signal's power spectrum into two equal halves. |
| Mel-Frequency Cepstral Coefficients | Computes the MEL cepstral coefficients. | Used to extract features representing the spectral characteristics of the signal, primarily used in speech analysis. |
| Multiscale Entropy | Computes the Multiscale entropy (MSE) of the signal, that performs entropy analysis over multiple scales. | Quantifies the signal's complexity at different scales, useful for detecting non-linear temporal behaviors. |
| Power Bandwidth | Computes power spectrum density bandwidth of the signal. | Measures the width of the frequency band where the majority of the signal's power is concentrated. |
| Spectral Centroid | Barycenter of the spectrum. | Identifies the ""center"" of the signal's frequency spectrum, used in sound and audio analysis. |
| Spectral Decrease | Represents the amount of decreasing of the spectra amplitude. | Measures how rapidly the spectral amplitude decreases across frequency, useful for identifying spectral roll-off. |
| Spectral Distance | Computes the signal spectral distance. | Quantifies the difference between the signal's spectrum and a reference, helpful in pattern recognition. |
| Spectral Entropy | Computes the spectral entropy of the signal based on Fourier transform. | Measures the randomness or complexity in the frequency domain of the signal. |
| Spectral Kurtosis | Measures the flatness of a distribution around its mean value. | Quantifies the tail heaviness of the signal's frequency distribution, identifying outliers or abnormal distributions. |
| Spectral Positive Turning | Computes number of positive turning points of the fft magnitude signal. | Counts the points where the signal's Fourier transform changes direction from negative to positive. |
| Spectral Roll-Off | Computes the spectral roll-off of the signal. | Measures the frequency below which a specified percentage of the total spectral energy is contained. |
| Spectral Roll-On | Computes the spectral roll-on of the signal. | Similar to roll-off but identifies the frequency above which a specified amount of energy is concentrated. |
| Spectral Skewness | Measures the asymmetry of a distribution around its mean value. | Measures the skew in the signal's frequency distribution, highlighting the presence of spectral biases. |
| Spectral Slope | Computes the spectral slope. | Quantifies the slope of the power spectral density, often used to distinguish between harmonic and non-harmonic signals. |
| Spectral Spread | Measures the spread of the spectrum around its mean value. | Measures the dispersion or spread of the signal's spectral energy. |
| Spectral Variation | Computes the amount of variation of the spectrum along time. | Quantifies how much the frequency content of the signal changes over time. |
| Spectrogram Mean Coefficients | Calculates the average power spectral density (PSD) for each frequency throughout the entire signal. | Averages the power spectral density across all time intervals, capturing the signal's overall spectral energy distribution. |
| Wavelet Absolute Mean | Computes CWT absolute mean value of each wavelet scale. | Measures the average wavelet transform magnitude across scales, useful for detecting changes in signal frequency. |
| Wavelet Energy | Computes CWT energy of each wavelet scale. | Quantifies the energy at each wavelet scale, reflecting the signal's energy distribution across frequencies. |
| Wavelet Entropy | Computes CWT entropy of the signal. | Measures the complexity or unpredictability of the signal at different wavelet scales. |
| Wavelet Standard Deviation | Computes CWT std value of each wavelet scale. | Measures the variation or spread of the wavelet transform across different scales. |
| Wavelet Variance | Computes CWT variance value of each wavelet scale. | Quantifies the dispersion of the signal at different wavelet scales. |

Table G7: **Fractal** Meta-feature Specifications

| Feature | Description | Functionality |
|---|---|---|
| Detrended Fluctuation Analysis | Computes the Detrended Fluctuation Analysis (DFA) of the signal. | Measures long-range correlations and self-similarity in the signal, used for identifying fractal behavior. |
| Higuchi Fractal Dimension | Computes the fractal dimension of a signal using Higuchi's method (HFD). | Measures the complexity of the signal's pattern by calculating its fractal dimension. |
| Hurst Exponent | Computes the Hurst exponent of the signal through the Rescaled range (R/S) analysis. | Measures the long-term memory or persistence in the signal, useful for identifying trends and randomness. |
| Lempel-Ziv Complexity | Computes the Lempel-Ziv's (LZ) complexity index, normalized by the signal's length. | Quantifies the randomness or predictability of the signal based on its compressibility. |
| Maximum Fractal Length | Computes the Maximum Fractal Length (MFL) of the signal. | Measures the fractal dimension at the smallest scale of the signal, reflecting its intricate pattern complexity. |
| Petrosian Fractal Dimension | Computes the Petrosian Fractal Dimension of a signal. | Measures the signal's fractal dimension based on its variation across different scales. |

Table H8: Full results for the long-term forecasting task. All the results are averaged from 4 different prediction lengths, that is $\{24, 36, 48, 60\}$ for ILI and $\{96, 192, 336, 720\}$ for the others.

| Models | TSGym (Ours) | | DUET (Qiu et al., 2025b) | | TimeMixer (Wang et al., 2024a) | | MICN (Wang et al., 2023) | | TimesNet (Wu et al., 2023) | | PatchTST (Nie et al., 2023) | | DLinear (Zeng et al., 2023b) | | Crossformer (Zhang and Yan, 2023) | | Autoformer (Wu et al., 2021) | | SegRNN (Lin et al., 2023) | | Mamba (Gu and Dao, 2024) | | iTransformer (Liu et al., 2024a) | | TimeXer (Wang et al., 2024c) | |
|---|---|---|---|---|---|---|---|---|---|---|---|---|---|---|---|---|---|---|---|---|---|---|---|---|---|---|
| Metric | MSE | MAE | MSE | MAE | MSE | MAE | MSE | MAE | MSE | MAE | MSE | MAE | MSE | MAE | MSE | MAE | MSE | MAE | MSE | MAE | MSE | MAE | MSE | MAE | MSE | MAE |
| ETTm1 | 0.362 | 0.380 | 0.407 | 0.409 | 0.384 | 0.399 | 0.402 | 0.429 | 0.432 | 0.430 | 0.390 | 0.404 | 0.404 | 0.407 | 0.501 | 0.532 | 0.496 | 0.501 | 0.388 | 0.404 | 0.501 | 0.466 | 0.414 | 0.415 | 0.386 | 0.400 |
| ETTm2 | 0.266 | 0.322 | 0.296 | 0.338 | 0.277 | 0.325 | 0.342 | 0.391 | 0.296 | 0.334 | 0.288 | 0.334 | 0.349 | 0.399 | 1.487 | 0.789 | 0.330 | 0.368 | 0.273 | 0.322 | 0.356 | 0.370 | 0.290 | 0.332 | 0.279 | 0.325 |
| ETTh1 | 0.427 | 0.439 | 0.433 | 0.437 | 0.448 | 0.438 | 0.589 | 0.537 | 0.474 | 0.464 | 0.449 | 0.465 | 0.461 | 0.544 | 0.520 | 0.492 | 0.485 | 0.422 | 0.429 | 0.544 | 0.504 | 0.462 | 0.452 | 0.446 | 0.443 | |
| ETTh2 | 0.367 | 0.403 | 0.380 | 0.403 | 0.383 | 0.406 | 0.585 | 0.530 | 0.415 | 0.424 | 0.385 | 0.409 | 0.566 | 0.520 | 0.908 | 0.446 | 0.460 | 0.465 | 0.448 | 0.382 | 0.406 | 0.372 | 0.399 | | | |
| ECL | 0.164 | 0.261 | 0.179 | 0.262 | 0.185 | 0.273 | 0.186 | 0.297 | 0.219 | 0.314 | 0.209 | 0.298 | 0.225 | 0.319 | 0.193 | 0.289 | 0.234 | 0.340 | 0.216 | 0.302 | 0.209 | 0.312 | 0.190 | 0.277 | 0.191 | 0.286 |
| Traffic | 0.433 | 0.301 | 0.797 | 0.427 | 0.496 | 0.313 | 0.544 | 0.320 | 0.645 | 0.348 | 0.497 | 0.321 | 0.673 | 0.419 | 1.458 | 0.782 | 0.637 | 0.397 | 0.807 | 0.411 | 0.679 | 0.380 | 0.474 | 0.318 | 0.509 | 0.333 |
| Weather | 0.240 | 0.276 | 0.252 | 0.277 | 0.244 | 0.274 | 0.264 | 0.316 | 0.261 | 0.287 | 0.256 | 0.279 | 0.265 | 0.317 | 0.253 | 0.312 | 0.339 | 0.379 | 0.251 | 0.298 | 0.291 | 0.315 | 0.259 | 0.280 | 0.243 | 0.273 |
| Exchange | 0.375 | 0.415 | 0.322 | 0.384 | 0.359 | 0.402 | 0.346 | 0.422 | 0.405 | 0.437 | 0.381 | 0.412 | 0.346 | 0.414 | 0.904 | 0.695 | 0.506 | 0.500 | 0.408 | 0.423 | 0.714 | 0.562 | 0.369 | 0.410 | 0.410 | 0.424 |
| ILI | 2.463 | 1.043 | 2.640 | 1.018 | 4.502 | 1.557 | 2.938 | 1.178 | 2.140 | 0.907 | 2.160 | 0.901 | 4.367 | 1.540 | 4.311 | 1.396 | 3.156 | 1.207 | 4.305 | 1.397 | 3.729 | 1.335 | 2.305 | 0.974 | 2.633 | 1.034 |
| 1st Count | 8 | | 2 | | 2 | | 0 | | 0 | | 1 | | 0 | | 0 | | 0 | | 2 | | 0 | | 0 | | 1 | |

| Models | PAttn (Tan et al., 2024) | | Koopa (Liu et al., 2023) | | TSMixer (Chen et al., 2023) | | FreTS (Yi et al., 2023) | | Pyraformer (Liu et al., 2022b) | | Nonstationary (Liu et al., 2022c) | | ETSformer (Woo et al., 2022) | | FEDformer (Zhou et al., 2022b) | | SCINet (Liu et al., 2022a) | | LightTS (Zhang et al., 2022) | | Informer (Zhou et al., 2021) | | Transformer (Vaswani et al., 2017) | | Reformer (Kitaev et al., 2020) | |
|---|---|---|---|---|---|---|---|---|---|---|---|---|---|---|---|---|---|---|---|---|---|---|---|---|---|---|
| Metric | MSE | MAE | MSE | MAE | MSE | MAE | MSE | MAE | MSE | MAE | MSE | MAE | MSE | MAE | MSE | MAE | MSE | MAE | MSE | MAE | MSE | MAE | MSE | MAE | MSE | MAE |
| ETTm1 | 0.384 | 0.399 | 0.367 | 0.396 | 0.527 | 0.512 | 0.409 | 0.417 | 0.695 | 0.593 | 0.509 | 0.467 | 0.636 | 0.592 | 0.438 | 0.450 | 0.409 | 0.412 | 0.438 | 0.445 | 0.969 | 0.736 | 0.836 | 0.678 | 0.998 | 0.723 |
| ETTm2 | 0.291 | 0.336 | 0.264 | 0.327 | 1.030 | 0.750 | 0.336 | 0.378 | 1.565 | 0.876 | 0.412 | 0.398 | 1.381 | 0.807 | 0.301 | 0.348 | 0.294 | 0.335 | 0.432 | 0.448 | 1.504 | 0.878 | 1.454 | 0.851 | 1.856 | 0.996 |
| ETTh1 | 0.468 | 0.454 | 0.472 | 0.471 | 0.615 | 0.579 | 0.476 | 0.464 | 0.814 | 0.692 | 0.610 | 0.543 | 0.750 | 0.651 | 0.448 | 0.461 | 0.520 | 0.488 | 0.530 | 0.505 | 1.057 | 0.798 | 0.930 | 0.768 | 0.973 | 0.739 |
| ETTh2 | 0.386 | 0.412 | 0.388 | 0.423 | 2.160 | 1.220 | 0.548 | 0.514 | 3.776 | 1.557 | 0.552 | 0.505 | 0.572 | 0.534 | 0.427 | 0.446 | 0.428 | 0.440 | 0.633 | 0.551 | 4.535 | 1.745 | 2.976 | 1.369 | 2.487 | 1.238 |
| ECL | 0.205 | 0.286 | 0.219 | 0.319 | 0.229 | 0.337 | 0.209 | 0.296 | 0.295 | 0.387 | 0.194 | 0.296 | 0.275 | 0.370 | 0.225 | 0.336 | 0.220 | 0.323 | 0.243 | 0.344 | 0.369 | 0.444 | 0.273 | 0.367 | 0.324 | 0.404 |
| Traffic | 0.513 | 0.328 | 0.595 | 0.413 | 0.599 | 0.403 | 0.597 | 0.377 | 0.697 | 0.391 | 0.642 | 0.351 | 1.035 | 0.584 | 0.615 | 0.379 | 0.654 | 0.419 | 0.656 | 0.428 | 0.830 | 0.464 | 0.708 | 0.384 | 0.694 | 0.380 |
| Weather | 0.257 | 0.280 | 0.230 | 0.271 | 0.242 | 0.301 | 0.255 | 0.299 | 0.284 | 0.349 | 0.289 | 0.312 | 0.365 | 0.424 | 0.315 | 0.369 | 0.256 | 0.283 | 0.245 | 0.295 | 0.572 | 0.523 | 0.599 | 0.531 | 0.475 | 0.472 |
| Exchange | 0.365 | 0.407 | 0.610 | 0.516 | 0.487 | 0.546 | 0.442 | 0.453 | 1.183 | 0.855 | 0.557 | 0.490 | 0.361 | 0.416 | 0.520 | 0.502 | 0.374 | 0.418 | 0.486 | 0.493 | 1.548 | 0.997 | 1.379 | 0.921 | 1.612 | 1.044 |
| ILI | 2.359 | 0.975 | 2.064 | 0.912 | 5.617 | 1.680 | 3.447 | 1.279 | 4.691 | 1.442 | 2.592 | 1.012 | 4.046 | 1.419 | 3.088 | 1.214 | 6.505 | 1.853 | 7.078 | 1.975 | 5.035 | 1.539 | 4.682 | 1.448 | 4.211 | 1.350 |
| 1st Count | 0 | | 4 | | 0 | | 0 | | 0 | | 0 | | 0 | | 0 | | 0 | | 0 | | 0 | | 0 | | 0 | |

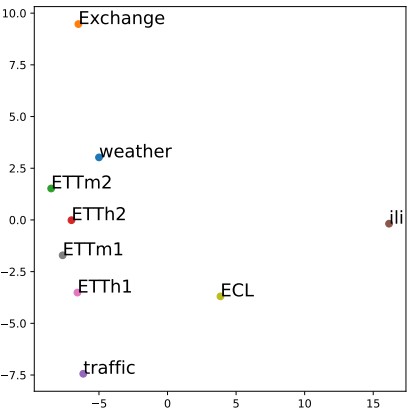

(a) PCA projection of meta-features for 9 long-term forecasting datasets

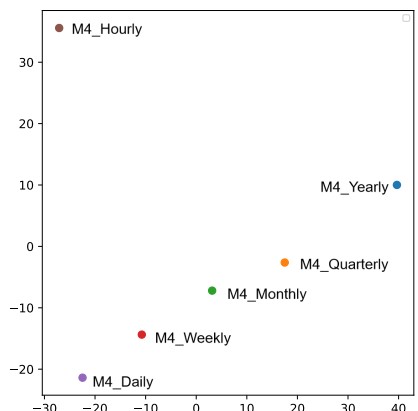

(b) PCA projection of meta-features for 6 short-term forecasting datasets

Figure G1: Distributions of meta-features after PCA dimensionality reduction, comparing datasets for long-term and short-term time series forecasting tasks.

# H  ADDITIONAL EXPERIMENTAL RESULTS

## H.1  COMPREHENSIVE RESULTS OF TSGYM AGAINST STATE-OF-THE-ART METHODS

Due to space limitations in the main text, here we provide complete experimental comparisons for both long-term and short-term forecasting tasks. Table H8 details the full long-term forecasting performance across all prediction horizons, while Table H9 presents the comprehensive short-term forecasting results. Following standard benchmarking conventions, we highlight top-performing methods in **red** and second-best results with underlined formatting. These extensive evaluations consistently validate TSGym's competitive performance across diverse temporal prediction scenarios.

## H.2  ADDITIONAL RESULTS OF LARGE EVALUATIONS ON DESIGN CHOICES

To systematically evaluate our architectural decisions, we conduct detailed ablation studies focusing on 17 component-level analyses, presented separately in Tables H10–H13 for clarity and due to space constraints. These comparative experiments assess the performance impact of different design choices for each component across nine datasets in the long-term forecasting task. **Bolded** values indicate the best-performing configuration for each dataset, while the summary row highlights the most frequently superior design choices, with **red-bolded** entries denoting the dominant configurations. This fine-grained analysis offers empirical insights to guide component selection in time-series forecasting systems.

Table H9: Full results for the short-term forecasting task in the M4 dataset. *. in the Transformers indicates the name of *former.

| | Models | TSGym (Ours) | TimeMixer | MICN | TimesNet | PatchTST | DLinear | Cross. | Auto. | SegRNN | Mamba | iTrans. | TimeXer | PAttn | TSMixer | FreTS | Pyra. | ETS | FED. | SCINet | LightTS | In. | Trans. | Re. | TiDE | FiLM |
|---|---|---|---|---|---|---|---|---|---|---|---|---|---|---|---|---|---|---|---|---|---|---|---|---|---|---|
| Yearly | OWA | 0.779 | 0.789 | 0.873 | 0.784 | 0.798 | 0.843 | 4.407 | 1.019 | 0.858 | 0.790 | 0.807 | 0.797 | 0.823 | 0.798 | 0.800 | 0.883 | 0.982 | 0.806 | 0.801 | 0.795 | 0.887 | 4.406 | 0.829 | 0.807 | 0.806 |
| | sMAPE | 13.336 | 13.467 | 14.586 | 13.344 | 13.606 | 14.402 | 71.464 | 17.294 | 14.323 | 13.388 | 13.681 | 13.551 | 13.893 | 13.539 | 13.579 | 14.967 | 16.105 | 13.631 | 13.573 | 13.516 | 15.086 | 69.405 | 14.064 | 14.006 | 13.988 |
| | MASE | 2.966 | 3.060 | 3.392 | 2.985 | 3.013 | 3.198 | 17.649 | 3.897 | 3.335 | 3.021 | 3.000 | 3.043 | 3.162 | 3.051 | 3.056 | 3.377 | 3.888 | 3.088 | 3.065 | 3.030 | 3.382 | 18.144 | 3.166 | 3.011 | 3.007 |
| Quarterly | OWA | 0.891 | 0.911 | 1.025 | 0.886 | 0.975 | 0.928 | 8.208 | 1.290 | 1.009 | 0.912 | 1.050 | 0.942 | 0.897 | 0.903 | 0.908 | 1.002 | 1.312 | 0.940 | 0.922 | 0.886 | 1.248 | 8.190 | 1.007 | 0.939 | 0.960 |
| | sMAPE | 10.232 | 10.313 | 11.427 | 10.063 | 10.975 | 10.500 | 74.297 | 14.085 | 11.193 | 10.320 | 11.752 | 10.556 | 10.203 | 10.218 | 10.339 | 11.259 | 13.600 | 10.655 | 10.426 | 10.166 | 13.644 | 73.944 | 11.334 | 10.719 | 10.742 |
| | MASE | 1.169 | 1.215 | 1.388 | 1.176 | 1.306 | 1.238 | 13.260 | 1.784 | 1.374 | 1.217 | 1.416 | 1.272 | 1.189 | 1.204 | 1.205 | 1.345 | 1.906 | 1.252 | 1.229 | 1.163 | 1.723 | 13.256 | 1.352 | 1.295 | 1.296 |
| Monthly | OWA | 0.863 | 0.895 | 0.986 | 0.939 | 1.020 | 0.936 | 7.637 | 1.369 | 1.074 | 0.931 | 0.982 | 0.944 | 0.951 | 0.898 | 0.915 | 1.043 | 1.281 | 1.001 | 0.924 | 0.884 | 1.151 | 7.668 | 1.448 | 0.942 | 0.942 |
| | sMAPE | 12.570 | 12.823 | 13.798 | 13.314 | 14.156 | 13.384 | 68.873 | 18.132 | 15.052 | 13.152 | 13.737 | 13.254 | 13.421 | 12.865 | 13.059 | 14.666 | 15.449 | 14.112 | 13.146 | 12.717 | 15.806 | 69.992 | 18.782 | 13.381 | 13.352 |
| | MASE | 0.909 | 0.959 | 1.080 | 1.015 | 1.126 | 1.004 | 11.165 | 1.576 | 1.175 | 1.000 | 1.076 | 1.000 | 1.034 | 0.962 | 0.984 | 1.138 | 1.586 | 1.090 | 0.996 | 0.943 | 1.283 | 11.149 | 1.694 | 1.017 | 1.019 |
| Weekly | OWA | 0.983 | 1.266 | 1.500 | 1.310 | 1.035 | 1.461 | 28.636 | 1.575 | 0.998 | 1.449 | 1.424 | 1.179 | 1.124 | 1.525 | 1.286 | 1.313 | 1.024 | 1.094 | 1.438 | 1.340 | 1.537 | 28.093 | 1.473 | 1.451 | 1.280 |
| | sMAPE | 9.467 | 11.555 | 11.790 | 11.569 | 9.546 | 11.805 | 198.371 | 12.727 | 9.149 | 12.495 | 12.050 | 10.455 | 10.326 | 12.364 | 11.394 | 11.742 | 9.363 | 9.635 | 12.757 | 12.060 | 11.967 | 191.424 | 11.522 | 12.425 | 11.539 |
| | MASE | 2.893 | 3.529 | 4.760 | 3.770 | 2.857 | 4.534 | 98.925 | 4.892 | 2.771 | 4.262 | 4.254 | 3.379 | 3.111 | 4.723 | 3.688 | 3.732 | 2.848 | 3.155 | 4.122 | 3.789 | 4.911 | 98.015 | 4.690 | 4.295 | 3.609 |
| Daily | OWA | 0.982 | 1.040 | 1.245 | 1.079 | 1.018 | 1.090 | 48.627 | 1.418 | 0.988 | 1.130 | 1.191 | 1.047 | 1.011 | 1.250 | 1.088 | 1.111 | 1.061 | 0.998 | 1.082 | 1.086 | 1.235 | 29.620 | 1.496 | 1.106 | 1.082 |
| | sMAPE | 3.095 | 3.162 | 3.786 | 3.294 | 3.103 | 3.319 | 179.226 | 4.248 | 3.009 | 3.417 | 3.607 | 3.198 | 3.089 | 3.677 | 3.304 | 3.398 | 3.216 | 3.060 | 3.288 | 3.313 | 3.727 | 99.709 | 4.521 | 3.342 | 3.307 |
| | MASE | 3.205 | 3.418 | 4.089 | 3.550 | 3.332 | 3.577 | 125.892 | 4.722 | 3.237 | 3.732 | 3.928 | 3.423 | 3.303 | 4.109 | 3.580 | 3.626 | 3.497 | 3.247 | 3.552 | 3.557 | 4.086 | 86.873 | 4.941 | 3.653 | 3.580 |
| Hourly | OWA | 0.902 | 1.372 | 1.526 | 3.171 | 2.625 | 1.040 | 11.691 | 1.623 | 1.704 | 0.750 | 1.214 | 1.656 | 1.683 | 1.243 | 1.393 | 2.315 | 1.201 | 1.126 | 1.372 | 1.183 | 3.166 | 6.498 | 3.204 | 1.231 | 1.445 |
| | sMAPE | 18.203 | 19.994 | 24.631 | 34.626 | 29.980 | 17.260 | 128.419 | 25.407 | 34.523 | 14.944 | 19.573 | 21.244 | 23.431 | 19.809 | 20.805 | 26.112 | 19.751 | 18.858 | 24.196 | 21.382 | 34.038 | 99.324 | 34.755 | 20.828 | 21.088 |
| | MASE | 1.947 | 3.966 | 4.100 | 10.680 | 8.667 | 2.732 | 39.269 | 4.465 | 3.663 | 1.549 | 3.266 | 5.067 | 5.009 | 3.372 | 3.964 | 7.685 | 3.178 | 2.937 | 3.420 | 2.881 | 10.730 | 18.188 | 10.821 | 3.183 | 4.172 |
| Average | OWA | 0.856 | 0.884 | 0.984 | 0.907 | 0.965 | 0.922 | 8.856 | 1.273 | 1.007 | 0.903 | 0.969 | 0.918 | 0.915 | 0.894 | 0.897 | 1.005 | 1.209 | 0.942 | 0.906 | 0.875 | 1.127 | 8.039 | 1.209 | 0.925 | 0.924 |
| | sMAPE | 11.781 | 11.985 | 13.025 | 12.199 | 12.848 | 12.511 | 76.147 | 16.392 | 13.599 | 12.120 | 12.838 | 12.268 | 12.351 | 12.023 | 12.137 | 13.478 | 14.635 | 12.708 | 12.219 | 11.925 | 14.673 | 72.619 | 15.344 | 12.489 | 12.471 |
| | MASE | 1.581 | 1.615 | 1.859 | 1.662 | 1.738 | 1.693 | 18.440 | 2.317 | 1.823 | 1.651 | 1.762 | 1.677 | 1.680 | 1.657 | 1.645 | 1.844 | 2.284 | 1.695 | 1.657 | 1.605 | 2.042 | 16.805 | 2.136 | 1.674 | 1.673 |
| 1st Count | | 14 | 0 | 0 | 2 | 0 | 0 | 0 | 0 | 1 | 4 | 0 | 0 | 0 | 0 | 0 | 0 | 0 | 0 | 0 | 1 | 0 | 0 | 0 | 0 | 0 |

Table H10: Long-term Forecasting Performance of Different Design Choices – Part I (6 Components). **MSE** distribution for each dataset under different configurations of the 6 components, characterized by the best (minimum) value, median, and interquartile range (IQR). **Bolded** entries indicate the best-performing result for the respective dataset and metric in each component.

| dataset | stat | Timestamp Embedding | | Series Sampling/Mixing | | Series Normalization | | | | Series Decomposition | | | | Channel Independent | | Series Tokenization | | |
|---|---|---|---|---|---|---|---|---|---|---|---|---|---|---|---|---|---|---|
| | | w/o Embedding | w/ Embedding | w/o Mixing | w/ Mixing | DishTS | w/o Norm | RevIN | Stat | DFT | MA | MoEMA | w/o Decomp | Channel Depen | Channel Indepen | Inverted Encoding | Series Encoding | Series Patching |
| ETTm1 | Best | **0.348** | 0.348 | **0.348** | 0.350 | 0.360 | 0.354 | **0.348** | 0.351 | 0.350 | 0.352 | **0.348** | 0.351 | 0.352 | **0.348** | 0.354 | 0.349 | **0.348** |
| | Median | **0.416** | 0.459 | **0.427** | 0.459 | 0.511 | 0.577 | **0.403** | 0.404 | 0.459 | **0.405** | 0.445 | 0.466 | 0.472 | **0.389** | 0.400 | 0.487 | **0.390** |
| | IQR | **0.147** | 0.181 | **0.158** | 0.198 | 0.211 | 0.250 | **0.099** | 0.107 | 0.166 | **0.139** | 0.178 | 0.183 | 0.198 | **0.114** | 0.129 | 0.208 | **0.111** |
| ETTm2 | Best | 0.251 | **0.248** | **0.248** | 0.255 | 0.261 | 0.275 | **0.248** | 0.251 | 0.253 | **0.248** | 0.255 | 0.252 | 0.252 | **0.248** | 0.252 | 0.255 | **0.248** |
| | Median | 0.353 | **0.345** | **0.327** | 0.394 | 0.689 | 0.999 | **0.294** | 0.299 | 0.336 | 0.384 | 0.370 | **0.320** | 0.377 | **0.307** | **0.300** | 0.404 | 0.306 |
| | IQR | 0.561 | **0.537** | **0.374** | 0.903 | 0.707 | 1.252 | **0.033** | 0.035 | **0.366** | 0.742 | 0.628 | 0.426 | 0.823 | **0.130** | 0.195 | 0.922 | **0.160** |
| ETTh1 | Best | **0.401** | 0.406 | **0.403** | 0.404 | 0.433 | 0.419 | 0.402 | **0.401** | 0.407 | **0.405** | 0.405 | 0.409 | 0.412 | **0.401** | 0.412 | 0.412 | **0.401** |
| | Median | 0.490 | **0.488** | **0.480** | 0.519 | 0.547 | 0.633 | 0.464 | **0.462** | 0.492 | 0.489 | 0.491 | **0.487** | 0.510 | **0.461** | 0.474 | 0.522 | **0.456** |
| | IQR | **0.129** | 0.130 | **0.088** | 0.219 | 0.207 | 0.381 | 0.049 | **0.048** | 0.152 | **0.111** | 0.154 | 0.117 | 0.206 | **0.041** | 0.054 | 0.243 | **0.037** |
| ETTh2 | Best | **0.322** | 0.329 | 0.332 | **0.322** | 0.374 | 0.378 | **0.321** | 0.331 | 0.335 | **0.322** | 0.332 | 0.333 | **0.322** | 0.337 | 0.346 | **0.321** | 0.341 |
| | Median | **0.447** | 0.492 | **0.451** | 0.493 | 1.049 | 1.594 | **0.388** | 0.390 | **0.433** | 0.452 | 0.500 | 0.503 | 0.527 | **0.398** | 0.452 | 0.487 | **0.391** |
| | IQR | **0.775** | 0.859 | **0.600** | 1.574 | 1.222 | 2.615 | **0.043** | 0.048 | **0.576** | 0.941 | 0.995 | 0.735 | 1.369 | **0.218** | 0.390 | 1.671 | **0.252** |
| ECL | Best | 0.159 | **0.157** | **0.157** | 0.159 | 0.159 | 0.160 | 0.159 | **0.157** | 0.158 | 0.163 | 0.161 | **0.157** | **0.157** | 0.163 | **0.157** | 0.158 | 0.164 |
| | Median | 0.208 | **0.208** | **0.204** | 0.208 | 0.218 | 0.227 | **0.191** | 0.191 | 0.205 | 0.208 | 0.206 | **0.203** | 0.206 | **0.202** | 0.195 | 0.212 | **0.190** |
| | IQR | 0.057 | **0.056** | 0.058 | **0.054** | 0.052 | 0.058 | **0.035** | 0.052 | 0.064 | **0.053** | 0.054 | 0.056 | 0.056 | **0.055** | **0.050** | 0.061 | 0.050 |
| traffic | Best | **0.394** | 0.396 | 0.398 | **0.394** | 0.411 | 0.441 | 0.398 | **0.394** | 0.398 | 0.400 | **0.394** | 0.400 | **0.394** | 0.409 | 0.399 | **0.394** | 0.409 |
| | Median | **0.558** | 0.600 | 0.580 | **0.579** | 0.545 | 0.658 | 0.550 | **0.506** | 0.609 | 0.571 | **0.563** | 0.567 | **0.570** | 0.626 | **0.531** | 0.602 | 0.607 |
| | IQR | **0.191** | 0.198 | 0.208 | **0.179** | 0.161 | **0.122** | 0.209 | 0.196 | **0.179** | 0.190 | 0.202 | 0.198 | **0.195** | 0.196 | 0.191 | 0.188 | **0.186** |
| weather | Best | 0.222 | **0.220** | **0.220** | 0.222 | 0.223 | 0.225 | **0.220** | 0.224 | 0.225 | 0.223 | **0.220** | 0.221 | **0.220** | 0.220 | **0.220** | 0.220 | 0.222 |
| | Median | **0.258** | 0.272 | **0.258** | 0.272 | 0.263 | 0.292 | **0.256** | 0.259 | 0.262 | 0.271 | **0.260** | 0.261 | 0.272 | **0.246** | 0.248 | 0.280 | **0.242** |
| | IQR | **0.040** | 0.085 | **0.047** | 0.070 | 0.066 | 0.213 | **0.034** | 0.037 | **0.048** | 0.049 | 0.049 | 0.053 | 0.064 | **0.037** | **0.033** | 0.079 | 0.033 |
| Exchange | Best | 0.239 | **0.209** | **0.208** | 0.242 | **0.209** | 0.247 | 0.349 | 0.336 | 0.240 | 0.237 | 0.244 | **0.209** | **0.209** | 0.237 | 0.239 | **0.212** | 0.237 |
| | Median | **0.488** | 0.491 | **0.455** | 0.552 | 0.635 | 0.937 | 0.426 | **0.422** | **0.471** | 0.496 | 0.484 | 0.518 | 0.563 | **0.390** | 0.412 | 0.581 | **0.388** |
| | IQR | 0.461 | **0.457** | **0.414** | 0.545 | 0.796 | 0.902 | **0.167** | 0.168 | 0.427 | 0.481 | **0.416** | 0.516 | 0.590 | **0.131** | 0.250 | 0.612 | **0.114** |
| ili | Best | 1.584 | **1.546** | **1.562** | 1.576 | 1.755 | 2.137 | 1.584 | **1.555** | 1.649 | 1.599 | 1.581 | **1.573** | **1.545** | 1.734 | 1.583 | **1.548** | 1.734 |
| | Median | **2.837** | 2.875 | 2.884 | **2.814** | 2.802 | 4.373 | 2.505 | **2.501** | 2.892 | **2.803** | 2.892 | 2.853 | **2.804** | 3.048 | 2.870 | 2.860 | **2.830** |
| | IQR | **1.613** | 1.690 | **1.639** | 1.677 | 1.205 | 0.967 | **0.739** | 0.786 | 1.637 | **1.604** | 1.702 | 1.661 | 1.666 | **1.649** | 1.752 | 1.668 | **1.395** |

Table H11: Long-term Forecasting Performance of Different Design Choices– Part II (4 Components) and Part II (7 Components). Same structure and evaluation metrics (MSE) as Table H10.

(a) Part II – 4 Components (Backbone, Attention, etc.)

| dataset | stat | Network Backbone | | | Attention | | | | | | Feature-Attention | | | | Sequence Length | | | |
| | | GRU | MLP | Trans-former | AutoCorr | De-stationary Attention | Frequency Attention | w/o Attention | Self Attention | Sparse Attention | Frequency Attention | w/o Attention | Self Attention | Sparse Attention | 192 | 48 | 512 | 96 |
|---|---|---|---|---|---|---|---|---|---|---|---|---|---|---|---|---|---|---|
| ETTm1 | Best | 0.352 | **0.347** | 0.351 | 0.359 | 0.382 | 0.354 | **0.347** | 0.359 | 0.354 | 0.354 | **0.348** | 0.355 | 0.360 | 0.351 | 0.473 | **0.347** | 0.379 |
| | Median | 0.462 | **0.409** | 0.449 | 0.499 | 0.455 | **0.409** | 0.439 | 0.486 | 0.441 | 0.446 | **0.411** | 0.458 | 0.459 | **0.385** | 0.545 | 0.392 | 0.424 |
| | IQR | 0.160 | 0.172 | **0.151** | 0.242 | **0.087** | 0.106 | 0.168 | 0.189 | 0.143 | 0.179 | **0.154** | 0.177 | 0.201 | 0.101 | 0.093 | 0.109 | **0.091** |
| ETTm2 | Best | 0.260 | **0.248** | 0.256 | 0.258 | 0.289 | 0.265 | **0.248** | 0.260 | 0.267 | 0.258 | **0.248** | 0.253 | 0.254 | 0.261 | 0.293 | **0.248** | 0.273 |
| | Median | 0.352 | **0.323** | 0.416 | 0.663 | **0.320** | 0.335 | 0.336 | 0.437 | 0.766 | 0.426 | **0.323** | 0.383 | 0.372 | **0.329** | 0.385 | 0.335 | 0.378 |
| | IQR | 0.432 | **0.330** | 0.854 | 0.900 | **0.033** | 0.911 | 0.370 | 0.711 | 1.086 | 0.866 | **0.254** | 0.824 | 0.800 | 0.733 | **0.410** | 0.813 | 0.632 |
| ETTh1 | Best | 0.408 | **0.402** | 0.406 | 0.418 | 0.471 | 0.413 | **0.401** | 0.432 | 0.406 | 0.417 | **0.401** | 0.415 | 0.412 | 0.420 | 0.443 | **0.401** | 0.433 |
| | Median | 0.495 | **0.480** | 0.502 | 0.510 | 0.526 | **0.475** | 0.487 | 0.527 | 0.504 | 0.496 | **0.479** | 0.502 | 0.508 | 0.483 | 0.501 | **0.481** | 0.486 |
| | IQR | 0.125 | **0.107** | 0.167 | 0.190 | **0.060** | 0.070 | 0.116 | 0.207 | 0.214 | 0.163 | **0.080** | 0.156 | 0.242 | **0.104** | 0.142 | 0.149 | 0.136 |
| ETTh2 | Best | 0.325 | **0.324** | 0.344 | 0.356 | 0.383 | 0.350 | **0.321** | 0.360 | 0.355 | 0.338 | **0.324** | 0.325 | 0.337 | 0.349 | 0.382 | **0.321** | 0.359 |
| | Median | 0.538 | **0.433** | 0.453 | 0.462 | **0.410** | 0.546 | 0.462 | 0.589 | 0.504 | 0.558 | **0.430** | 0.436 | 0.623 | 0.458 | 0.520 | **0.423** | 0.500 |
| | IQR | **0.685** | 0.743 | 1.453 | 2.021 | **0.030** | 0.754 | 0.707 | 1.345 | 1.393 | 1.530 | **0.333** | 0.967 | 1.499 | 1.177 | 0.847 | 0.845 | **0.645** |
| ECL | Best | 0.163 | 0.163 | **0.157** | 0.163 | 0.165 | 0.160 | 0.162 | 0.158 | **0.157** | **0.158** | 0.158 | 0.159 | 0.158 | 0.162 | 0.181 | **0.157** | 0.169 |
| | Median | 0.213 | 0.204 | **0.201** | 0.205 | **0.181** | 0.207 | 0.209 | 0.199 | 0.195 | **0.194** | 0.213 | 0.199 | 0.209 | 0.183 | 0.242 | **0.182** | 0.209 |
| | IQR | 0.055 | 0.059 | **0.054** | 0.054 | **0.048** | 0.055 | 0.059 | 0.048 | 0.052 | 0.052 | 0.060 | **0.050** | 0.054 | **0.026** | 0.044 | 0.046 | 0.040 |
| traffic | Best | 0.409 | 0.408 | **0.394** | 0.407 | 0.417 | 0.401 | 0.407 | **0.394** | 0.399 | 0.407 | 0.399 | **0.394** | 0.402 | 0.409 | 0.515 | **0.394** | 0.446 |
| | Median | 0.592 | 0.608 | **0.558** | 0.576 | **0.475** | 0.583 | 0.599 | 0.596 | 0.523 | 0.540 | 0.655 | **0.510** | 0.538 | 0.479 | 0.686 | **0.453** | 0.578 |
| | IQR | **0.181** | 0.210 | 0.195 | 0.208 | **0.102** | 0.181 | 0.195 | 0.199 | 0.190 | 0.149 | **0.140** | 0.190 | 0.162 | **0.140** | 0.128 | 0.177 | 0.143 |
| weather | Best | 0.222 | **0.220** | 0.221 | 0.227 | **0.210** | 0.229 | 0.220 | 0.226 | 0.221 | 0.227 | **0.220** | 0.221 | 0.223 | 0.225 | 0.253 | **0.220** | 0.237 |
| | Median | **0.261** | 0.267 | 0.266 | 0.279 | **0.233** | 0.264 | 0.264 | 0.274 | 0.250 | 0.273 | **0.254** | 0.267 | 0.273 | 0.248 | 0.286 | **0.241** | 0.257 |
| | IQR | **0.046** | 0.049 | 0.054 | 0.065 | **0.018** | 0.047 | 0.048 | 0.043 | 0.060 | 0.049 | **0.047** | 0.050 | 0.089 | 0.040 | 0.030 | 0.057 | **0.028** |
| Exchange | Best | **0.210** | 0.237 | 0.256 | 0.269 | 0.406 | 0.278 | **0.208** | 0.263 | 0.282 | 0.246 | 0.237 | **0.215** | 0.246 | 0.256 | **0.209** | 0.291 | 0.238 |
| | Median | 0.540 | **0.430** | 0.574 | 0.602 | 0.615 | 0.545 | **0.478** | 0.560 | 0.590 | 0.542 | **0.439** | 0.558 | 0.562 | 0.493 | **0.398** | 0.841 | 0.427 |
| | IQR | 0.465 | **0.404** | 0.517 | 0.499 | **0.164** | 0.600 | 0.451 | 0.492 | 0.492 | 0.620 | **0.300** | 0.547 | 0.484 | 0.344 | **0.192** | 0.799 | 0.258 |
| ili | Best | 1.608 | 1.561 | **1.551** | 1.597 | 1.665 | 1.672 | **1.561** | 1.637 | 1.642 | 1.603 | 1.618 | 1.629 | **1.552** | 1.869 | 1.715 | 2.269 | **1.546** |
| | Median | 2.946 | 2.855 | **2.761** | 2.731 | **2.451** | 2.949 | 2.889 | 2.791 | 2.728 | **2.648** | 3.058 | 2.788 | 2.760 | 2.641 | 2.703 | 3.797 | **2.472** |
| | IQR | 1.767 | **1.515** | 1.661 | 1.623 | **0.656** | 1.652 | 1.651 | 1.746 | 1.642 | **1.551** | 1.698 | 1.595 | 1.722 | **1.221** | 1.569 | 1.643 | 1.664 |

(b) Part III – 7 Components (d_model, d_ff, etc.)

| dataset | stat | Hidden Layer Dimensions | | FCN Layer Dimensions | | Encoder layers | | Training Epochs | | | Loss Function | | | Learning Rate | | Learning Rate Strategy | |
| | | 256 | 64 | 1024 | 256 | 2 | 3 | 10 | 20 | 50 | HUBER | MAE | MSE | 0.0001 | 0.001 | null | type |
|---|---|---|---|---|---|---|---|---|---|---|---|---|---|---|---|---|---|
| ETTm1 | Best | 0.349 | **0.348** | 0.349 | 0.348 | **0.348** | 0.348 | 0.351 | 0.351 | **0.347** | 0.348 | 0.351 | 0.350 | **0.347** | 0.351 | **0.348** | 0.348 |
| | Median | 0.461 | **0.425** | 0.461 | 0.425 | **0.424** | 0.452 | 0.444 | 0.443 | **0.434** | 0.417 | 0.463 | 0.442 | **0.427** | 0.447 | **0.433** | 0.444 |
| | IQR | 0.165 | **0.161** | 0.165 | 0.161 | **0.158** | 0.160 | 0.173 | **0.154** | 0.159 | **0.149** | 0.172 | 0.166 | **0.156** | 0.175 | **0.148** | 0.173 |
| ETTm2 | Best | **0.248** | 0.252 | **0.248** | 0.252 | **0.248** | 0.252 | 0.253 | **0.248** | 0.253 | 0.255 | **0.248** | 0.253 | 0.254 | **0.248** | **0.248** | 0.253 |
| | Median | 0.357 | **0.342** | 0.357 | **0.342** | 0.352 | **0.346** | 0.342 | 0.367 | **0.340** | 0.323 | **0.317** | 0.370 | **0.332** | 0.365 | 0.361 | **0.342** |
| | IQR | 0.698 | **0.405** | 0.698 | **0.405** | 0.636 | **0.467** | **0.444** | 0.613 | 0.556 | 0.392 | **0.331** | 0.665 | 0.461 | 0.670 | **0.500** | 0.599 |
| ETTh1 | Best | **0.401** | 0.406 | **0.401** | 0.406 | 0.406 | **0.401** | **0.401** | 0.404 | 0.406 | 0.405 | **0.401** | 0.417 | **0.401** | 0.407 | **0.401** | 0.402 |
| | Median | 0.491 | **0.487** | 0.491 | **0.487** | 0.486 | 0.493 | 0.493 | **0.485** | 0.489 | 0.491 | **0.485** | 0.498 | 0.479 | 0.501 | **0.486** | 0.494 |
| | IQR | **0.127** | 0.128 | **0.127** | 0.128 | **0.119** | 0.137 | 0.131 | **0.120** | 0.126 | 0.152 | 0.118 | **0.104** | 0.109 | 0.143 | **0.110** | 0.150 |
| ETTh2 | Best | 0.325 | **0.323** | 0.325 | **0.323** | 0.325 | **0.323** | 0.330 | 0.335 | **0.321** | 0.326 | 0.326 | 0.336 | 0.326 | **0.322** | **0.323** | 0.325 |
| | Median | 0.468 | **0.462** | 0.468 | **0.462** | 0.448 | 0.491 | 0.507 | **0.455** | 0.457 | 0.455 | 0.461 | 0.471 | 0.466 | **0.462** | **0.459** | 0.472 |
| | IQR | 0.859 | **0.778** | 0.859 | **0.778** | 0.760 | 0.871 | 0.945 | **0.733** | 0.750 | **0.630** | 0.906 | 0.901 | 0.811 | 0.827 | 0.836 | **0.825** |
| ECL | Best | **0.157** | 0.160 | **0.157** | 0.160 | 0.158 | **0.157** | 0.159 | 0.159 | **0.157** | 0.158 | 0.159 | **0.157** | **0.157** | 0.158 | **0.157** | 0.158 |
| | Median | **0.204** | 0.207 | **0.204** | 0.207 | 0.210 | **0.202** | 0.205 | 0.205 | 0.207 | 0.205 | **0.200** | 0.206 | 0.215 | 0.199 | **0.198** | 0.213 |
| | IQR | 0.057 | **0.056** | 0.057 | **0.056** | 0.057 | **0.054** | 0.056 | **0.056** | 0.057 | 0.052 | **0.046** | 0.057 | 0.061 | **0.050** | 0.049 | 0.059 |
| traffic | Best | **0.394** | 0.400 | **0.394** | 0.400 | 0.400 | **0.394** | 0.401 | 0.401 | **0.394** | 0.418 | 0.423 | **0.394** | 0.405 | **0.394** | 0.398 | **0.394** |
| | Median | **0.553** | 0.603 | **0.553** | 0.603 | **0.569** | 0.589 | 0.592 | 0.587 | **0.567** | 0.619 | 0.611 | **0.570** | 0.596 | **0.564** | 0.549 | 0.607 |
| | IQR | 0.195 | 0.202 | **0.195** | 0.202 | 0.194 | 0.194 | 0.207 | 0.193 | **0.191** | 0.143 | 0.194 | 0.195 | 0.216 | **0.187** | **0.184** | 0.208 |
| weather | Best | **0.220** | 0.220 | **0.220** | 0.220 | **0.220** | 0.220 | 0.224 | **0.220** | 0.220 | 0.222 | 0.224 | **0.220** | 0.222 | **0.220** | 0.221 | **0.220** |
| | Median | 0.268 | **0.260** | 0.268 | **0.260** | 0.265 | 0.265 | 0.264 | **0.260** | 0.267 | 0.266 | **0.254** | 0.266 | 0.264 | 0.264 | 0.262 | 0.268 |
| | IQR | 0.051 | **0.049** | 0.051 | **0.049** | 0.052 | **0.049** | 0.044 | **0.046** | 0.057 | 0.051 | **0.049** | 0.050 | 0.047 | 0.051 | **0.047** | 0.051 |
| Exchange | Best | 0.237 | **0.209** | 0.237 | **0.209** | 0.210 | 0.236 | 0.238 | **0.209** | 0.246 | 0.239 | **0.210** | 0.241 | **0.209** | 0.237 | 0.243 | **0.209** |
| | Median | 0.517 | **0.461** | 0.517 | **0.461** | 0.490 | **0.489** | 0.486 | **0.481** | 0.507 | 0.483 | **0.475** | 0.509 | **0.438** | 0.545 | 0.496 | **0.486** |
| | IQR | 0.513 | **0.410** | 0.513 | **0.410** | 0.484 | **0.441** | 0.459 | **0.439** | 0.467 | 0.482 | **0.438** | 0.468 | **0.370** | 0.549 | 0.488 | **0.443** |
| ili | Best | 1.546 | 1.630 | **1.546** | 1.630 | 1.561 | **1.553** | 1.586 | **1.553** | 1.613 | **1.582** | 1.590 | 1.585 | 1.662 | **1.545** | 1.587 | 1.563 |
| | Median | 2.741 | 2.987 | **2.741** | 2.987 | **2.858** | 2.861 | 2.895 | **2.816** | 2.866 | 2.924 | 2.896 | **2.801** | 3.214 | **2.631** | **2.690** | 3.206 |
| | IQR | **1.575** | 1.735 | **1.575** | 1.735 | **1.633** | 1.669 | 1.700 | **1.619** | 1.632 | **1.641** | 1.649 | 1.653 | 1.775 | **1.399** | **1.421** | 1.836 |

Table H12: Long-term Forecasting Performance of Different Design Choices – Part I (6 Components). **MAE** distribution for each dataset under different configurations of the 6 components, characterized by the best (minimum) value, median, and interquartile range (IQR). **Bolded** entries indicate the best-performing result for the respective dataset and metric in each component.

| dataset | stat | Timestamp Embedding | | Series Sampling/Mixing | | Series Normalization | | | | Series Decomposition | | | | Channel Independent | | Series Tokenization | | |
|---|---|---|---|---|---|---|---|---|---|---|---|---|---|---|---|---|---|---|
| | | w/o Embedding | w/ Embedding | w/o Mixing | w/ Mixing | DishTS | w/o Norm | RevIN | Stat | DFT | MA | MoEMA | w/o Decomp | Channel Depen | Channel Indepen | Inverted Encoding | Series Encoding | Series Patching |
| ETTm1 | Best | **0.369** | 0.37 | **0.368** | 0.376 | 0.377 | 0.374 | **0.369** | 0.371 | 0.372 | 0.373 | **0.371** | 0.371 | 0.373 | **0.37** | 0.373 | **0.371** | 0.371 |
| | Median | **0.422** | 0.447 | **0.429** | 0.445 | 0.487 | 0.538 | **0.413** | 0.413 | 0.445 | **0.416** | 0.438 | 0.448 | 0.452 | **0.402** | 0.412 | 0.454 | **0.4** |
| | IQR | **0.079** | 0.105 | **0.08** | 0.127 | 0.129 | 0.17 | **0.049** | 0.053 | 0.101 | **0.067** | 0.103 | 0.121 | 0.127 | **0.056** | 0.064 | 0.138 | **0.053** |
| ETTm2 | Best | 0.308 | **0.306** | **0.306** | 0.312 | 0.319 | 0.329 | **0.306** | 0.308 | 0.308 | **0.306** | 0.312 | 0.307 | 0.307 | **0.306** | 0.307 | 0.312 | **0.306** |
| | Median | 0.385 | **0.382** | **0.368** | 0.403 | 0.531 | 0.687 | **0.336** | 0.339 | 0.371 | 0.399 | 0.393 | **0.359** | 0.396 | **0.353** | **0.342** | 0.406 | 0.354 |
| | IQR | 0.248 | 0.268 | **0.19** | 0.377 | 0.242 | 0.51 | **0.026** | 0.027 | **0.184** | 0.314 | 0.297 | 0.226 | 0.36 | **0.096** | 0.131 | 0.4 | **0.114** |
| ETTh1 | Best | **0.417** | 0.421 | **0.418** | 0.42 | 0.439 | 0.431 | 0.419 | **0.418** | **0.42** | 0.42 | 0.42 | 0.421 | 0.425 | **0.417** | 0.425 | 0.421 | **0.418** |
| | Median | 0.474 | 0.474 | **0.466** | 0.492 | 0.515 | 0.575 | **0.45** | 0.45 | **0.472** | 0.474 | 0.475 | 0.473 | 0.487 | **0.452** | 0.461 | 0.494 | **0.446** |
| | IQR | **0.082** | 0.093 | **0.062** | 0.148 | 0.139 | 0.249 | **0.036** | 0.037 | 0.105 | **0.081** | 0.096 | 0.081 | 0.144 | **0.034** | 0.045 | 0.166 | **0.029** |
| ETTh2 | Best | **0.377** | 0.382 | 0.382 | **0.378** | 0.405 | 0.409 | **0.377** | 0.381 | 0.381 | **0.378** | 0.383 | 0.384 | **0.379** | 0.381 | 0.387 | **0.378** | 0.382 |
| | Median | 0.452 | 0.474 | **0.45** | 0.473 | 0.687 | 0.973 | **0.41** | 0.41 | **0.437** | 0.452 | 0.477 | 0.475 | 0.487 | **0.417** | 0.446 | 0.47 | **0.414** |
| | IQR | 0.333 | 0.367 | **0.281** | 0.604 | 0.405 | 0.951 | **0.025** | 0.027 | **0.263** | 0.4 | 0.425 | 0.329 | 0.552 | **0.119** | 0.198 | 0.667 | **0.134** |
| ECL | Best | 0.254 | **0.252** | **0.252** | 0.257 | 0.253 | 0.258 | 0.256 | **0.252** | **0.252** | 0.258 | 0.258 | 0.255 | **0.252** | 0.259 | **0.252** | 0.255 | 0.261 |
| | Median | 0.303 | **0.301** | **0.3** | 0.306 | 0.317 | 0.323 | **0.286** | 0.287 | **0.299** | 0.306 | 0.303 | 0.3 | 0.305 | **0.291** | 0.288 | 0.312 | **0.286** |
| | IQR | 0.051 | **0.048** | **0.049** | 0.049 | 0.044 | 0.057 | **0.03** | 0.042 | 0.053 | **0.047** | 0.05 | 0.048 | 0.052 | **0.038** | 0.04 | 0.056 | **0.034** |
| traffic | Best | 0.278 | **0.273** | **0.273** | 0.277 | 0.278 | 0.289 | 0.281 | **0.273** | 0.28 | **0.273** | 0.279 | 0.273 | **0.273** | 0.281 | **0.273** | 0.277 | 0.281 |
| | Median | **0.35** | 0.36 | 0.358 | **0.351** | 0.357 | 0.381 | 0.349 | **0.337** | 0.361 | 0.356 | 0.352 | **0.349** | **0.353** | 0.371 | **0.341** | 0.362 | 0.356 |
| | IQR | **0.068** | 0.071 | 0.077 | **0.058** | **0.055** | 0.081 | 0.071 | 0.067 | 0.071 | 0.073 | 0.069 | **0.067** | **0.068** | 0.075 | 0.082 | **0.064** | 0.07 |
| weather | Best | **0.253** | 0.256 | **0.253** | 0.254 | 0.259 | 0.264 | **0.253** | 0.256 | 0.262 | **0.253** | 0.258 | 0.254 | 0.257 | **0.253** | 0.257 | 0.257 | **0.253** |
| | Median | **0.293** | 0.298 | **0.294** | 0.298 | 0.319 | 0.34 | **0.282** | 0.283 | 0.295 | **0.292** | 0.292 | 0.296 | 0.302 | **0.284** | 0.283 | 0.307 | **0.281** |
| | IQR | **0.041** | 0.075 | **0.046** | 0.07 | 0.062 | 0.162 | **0.021** | 0.024 | 0.053 | 0.051 | **0.046** | 0.051 | 0.066 | **0.025** | 0.027 | 0.079 | **0.026** |
| Exchange | Best | 0.35 | **0.33** | **0.33** | 0.351 | **0.331** | 0.356 | 0.396 | 0.391 | 0.353 | 0.349 | 0.351 | **0.331** | **0.331** | 0.348 | 0.353 | **0.333** | 0.348 |
| | Median | 0.47 | 0.471 | **0.455** | 0.501 | 0.533 | 0.734 | 0.435 | **0.433** | **0.462** | 0.474 | 0.466 | 0.481 | 0.5 | **0.424** | 0.434 | 0.512 | **0.422** |
| | IQR | 0.215 | 0.42 | **0.193** | 0.24 | 0.272 | 0.42 | 0.069 | **0.067** | **0.198** | 0.223 | 0.207 | 0.226 | 0.258 | **0.078** | 0.127 | 0.264 | **0.072** |
| ili | Best | 0.804 | **0.76** | 0.782 | **0.761** | 0.832 | 0.974 | **0.763** | 0.787 | 0.807 | 0.807 | 0.78 | **0.771** | **0.76** | 0.82 | 0.786 | **0.761** | 0.82 |
| | Median | **1.142** | 1.157 | 1.151 | **1.149** | 1.116 | 1.443 | 1.059 | **1.057** | 1.16 | **1.14** | 1.156 | 1.151 | **1.14** | 1.187 | 1.156 | 1.151 | **1.137** |
| | IQR | **0.369** | 0.387 | **0.374** | 0.389 | 0.336 | 0.212 | **0.211** | 0.229 | **0.372** | 0.375 | 0.385 | 0.378 | **0.376** | 0.391 | 0.399 | 0.379 | **0.332** |

Table H13: Long-term Forecasting Performance of Different Design Choices– Part II (4 Components) and Part II (7 Components). Same structure and evaluation metrics (MAE) as Table H12.

(a) Part II – 4 Components (Backbone, Attention, etc.)

| dataset | stat | Network Backbone | | | Attention | | | | | | Feature-Attention | | | | Sequence Length | | | |
|---|---|---|---|---|---|---|---|---|---|---|---|---|---|---|---|---|---|---|
| | | GRU | MLP | Trans-former | AutoCorr | De-stationary Attention | Frequency Attention | w/o Attention | Self Attention | Sparse Attention | Frequency Attention | w/o Attention | Self Attention | Sparse Attention | 192 | 48 | 512 | 96 |
| ETTm1 | Best | 0.375 | **0.37** | 0.37 | 0.377 | 0.406 | **0.37** | 0.37 | 0.386 | 0.379 | 0.381 | **0.368** | 0.379 | 0.382 | 0.371 | 0.427 | **0.37** | 0.382 |
| | Median | 0.448 | **0.419** | 0.439 | 0.463 | 0.437 | **0.416** | 0.435 | 0.455 | 0.431 | 0.443 | **0.42** | 0.449 | 0.451 | **0.406** | 0.473 | 0.413 | 0.426 |
| | IQR | 0.091 | 0.096 | **0.087** | 0.171 | **0.03** | 0.059 | 0.092 | 0.131 | 0.079 | 0.119 | **0.072** | 0.112 | 0.128 | 0.073 | 0.076 | 0.083 | **0.065** |
| ETTm2 | Best | 0.312 | **0.306** | 0.31 | 0.315 | 0.335 | 0.312 | **0.306** | 0.319 | 0.315 | 0.313 | **0.306** | 0.308 | 0.309 | 0.309 | 0.329 | **0.306** | 0.316 |
| | Median | 0.384 | **0.364** | 0.42 | 0.505 | **0.355** | 0.361 | 0.373 | 0.442 | 0.512 | 0.42 | **0.362** | 0.396 | 0.394 | **0.366** | 0.408 | 0.373 | 0.401 |
| | IQR | 0.213 | **0.167** | 0.365 | 0.408 | **0.018** | 0.313 | 0.188 | 0.332 | 0.428 | 0.396 | **0.14** | 0.358 | 0.341 | 0.287 | **0.229** | 0.25 | 0.293 |
| ETTh1 | Best | 0.422 | **0.418** | 0.421 | 0.427 | 0.456 | 0.425 | **0.418** | 0.433 | 0.422 | 0.427 | **0.417** | 0.429 | 0.429 | 0.423 | 0.43 | **0.417** | 0.425 |
| | Median | 0.478 | **0.464** | 0.483 | 0.488 | 0.493 | **0.462** | 0.472 | 0.503 | 0.484 | 0.478 | **0.466** | 0.481 | 0.487 | **0.469** | 0.478 | 0.478 | 0.469 |
| | IQR | **0.08** | 0.081 | 0.117 | 0.138 | **0.028** | 0.049 | 0.082 | 0.128 | 0.146 | 0.118 | **0.06** | 0.11 | 0.168 | **0.074** | 0.104 | 0.093 | 0.099 |
| ETTh2 | Best | **0.378** | 0.38 | 0.389 | 0.395 | 0.407 | 0.393 | **0.377** | 0.396 | 0.395 | 0.385 | **0.38** | 0.38 | 0.387 | 0.386 | 0.396 | **0.378** | 0.388 |
| | Median | 0.491 | **0.44** | 0.456 | 0.46 | **0.422** | 0.51 | 0.457 | 0.52 | 0.477 | 0.51 | **0.438** | 0.442 | 0.537 | 0.459 | 0.484 | **0.443** | 0.472 |
| | IQR | 0.316 | **0.313** | 0.552 | 0.7 | **0.017** | 0.327 | 0.318 | 0.563 | 0.543 | 0.615 | **0.164** | 0.425 | 0.609 | 0.456 | 0.356 | 0.362 | **0.286** |
| ECL | Best | 0.263 | 0.257 | **0.252** | 0.259 | 0.263 | 0.258 | 0.257 | 0.253 | **0.252** | 0.255 | **0.252** | 0.258 | 0.253 | 0.258 | 0.275 | **0.252** | 0.264 |
| | Median | 0.309 | 0.299 | **0.298** | 0.302 | **0.276** | 0.306 | 0.305 | 0.296 | 0.289 | **0.292** | 0.308 | 0.298 | 0.307 | **0.282** | 0.325 | 0.286 | 0.301 |
| | IQR | 0.05 | **0.048** | 0.048 | 0.05 | **0.034** | 0.044 | 0.049 | 0.046 | 0.049 | 0.048 | 0.049 | **0.047** | 0.051 | **0.032** | 0.038 | 0.049 | 0.043 |
| traffic | Best | 0.281 | 0.284 | **0.273** | 0.284 | 0.291 | **0.273** | 0.281 | 0.279 | 0.278 | **0.273** | 0.276 | 0.278 | 0.285 | **0.273** | 0.321 | 0.278 | 0.28 |
| | Median | 0.36 | 0.369 | **0.344** | 0.35 | 0.333 | 0.351 | 0.364 | 0.35 | **0.331** | 0.345 | 0.378 | **0.338** | 0.342 | 0.332 | 0.399 | 0.342 | 0.358 |
| | IQR | **0.06** | 0.081 | 0.068 | 0.066 | **0.051** | 0.062 | 0.071 | 0.075 | 0.069 | **0.049** | 0.061 | 0.069 | 0.069 | **0.047** | 0.066 | 0.048 | 0.051 |
| weather | Best | **0.253** | 0.257 | 0.259 | 0.263 | 0.254 | 0.266 | **0.253** | 0.263 | 0.261 | 0.265 | **0.253** | 0.257 | 0.258 | 0.254 | 0.281 | **0.253** | 0.266 |
| | Median | **0.293** | 0.295 | 0.302 | 0.313 | **0.267** | 0.3 | 0.294 | 0.305 | 0.295 | 0.297 | **0.291** | 0.303 | 0.304 | 0.287 | 0.308 | **0.285** | 0.291 |
| | IQR | **0.043** | 0.051 | 0.055 | 0.065 | **0.017** | 0.043 | 0.047 | 0.05 | 0.06 | 0.06 | **0.035** | 0.049 | 0.085 | **0.038** | 0.056 | 0.052 | 0.039 |
| Exchange | Best | **0.331** | 0.348 | 0.366 | 0.377 | 0.429 | 0.372 | **0.33** | 0.369 | 0.377 | 0.352 | 0.348 | **0.336** | 0.356 | 0.367 | **0.33** | 0.383 | 0.348 |
| | Median | 0.494 | **0.443** | 0.508 | 0.527 | 0.511 | 0.514 | **0.465** | 0.503 | 0.516 | 0.494 | **0.449** | 0.493 | 0.497 | 0.474 | **0.424** | 0.622 | 0.443 |
| | IQR | 0.222 | **0.177** | 0.255 | 0.241 | **0.076** | 0.284 | 0.205 | 0.237 | 0.24 | 0.265 | **0.158** | 0.243 | 0.222 | 0.161 | **0.101** | 0.306 | 0.135 |
| ili | Best | 0.805 | 0.774 | **0.769** | 0.816 | 0.812 | 0.832 | **0.774** | 0.796 | 0.795 | 0.807 | 0.818 | 0.816 | 0.76 | 0.876 | 0.811 | 1.009 | **0.76** |
| | Median | 1.174 | 1.147 | **1.133** | 1.13 | **1.029** | 1.179 | 1.157 | 1.135 | 1.114 | **1.112** | 1.184 | 1.141 | 1.14 | 1.086 | 1.079 | 1.388 | **1.058** |
| | IQR | 0.394 | **0.358** | 0.381 | 0.381 | **0.2** | 0.383 | 0.379 | 0.381 | 0.384 | 0.353 | 0.396 | 0.36 | 0.388 | **0.269** | 0.401 | 0.34 | 0.368 |

(b) Part III – 7 Components (d_model, d_ff, etc.)

| dataset | stat | Hidden Layer Dimensions | | FCN Layer Dimensions | | Encoder layers | | Training Epochs | | | Loss Function | | | Learning Rate | | Learning Rate Strategy | |
|---|---|---|---|---|---|---|---|---|---|---|---|---|---|---|---|---|---|
| | | 256 | 64 | 1024 | 256 | 2 | 3 | 10 | 20 | 50 | HUBER | MAE | MSE | 0.0001 | 0.001 | null | type |
| ETTm1 | Best | 0.372 | **0.368** | 0.372 | **0.368** | **0.368** | 0.372 | 0.37 | 0.375 | **0.369** | 0.371 | **0.368** | 0.379 | 0.371 | **0.369** | 0.371 | **0.369** |
| | Median | 0.443 | **0.428** | 0.443 | **0.428** | 0.427 | 0.444 | 0.437 | 0.437 | **0.433** | 0.418 | 0.441 | 0.44 | **0.43** | 0.441 | **0.431** | 0.441 |
| | IQR | 0.1 | **0.084** | 0.1 | **0.084** | **0.088** | 0.094 | 0.097 | **0.081** | 0.099 | **0.074** | 0.115 | 0.092 | 0.086 | 0.098 | **0.085** | 0.097 |
| ETTm2 | Best | **0.306** | 0.307 | **0.306** | 0.307 | **0.306** | 0.308 | 0.308 | **0.306** | 0.308 | 0.312 | **0.306** | 0.316 | 0.31 | **0.306** | **0.306** | 0.31 |
| | Median | 0.385 | **0.374** | 0.385 | **0.374** | 0.38 | 0.38 | **0.372** | 0.391 | 0.38 | 0.363 | **0.354** | 0.395 | **0.372** | 0.389 | 0.385 | **0.376** |
| | IQR | 0.307 | **0.218** | 0.307 | **0.218** | 0.286 | **0.241** | 0.24 | 0.294 | 0.26 | 0.187 | **0.17** | 0.307 | 0.246 | 0.284 | **0.249** | 0.288 |
| ETTh1 | Best | **0.417** | 0.42 | **0.417** | 0.42 | 0.419 | **0.418** | **0.418** | 0.419 | 0.422 | 0.422 | **0.417** | 0.429 | **0.418** | 0.419 | **0.418** | 0.418 |
| | Median | 0.475 | **0.472** | 0.475 | **0.472** | 0.472 | 0.476 | 0.481 | **0.472** | 0.473 | 0.475 | **0.468** | 0.484 | **0.466** | 0.48 | **0.47** | 0.477 |
| | IQR | 0.091 | **0.081** | 0.091 | **0.081** | **0.079** | 0.098 | 0.095 | **0.081** | 0.086 | 0.106 | 0.077 | **0.071** | **0.074** | 0.102 | **0.076** | 0.101 |
| ETTh2 | Best | 0.381 | **0.378** | 0.381 | **0.378** | **0.377** | 0.38 | 0.384 | 0.381 | **0.378** | 0.38 | **0.377** | 0.389 | 0.382 | **0.377** | 0.38 | **0.377** |
| | Median | 0.459 | **0.457** | 0.459 | **0.457** | **0.45** | 0.472 | 0.481 | 0.469 | **0.453** | 0.45 | 0.451 | 0.462 | 0.46 | **0.456** | 0.454 | 0.464 |
| | IQR | 0.367 | **0.343** | 0.367 | **0.343** | 0.342 | 0.371 | 0.392 | 0.333 | **0.323** | 0.288 | 0.376 | 0.388 | 0.358 | 0.361 | 0.362 | **0.353** |
| ECL | Best | **0.252** | 0.255 | **0.252** | 0.255 | **0.252** | 0.253 | 0.253 | 0.254 | **0.252** | **0.252** | 0.253 | 0.255 | **0.252** | 0.253 | **0.252** | 0.253 |
| | Median | **0.3** | 0.304 | **0.3** | 0.304 | 0.306 | **0.298** | 0.303 | **0.301** | 0.302 | 0.298 | **0.293** | 0.303 | 0.311 | **0.294** | **0.294** | 0.31 |
| | IQR | **0.049** | 0.05 | **0.049** | 0.05 | 0.051 | **0.046** | 0.048 | **0.048** | 0.05 | 0.043 | **0.038** | 0.05 | 0.051 | **0.043** | **0.046** | 0.052 |
| traffic | Best | **0.273** | 0.28 | **0.273** | 0.28 | **0.273** | 0.276 | 0.281 | 0.282 | **0.273** | 0.278 | **0.273** | 0.278 | **0.273** | 0.277 | **0.273** | 0.278 |
| | Median | **0.343** | 0.365 | **0.343** | 0.365 | 0.356 | **0.355** | 0.361 | 0.354 | **0.353** | 0.358 | **0.344** | 0.355 | 0.366 | **0.347** | **0.346** | 0.366 |
| | IQR | **0.068** | 0.073 | **0.068** | 0.073 | 0.072 | **0.068** | 0.077 | 0.067 | **0.066** | 0.053 | 0.069 | 0.071 | 0.093 | **0.058** | **0.062** | 0.088 |
| weather | Best | 0.254 | 0.253 | 0.254 | 0.253 | **0.253** | 0.254 | **0.253** | 0.258 | 0.254 | 0.259 | **0.253** | 0.261 | 0.254 | **0.253** | 0.256 | **0.253** |
| | Median | 0.297 | **0.294** | 0.297 | **0.294** | 0.297 | **0.294** | 0.296 | **0.293** | 0.297 | 0.292 | **0.28** | 0.298 | 0.295 | 0.297 | **0.294** | 0.297 |
| | IQR | 0.053 | **0.049** | 0.053 | **0.049** | 0.057 | **0.044** | 0.05 | 0.05 | 0.052 | 0.037 | **0.034** | 0.064 | **0.049** | 0.055 | **0.049** | 0.053 |
| Exchange | Best | 0.348 | **0.331** | 0.348 | **0.331** | **0.331** | 0.347 | 0.347 | **0.33** | 0.356 | 0.349 | **0.331** | 0.35 | **0.331** | 0.349 | 0.35 | **0.33** |
| | Median | 0.48 | **0.46** | 0.48 | **0.46** | **0.471** | 0.471 | 0.469 | **0.466** | 0.478 | 0.469 | **0.46** | 0.482 | **0.451** | 0.489 | 0.474 | **0.469** |
| | IQR | 0.223 | **0.198** | 0.223 | **0.198** | 0.22 | **0.206** | 0.22 | **0.197** | 0.219 | 0.209 | **0.183** | 0.235 | **0.187** | 0.229 | 0.22 | **0.204** |
| ili | Best | **0.76** | 0.805 | **0.76** | 0.805 | 0.782 | **0.761** | 0.807 | 0.437 | 0.795 | 0.791 | **0.763** | 0.828 | 0.808 | **0.76** | 0.798 | **0.76** |
| | Median | 1.126 | 1.181 | 1.126 | 1.181 | **1.15** | 1.151 | 1.158 | **1.135** | 1.156 | 1.163 | 1.15 | **1.145** | 1.241 | **1.084** | 1.099 | 1.237 |
| | IQR | **0.359** | 0.397 | **0.359** | 0.397 | **0.376** | 0.382 | 0.385 | 0.376 | **0.374** | 0.373 | 0.377 | 0.385 | 0.392 | **0.345** | **0.33** | 0.415 |

### H.2.1 DESIGN CHOICES EVALUATION RESULTS FOR LONG-TERM FORECASTING USING MSE AS THE METRIC

**Spider Chart Analysis**. Fig. H2 presents the large compoents-level experiments results by employing multi-dimensional spider charts, where each vertex corresponds to a benchmark dataset. Closer proximity to the outer edge of a vertex indicates better performance of the associated design choice on that particular dataset. These visual representations offer an intuitive understanding of how different architectural decisions influence model effectiveness across diverse forecasting domains. Notably, configurations for components including Series Sampling/Mixing (Fig. H2c), Hidden Layer Dimensions (Fig. H2j), FCN Layer Dimensions (Fig. H2k), Learning Rate (Fig. H2n), and Learning Rate Strategy (Fig. H2o) demonstrate similar spatial patterns in the radar charts. Specifically, ECL, ILI, and Traffic datasets exhibit consistent parameter preferences across these components, suggesting intrinsic alignment between their temporal patterns and specific architectural configurations.

In addition, Fig. H3 provides a evaluation of large-scale time series models, revealing that conventional architectures still maintain a competitive advantage over LLM-based models, especially in domain-specific forecasting tasks where structural inductive biases play a crucial role.

**Box Plots Analysis**. The impact of various design choices for each architectural component is further illustrated through box plots in Fig. H4 and Fig. H5. These visualizations complement the spider charts by providing a statistical perspective on performance variability and robustness across multiple benchmark datasets. Together, the two forms of analysis offer a comprehensive view of how different configurations affect forecasting accuracy.

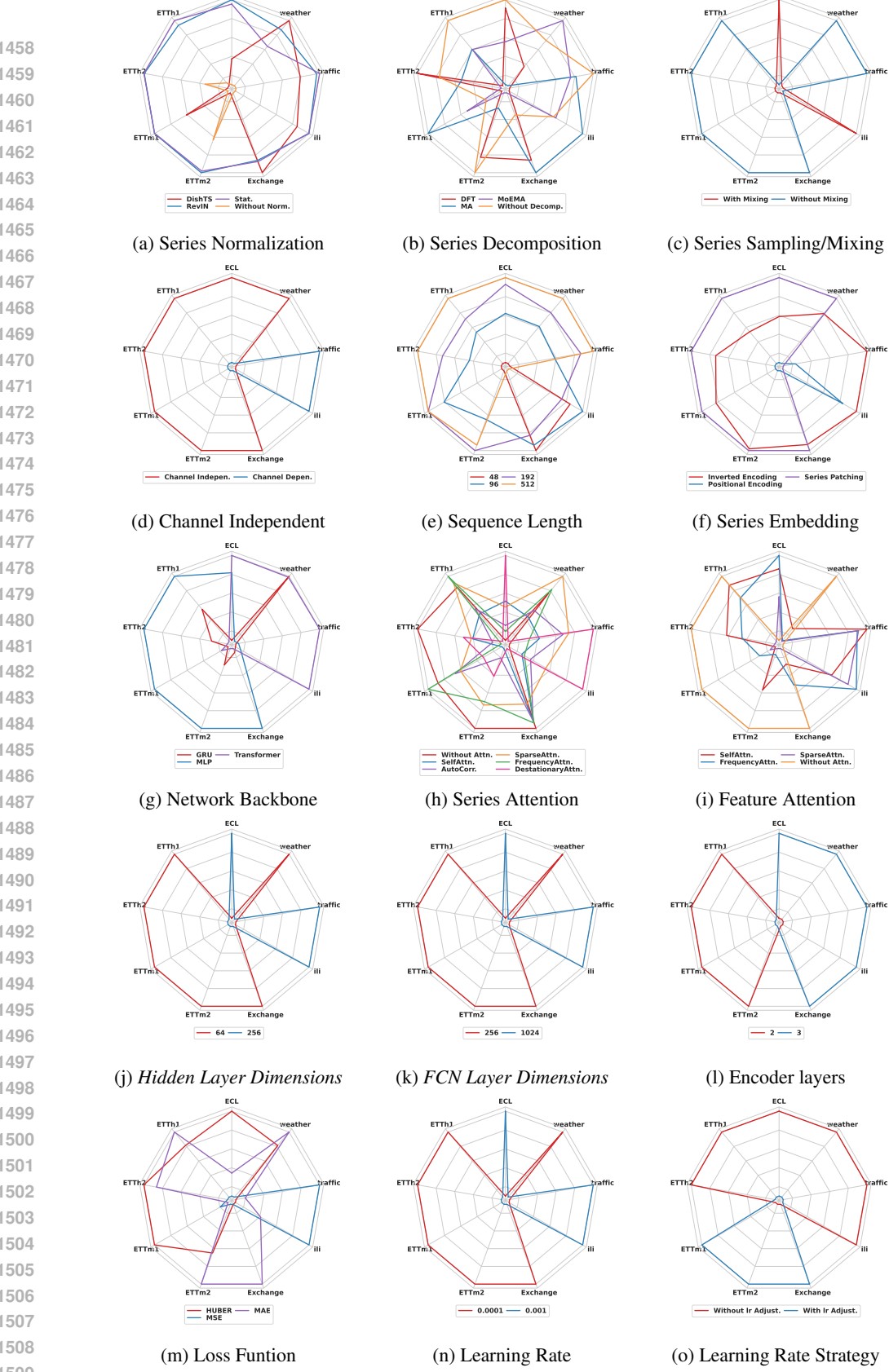

Figure H2: Overall performance across additional design dimensions in long-term forecasting. The results (MSE) are based on the top 25th percentile across all forecasting horizons.

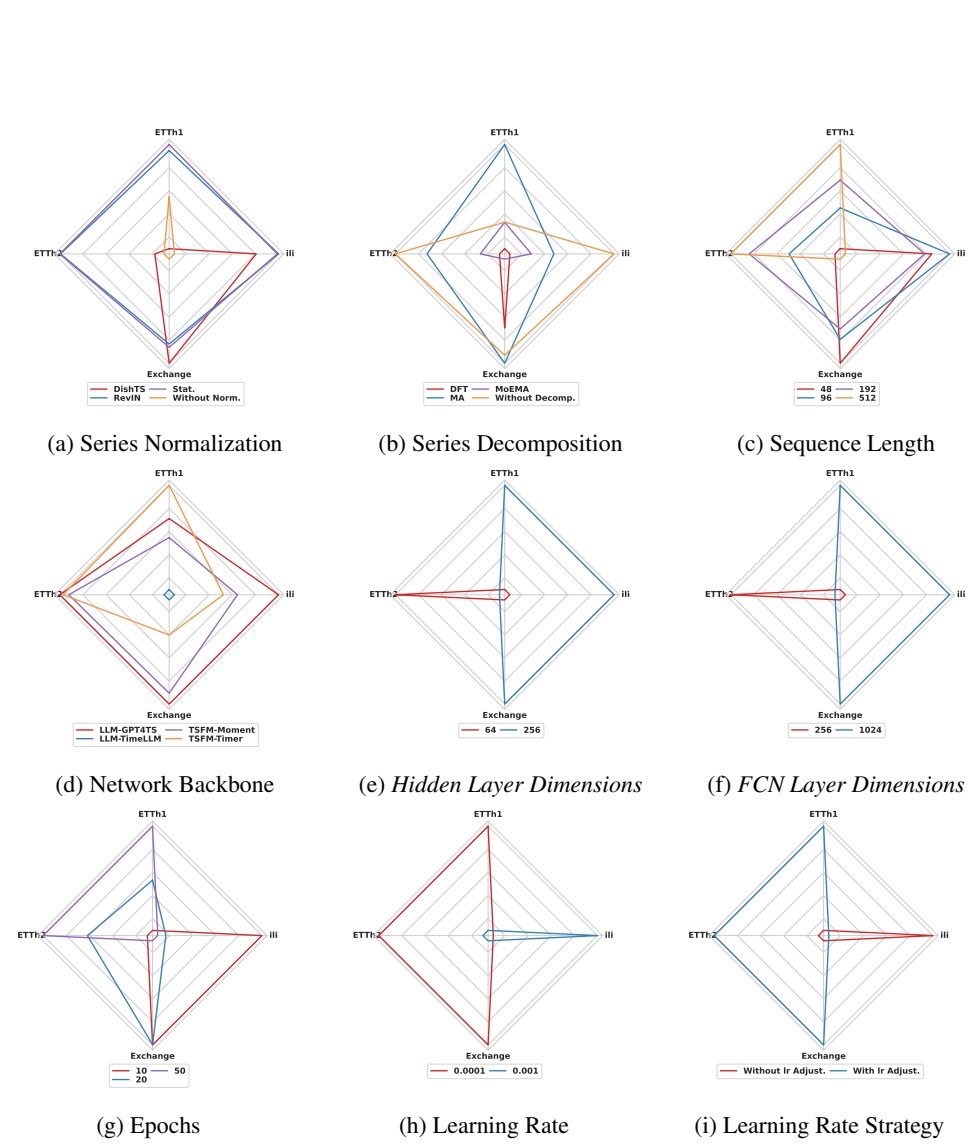

Figure H3: Overall performance across all design dimensions when using LLMs or TSFMs in long-term forecasting. The results (MSE) are based on the top 25th percentile across all forecasting horizons.

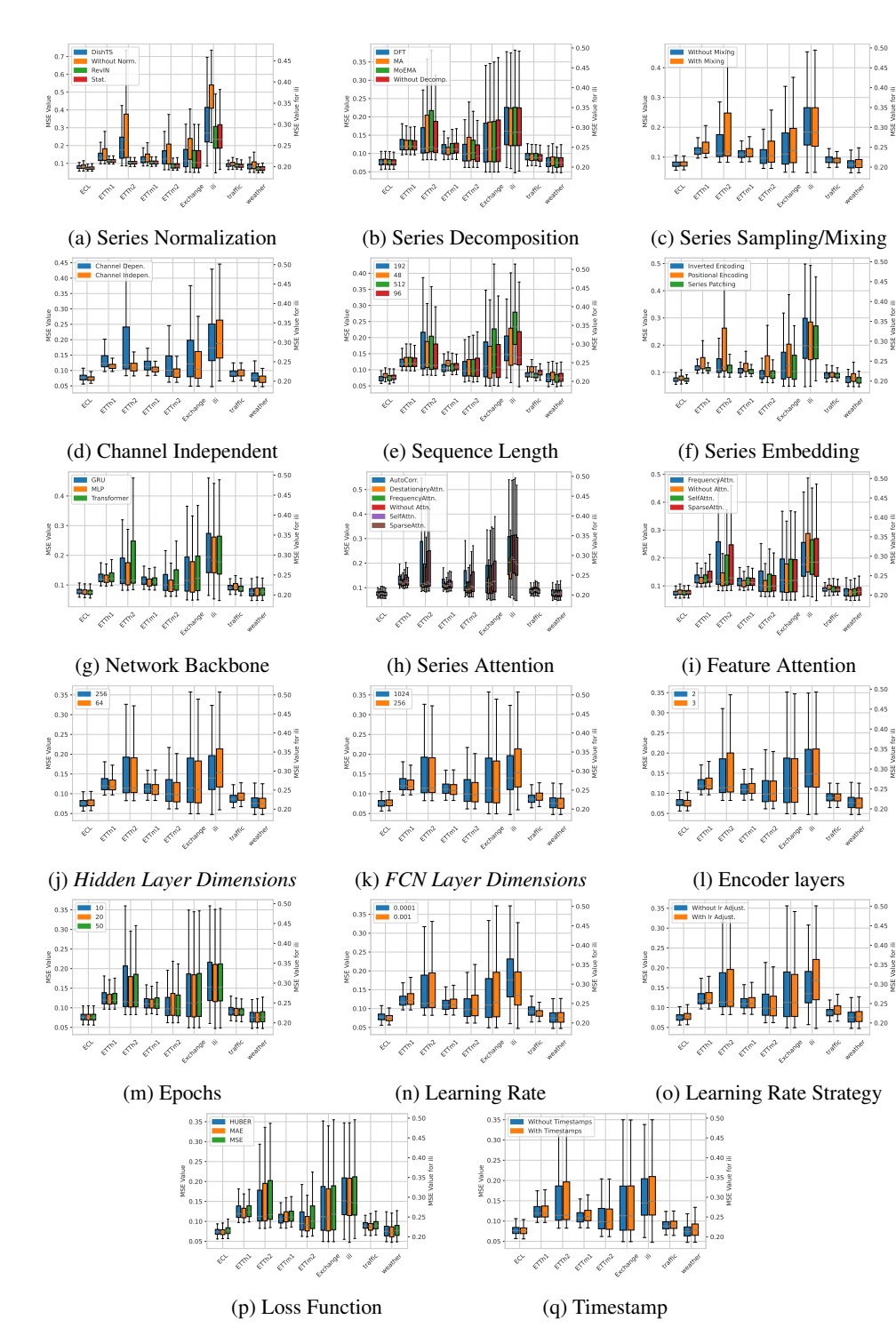

Figure H4: Overall performance across all design dimensions in long-term forecasting. The results (**MSE**) are averaged across all forecasting horizons. Due to the significantly different value range and variability of the ILI dataset compared to other datasets, its box plot is plotted using the right-hand *y*-axis, while all other datasets share the left-hand *y*-axis.

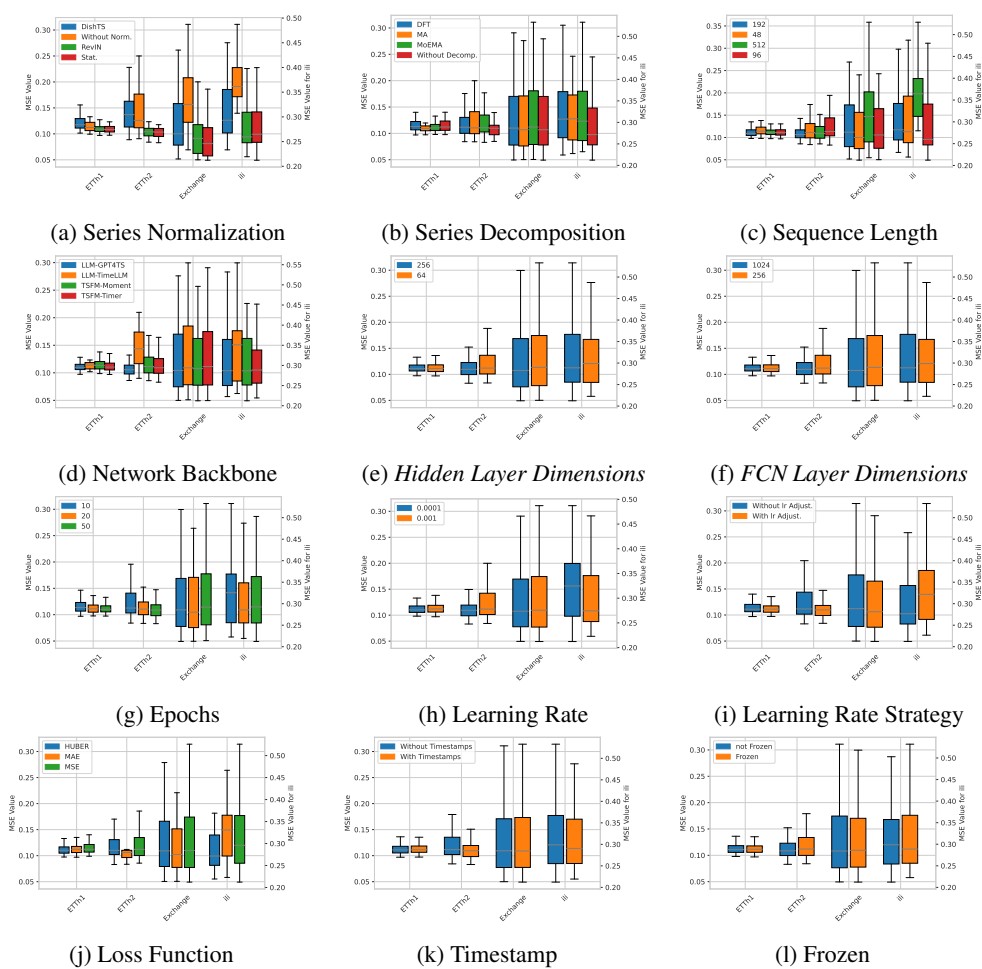

(a) Series Normalization  (b) Series Decomposition  (c) Sequence Length

(d) Network Backbone  (e) *Hidden Layer Dimensions*  (f) *FCN Layer Dimensions*

(g) Epochs  (h) Learning Rate  (i) Learning Rate Strategy

(j) Loss Function  (k) Timestamp  (l) Frozen

Figure H5: Overall performance across all design dimensions when using LLMs or TSFMs in long-term forecasting. The results (**MSE**) are averaged across all forecasting horizons. Due to the significantly different value range and variability of the ILI dataset compared to other datasets, its box plot is plotted using the right-hand *y*-axis, while all other datasets share the left-hand *y*-axis.

### H.2.2 DESIGN CHOICES EVALUATION RESULTS FOR LONG-TERM FORECASTING USING MAE AS THE METRIC

For the MAE-based performance evaluation, we analyze the effects of different design choices using both spider charts and box plots (Fig. H6 and Fig. H7). These visualizations complement the MSE-based analysis and confirm the generalizability of our findings across error metrics. In particular, normalization methods such as RevIN and Stationary consistently achieve the lowest MAE values, underscoring their effectiveness in mitigating non-stationarity. Similarly, decomposition strategies exhibit selective benefits: MA-based methods improve predictions on datasets like ETTh1 and ETTm2, while raw-series modeling remains more effective on ECL and Traffic, where decomposition tends to degrade performance.

Beyond preprocessing, MAE evaluations further validate the consistency of our architectural insights. Channel-independent designs retain strong performance across most datasets, except on Traffic and ILI, where localized dependencies dominate. Tokenization methods show stable ranking across both metrics, with patch-wise encoding consistently outperforming point-wise approaches. Notably, complex architectures such as Transformers provide only marginal gains over MLPs in certain cases (e.g., Traffic), suggesting that their benefits may not justify the added complexity. Overall, the alignment between MAE and MSE results reinforces the robustness of our design principles, demonstrating that the observed patterns are not metric-specific but instead reflect core relationships between architecture and forecasting performance.

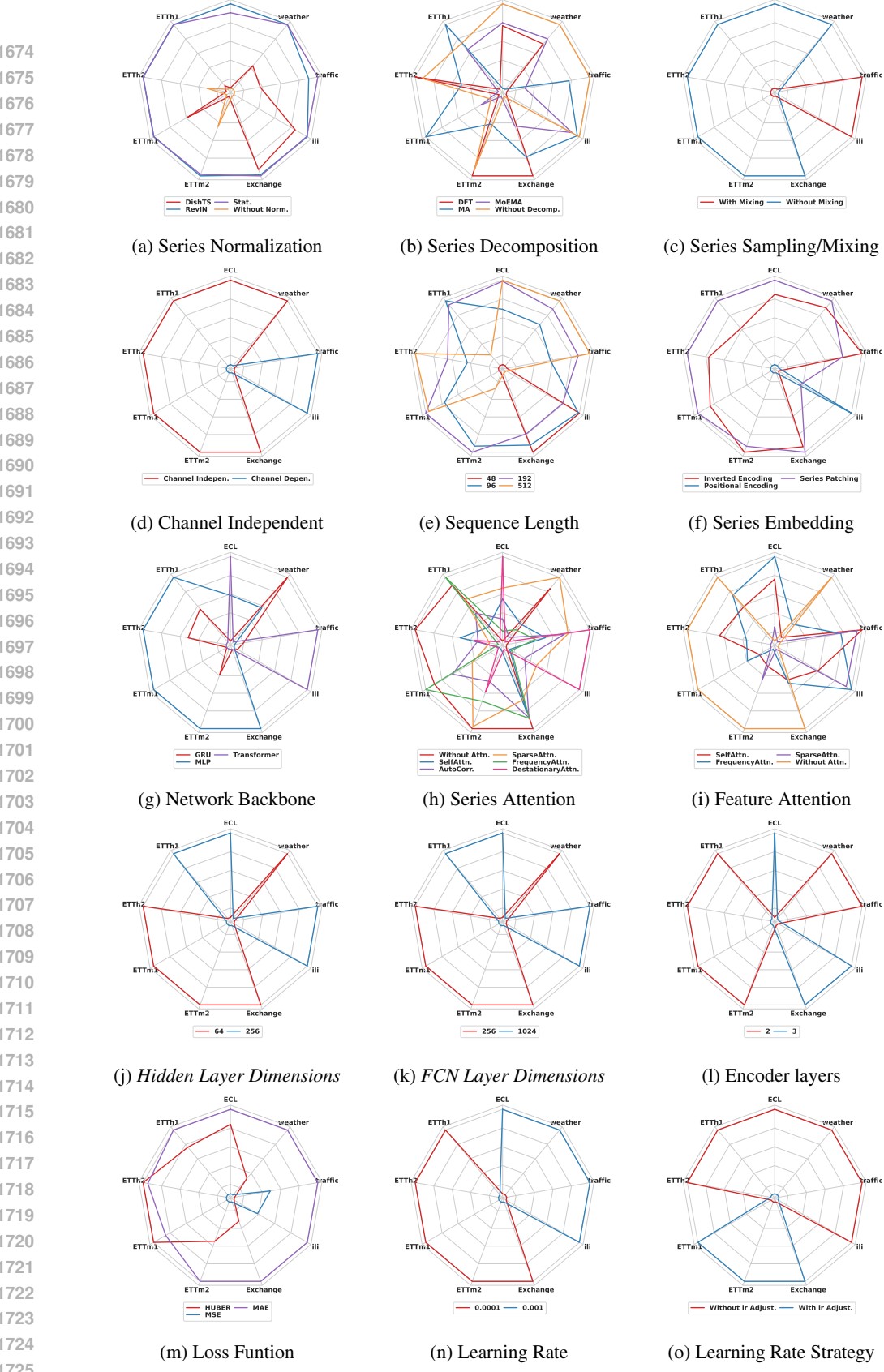

Figure H6: Overall performance across key design dimensions in long-term forecasting. The results (**MAE**) are based on the top 25th percentile across all forecasting horizons.

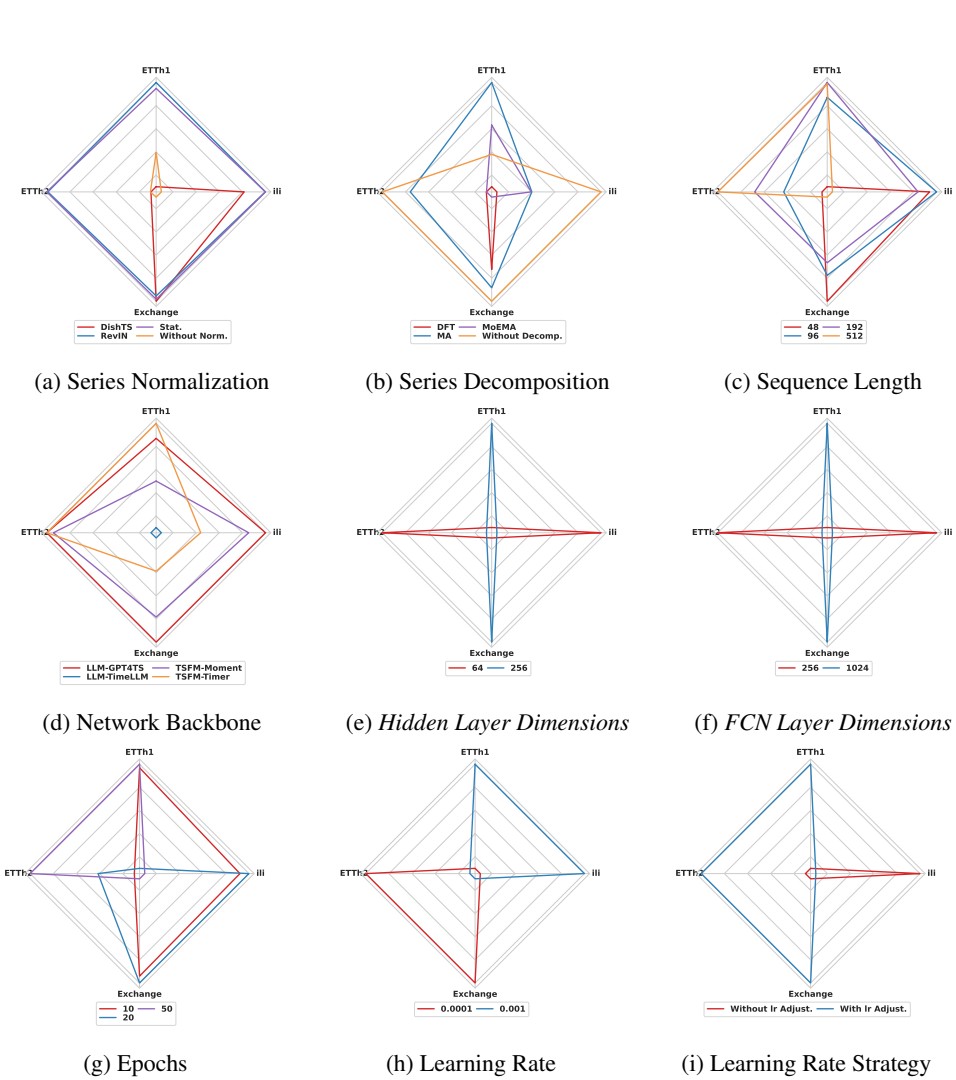

(a) Series Normalization  (b) Series Decomposition  (c) Sequence Length

(d) Network Backbone  (e) *Hidden Layer Dimensions*  (f) *FCN Layer Dimensions*

(g) Epochs  (h) Learning Rate  (i) Learning Rate Strategy

Figure H7: Overall performance across all design dimensions when using LLMs or TSFMs in long-term forecasting. The results (**MAE**) are based on the top 25th percentile across all forecasting horizons.

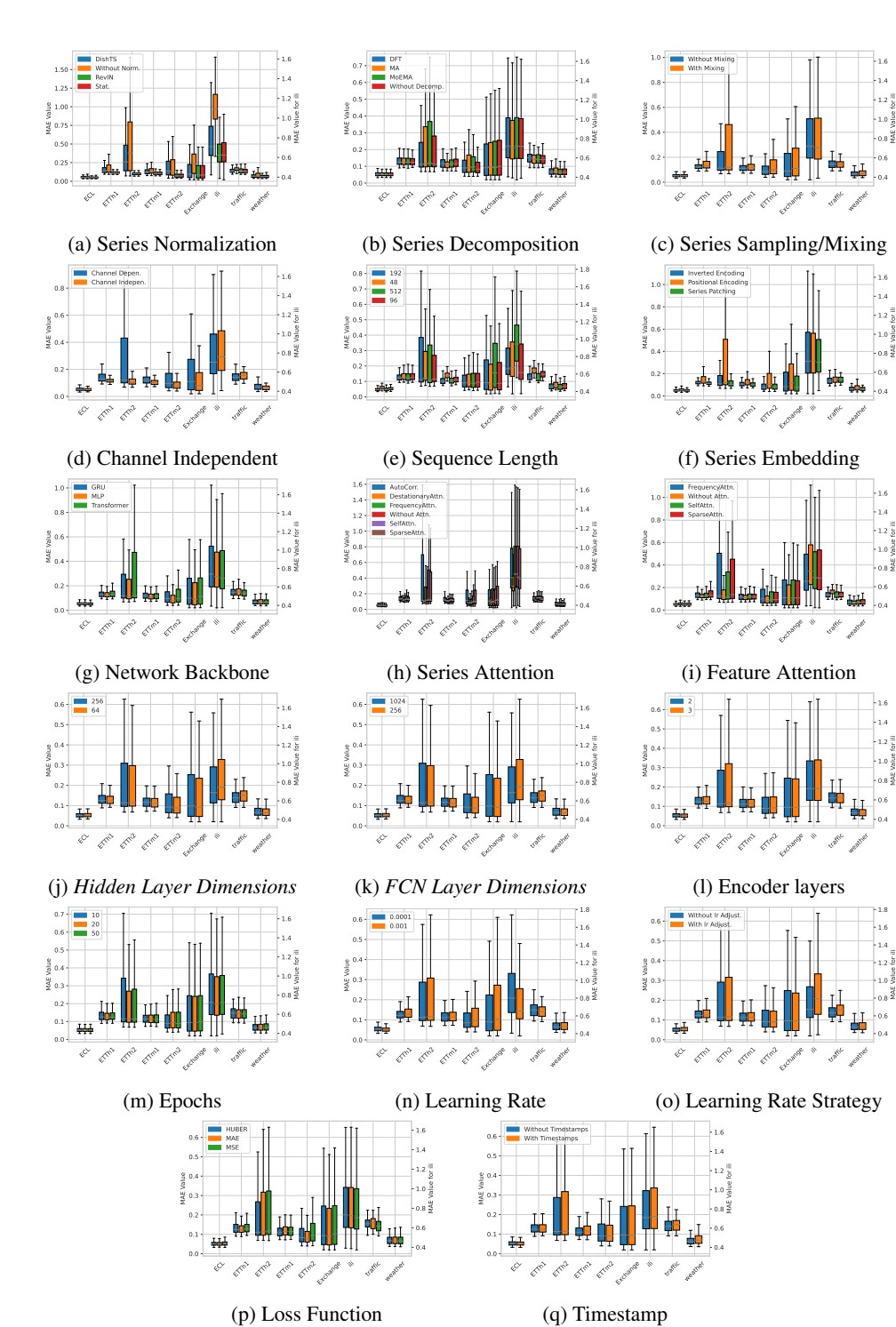

Figure H8: Overall performance across all design dimensions in long-term forecasting. The results (**MAE**) are averaged across all forecasting horizons.

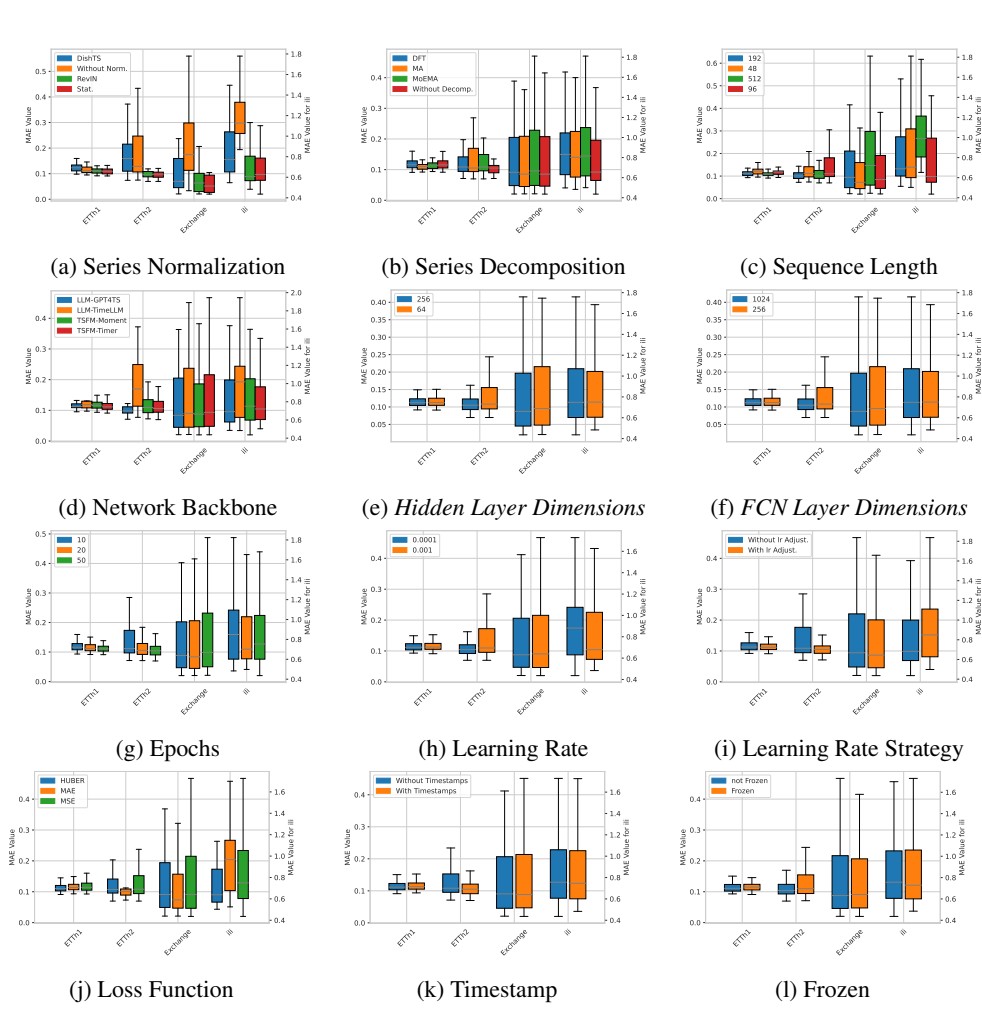

Figure H9: Overall performance across all design dimensions when using LLMs or TSFMs in long-term forecasting. The results (**MAE**) are averaged across all forecasting horizons.

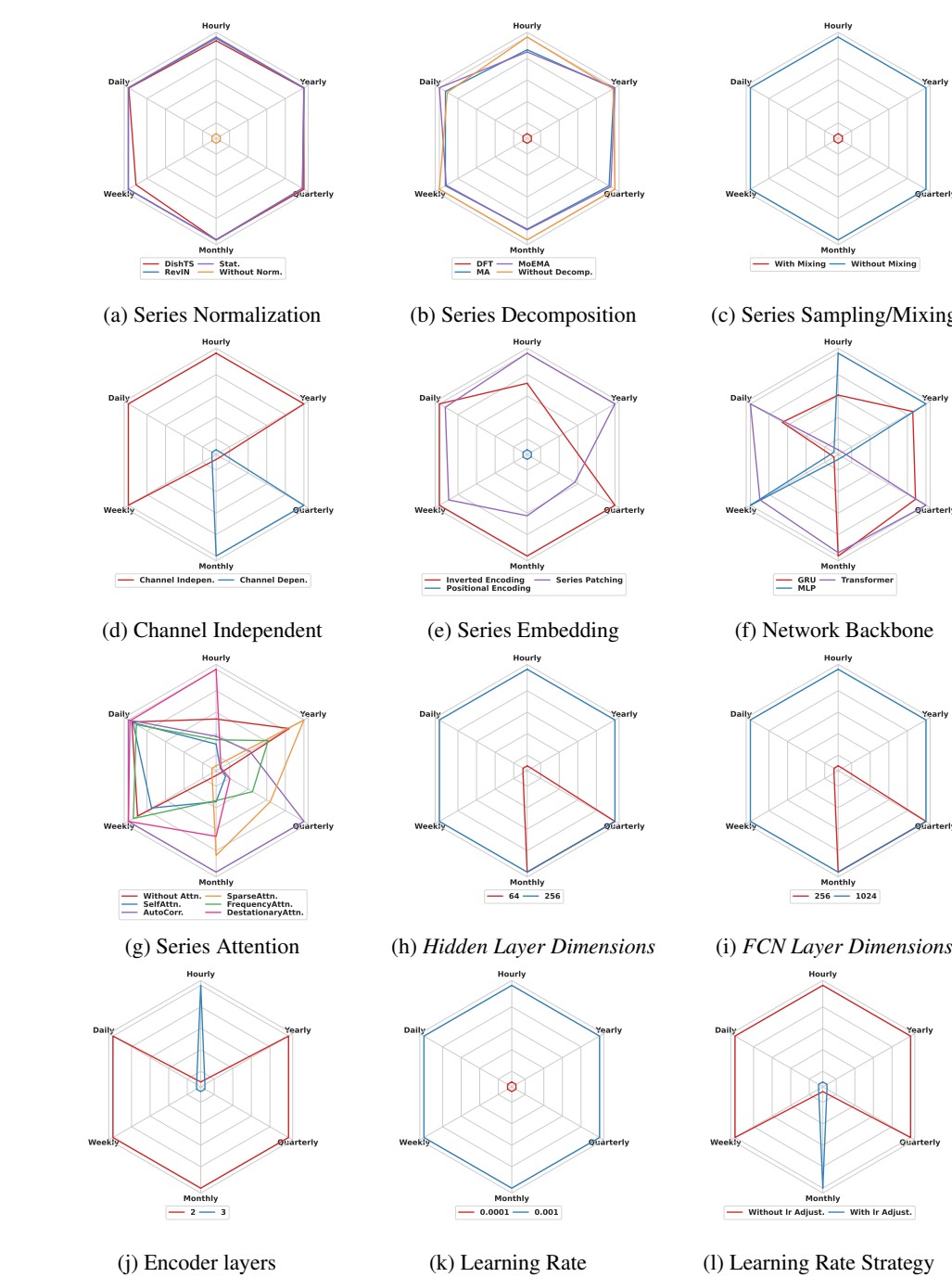

Figure H10: Overall performance across all design dimensions in short-term forecasting. The results (**MASE**) are based on the top 25th percentile across all forecasting horizons.

## H.3    COMPLETE EVALUATION RESULTS OF SHORT-TERM FORECASTING USING MASE, OWA AND SMAPE AS THE METRIC

For short-term forecasting, we comprehensively evaluate different design dimensions using both spider charts and box plots. The spider charts—shown in Figure H10, Figure H11, and Figure H12—visualize performance across datasets, with each vertex representing a benchmark dataset. Closer proximity to a vertex indicates stronger performance of a particular design choice in that dataset.

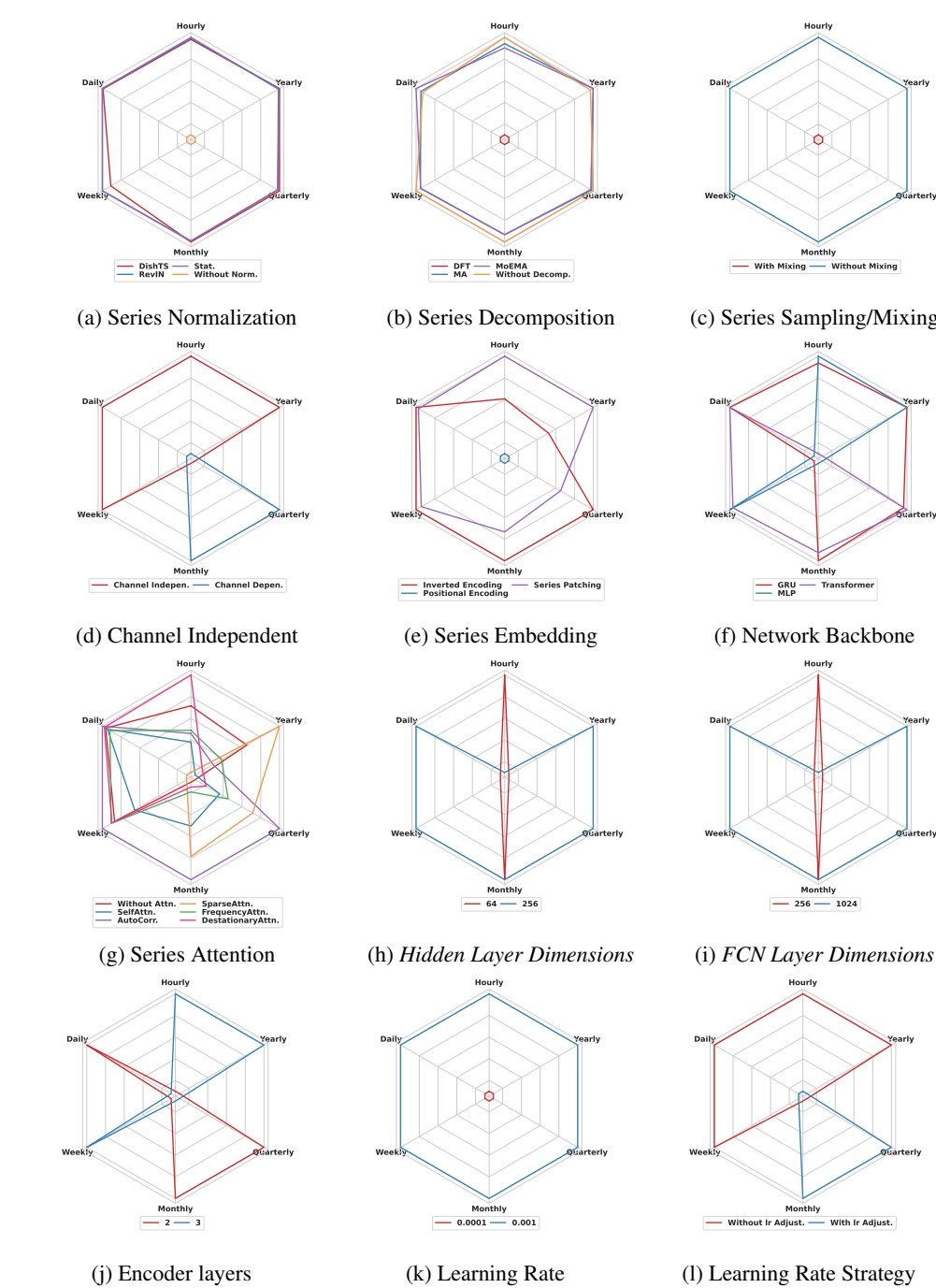

Figure H11: Overall performance across all design dimensions in short-term forecasting. The results (**OWA**) are based on the top 25th percentile across all forecasting horizons.

Complementary box plots are provided in Figure H13, Figure H14, and Figure H15, offering a statistical perspective on the distribution and robustness of performance across evaluation metrics.

Overall, the relative performance trends observed under MASE, OWA, and sMAPE metrics are consistent with those found in long-term forecasting tasks, reinforcing the generalizability and stability of our architectural choices.

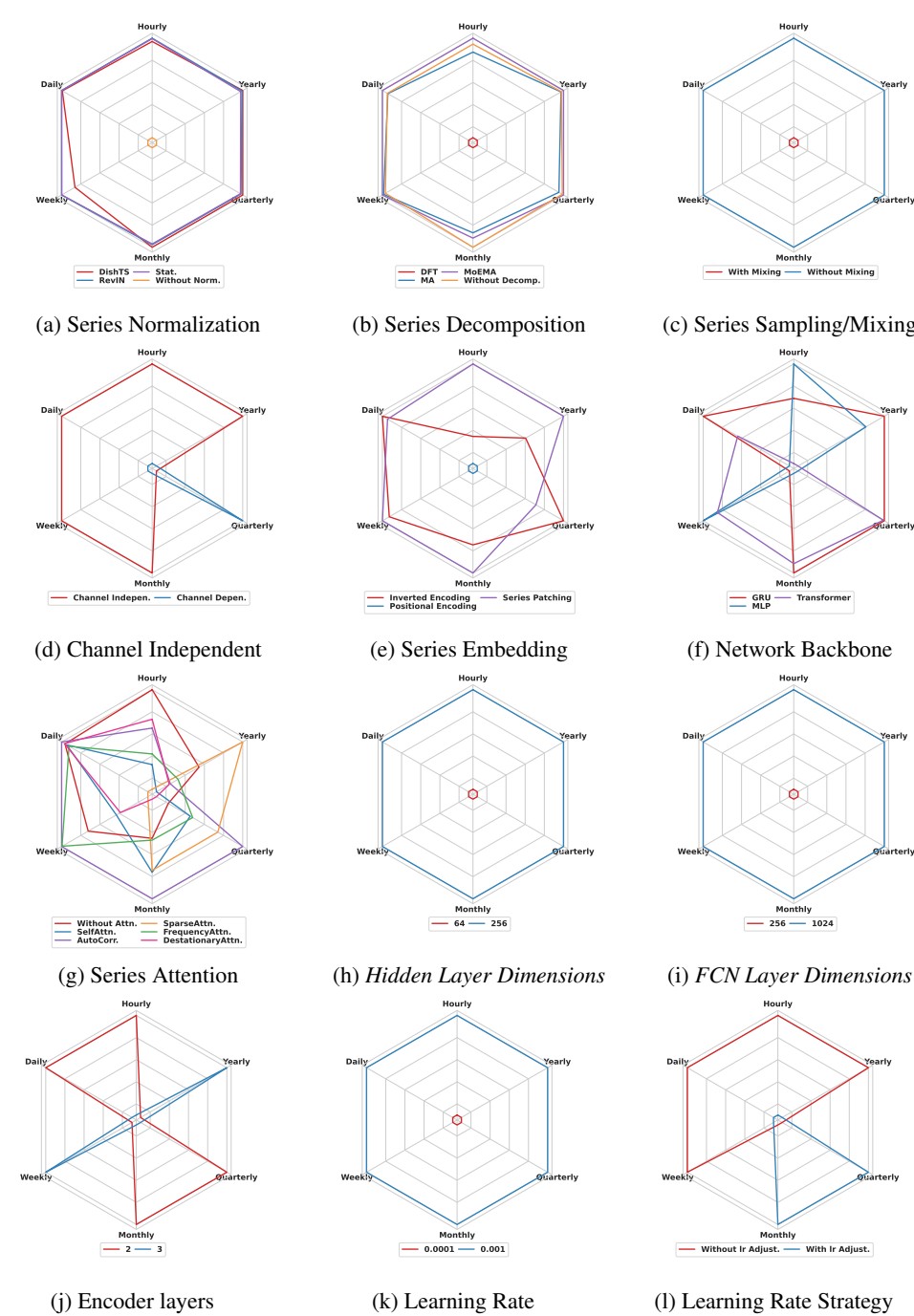

Figure H12: Overall performance across all design dimensions in short-term forecasting. The results (**SMAPE**) are based on the top 25th percentile across all forecasting horizons.

## H.4 EXPLAINING DESIGN DRIVERS VIA META-FEATURE IMPORTANCE ANALYSIS

To directly investigate the impact of individual meta-features, we conducted an additional analysis using an interpretable XGBoost-based meta-learner. Although this machine learning–based variant slightly underperforms compared to the original deep learning–based meta-learner (average MAE 0.447 vs. 0.426), due to its limited capacity in modeling rich, high-dimensional interactions among features, it remains competitive and provides a clear advantage in interpretability.

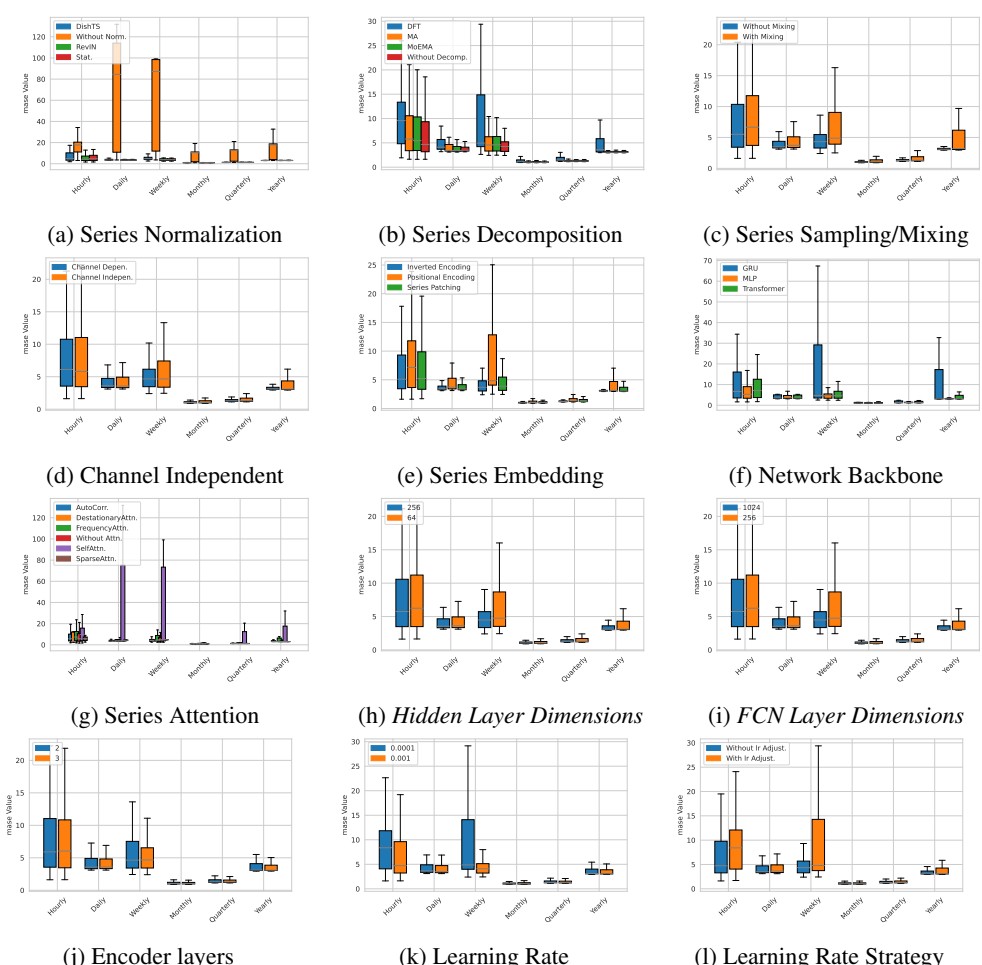

Figure H13: Overall performance across all design dimensions in short-term forecasting. The results are based on **MASE**.

Table H14: Top 5 Most Important Meta-Features per Dataset Estimated via XGBoost

| Dataset | Top 5 Meta-Features (Importance) | | | | |
|---|---|---|---|---|---|
| ETTm1 | mean_Negativeturningpoints (0.08) | series norm (0.06) | mean_Centroid (0.05) | mean_MFCC_0 (0.05) | min_Positiveturningpoints (0.05) |
| ETTm2 | mean_Negativeturningpoints (0.10) | series norm (0.05) | mean_MFCC (0.05) | mean_Centroid (0.05) | q25_Kurtosis (0.04) |
| ETTh1 | mean_MFCC_10 (0.08) | series norm (0.06) | mean_Spectralroll-on (0.05) | mean_Spectraldistance (0.04) | mean_MFCC (0.04) |
| ETTh2 | mean_MFCC_0 (0.07) | series norm (0.05) | mean_Medianfrequency (0.05) | mean_Centroid (0.04) | min_Meanabsolutediff (0.04) |
| ECL | mean_MFCC (0.08) | std_MFCC (0.08) | series norm (0.07) | mean_Centroid (0.06) | mean_Maxpowerspectrum (0.04) |
| Traffic | q25_Kurtosis (0.08) | series norm (0.07) | mean_Centroid (0.06) | mean_Maxpowerspectrum (0.05) | mean_MFCC (0.05) |
| Weather | mean_MFCC (0.14) | mean_Negativeturningpoints (0.11) | series norm (0.06) | min_Negativeturningpoints (0.05) | mean_Centroid (0.04) |
| Exchange | std_MFCC (0.11) | mean_Negativeturningpoints (0.10) | mean_MFCC (0.07) | series norm (0.06) | mean_Medianfrequency (0.03) |
| ILI | mean_MFCC (0.09) | mean_Maximumfrequency (0.06) | channel independent (0.05) | series norm (0.05) | mean_LPCC (0.05) |

This analysis allows us to quantify the relative importance of meta-features and structural design dimensions in determining model performance. As summarized in Table H14, certain temporal and spectral features—such as MFCC descriptors and Negative Turning Points—consistently appear among the most influential across datasets. In addition, architectural design choices like series normalization emerge as universally important factors, further validating the findings of our component-level ablation study.

## H.5 META-FEATURE SIMILARITY ENABLES TARGETED KNOWLEDGE TRANSFER

In Fig. G1, we visualize the dimension-reduced meta-features across different datasets using PCA. The visualization confirms that datasets tend to cluster based on inherent properties, such as domain (e.g., ETT family) and temporal frequency (e.g., M4-Hourly vs. M4-Yearly). This indicates that meta-feature similarity reflects structural characteristics of datasets, and suggests the potential for targeted knowledge transfer between similar datasets.

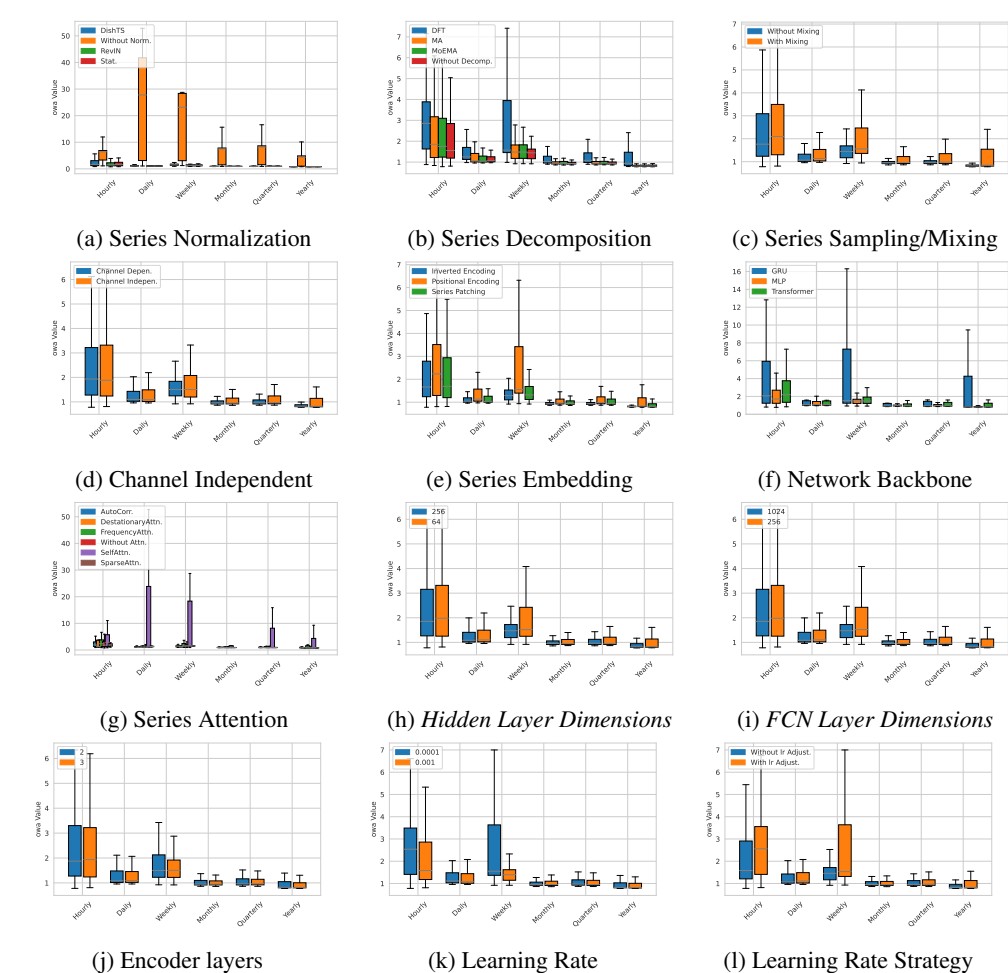

Figure H14: Overall performance across all design dimensions in short-term forecasting. The results are based on **OWA**.

Table H15: Ablation study of TSGym incorporating LLM and TSFM in 4 datasets. The average results of all prediction lengths are listed here.

| Models | TSGym(Ours) | | TSGym (LGB) | | TSGym (XGB) | |
|---|---|---|---|---|---|---|
| Metric | MSE | MAE | MSE | MAE | MSE | MAE |
| **ETTm1** | 0.362 | 0.38 | **0.352** | **0.374** | 0.362 | 0.387 |
| **ETTm2** | 0.266 | 0.322 | 0.258 | 0.315 | **0.256** | **0.311** |
| **ETTh1** | **0.427** | 0.439 | 0.464 | 0.457 | 0.434 | **0.428** |
| **ETTh2** | 0.367 | 0.403 | **0.351** | **0.394** | 0.37 | 0.396 |
| **ECL** | **0.164** | **0.261** | 0.173 | 0.268 | 0.177 | 0.268 |
| **Traffic** | 0.433 | 0.301 | **0.421** | **0.282** | 0.422 | 0.29 |
| **Weather** | 0.240 | 0.276 | 0.247 | 0.268 | **0.235** | **0.266** |
| **Exchange** | **0.375** | **0.415** | 0.423 | 0.433 | 0.415 | 0.436 |
| **ILI** | **2.463** | **1.043** | 2.575 | 1.091 | 3.620 | 1.314 |

To further explore this, we conducted a case study focusing on the ILI dataset—a relatively difficult and data-scarce task. We enriched the meta-learner's training pool by adding two datasets (COVID-19 and FRED-MD) that are more similar to ILI in the meta-feature space. As shown in Table H16, TSGym's performance on ILI improves

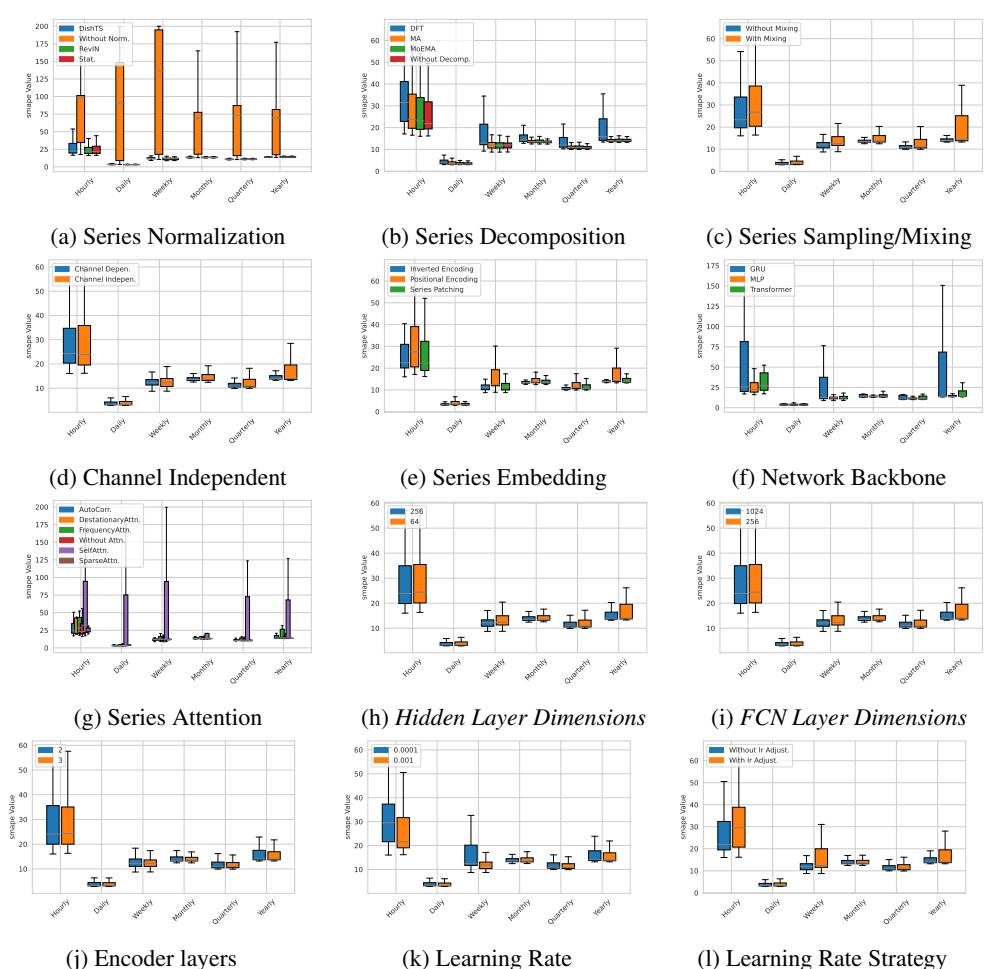

Figure H15: Overall performance across all design dimensions in short-term forecasting. The results are based on **SMAPE**.

Table H16: Performance Comparison Before and After Adding Similar Datasets (COVID-19 and FRED-MD) to the Meta-Learner Training Pool

| Metric | TSGym | | +COVID-19, FRED-MD | |
|---|---|---|---|---|
| | MSE | MAE | MSE | MAE |
| **ETTm1** | 0.362 | **0.380** | **0.358** | 0.381 |
| **ETTm2** | 0.266 | 0.322 | **0.259** | **0.315** |
| **ETTh1** | 0.427 | **0.439** | **0.424** | 0.442 |
| **ETTh2** | 0.367 | 0.403 | **0.357** | **0.396** |
| **ECL** | **0.164** | 0.261 | 0.164 | **0.259** |
| **Traffic** | 0.433 | 0.301 | **0.421** | **0.284** |
| **Weather** | 0.240 | 0.276 | **0.238** | **0.269** |
| **Exchange** | **0.375** | **0.415** | 0.438 | 0.438 |
| **ILI** | 2.463 | 1.043 | **2.020** | **0.881** |

significantly, while performance on other datasets remains stable or even improves slightly. This highlights the potential of incorporating similar datasets to enhance performance on low-resource or underperforming tasks.

## H.6 META-LEARNER PERFORMANCE SCALING WITH CANDIDATE POOL SIZE

To investigate how the size of the candidate model pool $M_s$ affects meta-learner performance, we conducted a scaling analysis across all datasets. We trained the meta-learner on progressively larger subsets of $M_s$ (ranging from 5% to 100%), and measured the average rank of the model selected by TSGym.

As shown in Table H17, the performance improves significantly as the pool size increases up to 25%, after which the gains plateau. Remarkably, even with just 10% of the full pool, TSGym already outperforms strong baselines such as DUET (which achieves average ranks of 4.11 for MSE and 3.67 for MAE). This highlights the high sample efficiency of TSGym and suggests that a moderately sized pool is sufficient to reach near-optimal performance. These results further motivate the use of smarter sampling strategies, such as Bayesian Optimization, to construct high-quality training pools with minimal cost.

Table H17: Effect of candidate pool size on meta-Learner selection accuracy

| Subset Size of $M_s$ | MSE (Avg. Rank) | MAE (Avg. Rank) |
|:---:|:---:|:---:|
| 5% | 3.67 | 3.44 |
| 10% | 2.67 | 3.00 |
| 25% | 1.67 | 1.78 |
| 50% | 1.89 | 2.67 |
| 75% | 1.89 | 2.22 |
| 100% | 1.67 | 2.00 |

## H.7 PERFORMANCE COMPARISON ACROSS SAMPLING STRATEGIES

Table H18: Comparison of MSE distribution between Optuna and random search across datasets

| Dataset | Method | Mean_mse | Std_mse | Min_mse | Q1_mse | Median_mse | Q3_mse | Max_mse | Total Experiment Count |
|:---:|:---:|:---:|:---:|:---:|:---:|:---:|:---:|:---:|:---:|
| ETTm1 | Optuna | **11.223** | **215.871** | 0.293 | **0.341** | **0.405** | **0.472** | **4317.839** | **400** |
| | Random | 264.510 | 8964.756 | **0.286** | 0.376 | 0.449 | 0.544 | 304538.500 | 1154 |
| ETTm2 | Optuna | **0.454** | 0.762 | **0.159** | **0.221** | **0.279** | **0.385** | **9.344** | **400** |
| | Random | 0.875 | **6.595** | **0.159** | 0.250 | 0.354 | 0.525 | 198.023 | 1231 |
| ETTh1 | Optuna | **0.506** | **0.157** | **0.355** | **0.416** | **0.450** | **0.519** | **1.210** | **400** |
| | Random | 0.560 | 0.199 | **0.355** | 0.442 | 0.496 | 0.587 | 2.085 | 2897 |
| ETTh2 | Optuna | **0.749** | **0.992** | 0.268 | **0.342** | **0.400** | **0.539** | **9.060** | **400** |
| | Random | 11.414 | 578.115 | **0.270** | 0.383 | 0.454 | 1.105 | 32581.227 | 3176 |
| ECL | Optuna | **0.189** | **0.041** | **0.131** | **0.158** | **0.182** | **0.213** | **0.414** | **400** |
| | Random | 0.217 | 0.050 | 0.134 | 0.180 | 0.212 | 0.247 | 0.862 | 1603 |
| Traffic | Optuna | **0.534** | **0.125** | 0.387 | **0.438** | **0.491** | **0.612** | **1.051** | **400** |
| | Random | 0.600 | 0.130 | **0.379** | 0.491 | 0.595 | 0.686 | 1.473 | 1145 |
| Weather | Optuna | **0.342** | **1.490** | 0.144 | **0.193** | **0.245** | **0.312** | **29.895** | **400** |
| | Random | 574.721 | 18520.072 | **0.143** | 0.207 | 0.263 | 0.343 | 597254.813 | 1040 |
| Exchange | Optuna | **0.687** | 1.305 | 0.081 | **0.169** | **0.280** | **0.681** | **15.054** | **400** |
| | Random | 0.761 | **1.050** | **0.079** | 0.184 | 0.375 | 0.963 | 17.898 | 5509 |
| ILI | Optuna | **2.687** | **1.102** | 1.506 | **1.891** | **2.302** | **3.080** | **7.503** | **400** |
| | Random | 3.278 | 1.132 | **1.495** | 2.397 | 2.899 | 4.046 | 7.642 | 10734 |

To enhance the quality of the randomly sampled component pool and thereby improve final model performance, we introduced a smarter sampling strategy using Optuna, a Bayesian optimization-based method. The sampling process began with a cold start of 50 random configurations to provide a diverse baseline for the Bayesian optimizer and mitigate the risk of early local convergence. Building upon this initial exploration, Optuna guided the sampling of an additional 50 high-quality candidates.

Table H18 reports the MSE distribution statistics for configurations sampled by Optuna and random search across various datasets. Optuna produces a result distribution that is markedly better than that of random sampling. We also note that Optuna can provide interpretability. Table H19 shows the importance of each design dimension estimated by Optuna's built-in fANOVA analysis. Sequence Length and Series Normalization contribute the most to performance variation, suggesting their critical role in architecture design.

Table H19: Relative importance of design dimensions estimated by Optuna's fANOVA analysis

| Rank | Design Dimensions | Importance |
|:---:|:---:|:---:|
| 1 | Sequence Length | 0.270 |
| 2 | Series Normalization | 0.255 |
| 3 | Series Embedding | 0.134 |
| 4 | Feature Attention | 0.077 |
| 5 | Series Decomposition | 0.053 |
| 6 | Channel Independent | 0.050 |
| 7 | Series Sampling/Mixing | 0.029 |
| 8 | Epochs | 0.025 |
| 9 | d_model d_ff | 0.020 |
| 10 | Learning Rate | 0.020 |
| 11 | With/Without Timestamps | 0.018 |
| 12 | Network Type | 0.017 |
| 13 | Encoder Layers | 0.013 |
| 14 | Learning Rate Strategy | 0.012 |
| 15 | Loss Function | 0.010 |
| 16 | Series Attention | 0.000 |

## I LLM Usage Statement

We used a large language model (LLM) solely for English-language polishing (grammar, tone, and minor phrasing) and for minor LaTeX table formatting adjustments. The LLM did not contribute to research ideation, problem formulation, experimental design, data collection, or citation generation.

## J Reproducibility Statement

To facilitate the verification and extension of our work, we hereby affirm our commitment to the reproducibility of all experimental results presented in this paper, particularly those in Section 4.

Upon acceptance of this paper, we will release the following resources under an open-source license:

- **Complete Codebase:** The full source code for data preprocessing, model training, hyperparameter configurations, and evaluation metrics.
- **Environment Specifications:** A detailed list of dependencies (e.g., `requirements.txt`).
- **Processed Datasets:** The cleaned and structured datasets used in our experiments, along with scripts to load them.

Minor variations in results due to stochasticity or hardware differences are expected, but the primary conclusions and performance rankings are robust and reproducible. The resources will be made publicly available at a permanent repository, and the link will be included in the final version of the paper.

