# OpenReview forum: "TSGym: Automatic Model Design Framework for Deep Multivariate Time-Series Forecasting"
_ICLR.cc/2026/Conference — ICLR 2026 Conference Withdrawn Submission_

### Official Review · Reviewer_1pkZ · 2025-10-27

**Soundness:** 2
**Presentation:** 2
**Contribution:** 2
**Rating:** 2
**Confidence:** 5

**Summary:**

Time series modeling is a very active area of research, ranging from Statistical models to ML, then DL, and finally FM. This paper explicitly focuses on the multivariate aspect and the DL side of the vertical. This paper conducted an in-depth study of all components used in DL, including data preprocessing, Modeling, etc.

While attempting to standardize, this should be an objective of an OSS library such as sktime, and using their components preferably.  I have provided several comments for manuscript improvement.

**Strengths:**

1. The concept of generating guidance for a new TSFM pipeline based on characteristics of the dataset is interesting (partially) and tested for DL DL-based component. This may potentially reduce the barrier for the end user. Not everyone is an expert in the field, and the author should put the work from this angle. AutoML's goal was to reach the general audience, and you are making life easier for them.

**Weaknesses:**

1. The word Gym has a different meaning in the RL community. The author should explain the word and its association with the RL world.
2. The font in all the images is tiny (Figure 1, Figure 2, Table 1, Table 2). Also, the color combination in Figure 1 needs some work to make it more readable.
3. The meta learning approach for time series data is not novel, and it has been there for a long time.
4. The contribution section of TSGym needs to be more quantitative, say, overall what the important insights are they find, the improvement, and how it performs with SOTA.
5. There has been an approach to AutoML for Time Series using the ML approach and the Foundation model approach. Why were those not considered for the baseline approach?
6. The datasets are the same as Gift-Eval, and this is why we do not use a standard approach and the baseline from these platforms to compare the net-gain the TSGym achieved.
7. The meta-learning needs to be compared with some baseline from the literature. Say ranking-based method was already discussed in some prior art - Catch22, and then some of their
8. As highlighted, this toolkit is developed for whom> what are the end user personas?
9. Did the author standardize their component with any OSS library, say sktime? Or say IBM-AutoAI-TS or H2O-AutoAI-TS?
10. Was their any hyper-parameter grid being prepared?

**Questions:**

Please address all weak points

---

### Official Review · Reviewer_tqYV · 2025-10-29

**Soundness:** 2
**Presentation:** 3
**Contribution:** 2
**Rating:** 2
**Confidence:** 3

**Summary:**

This paper proposes automatic model selection for time-series forecasting. It is claimed to select a component-level configuration such that a customized model can be produced. It does so via a meta learner trained to select the best model for given datasets.

**Strengths:**

1) the proposed method performs well;
2) method has some novelty.

**Weaknesses:**

1) I wonder how this proposed method differs from a structural learning approach such as NAS.
2) It is necessary to compare this method against other structural learning methods, although comparisons with other model selections have been done.
3) complexity analysis should be provided.

**Questions:**

1) I wonder how this proposed method differs from a structural learning approach such as NAS.
2) It is necessary to compare this method against other structural learning methods, although comparisons with other model selections have been done.
3) complexity analysis should be provided.

---

### Official Review · Reviewer_K9mH · 2025-10-29

**Soundness:** 2
**Presentation:** 2
**Contribution:** 2
**Rating:** 2
**Confidence:** 5

**Summary:**

This paper presents TSGym, an automated model-design framework for multivariate time-series forecasting. The authors decouple existing deep MTSF pipelines into fine-grained design dimensions (preprocessing, encoding, architecture, optimization), define a design space, and sample configurations to build a performance matrix over multiple datasets. They train a meta-predictor that maps dataset meta-features and component embeddings to relative performance ranks, enabling zero-shot selection of a top model for a new dataset. Extensive experiments across standard benchmarks show that the automatically constructed pipelines often outperform strong baselines and AutoML competitors.

**Strengths:**

1. The paper systematically breaks MTSF pipelines into meaningful component dimensions and enumerates many realistic design choices; this level of granularity is useful for both analysis and automated construction.
2. The work evaluates a broad set of configurations across multiple widely used time-series benchmarks and includes several thoughtful ablations (sampling strategy, pool size, architecture subsets), producing extensive experimental evidence.

**Weaknesses:**

1. Incremental novelty. The main novelty is engineering and scale — expanding the AutoML/search space to include data-processing choices and recent model types (LLMs/TSFMs). Conceptually this is a natural, incremental extension of prior AutoML work.
2. Reliance on predictor correctness for insights. Many of the paper’s component-level analyses and insights about which design choices work best depend heavily on the meta-predictor’s accuracy. In contrast, [1] employs a sampling-based statistical strategy, which potentially lead to more reliable insights into design choices.
3. Lack of Computational Cost Analysis: There is no discussion of the GPU hours required to collect the training samples. As shown in Table F3, the authors collected 57,600 training samples—this process could take a very long time to train even if they use 12 GPUs.

[1] Designing Network Design Spaces.

**Questions:**

see Weaknesses.

---

### Official Review · Reviewer_4ymY · 2025-11-01

**Soundness:** 3
**Presentation:** 3
**Contribution:** 3
**Rating:** 8
**Confidence:** 4

**Summary:**

The paper introduces TSGym, a framework for large-scale evaluation, component-level analysis and automated model construction in deep multivariate time series forecasting tasks. This leads to three key contributions: First, the large-scale component-level analysis allows insights into general design choices for forecasting models. Second, the automated component-based model construction yields models that outperform current state of the art models. Third, TSGym widens the scope of multivariate time series forecasting by including novel time-series architectures like LLMs or time series foundation models in the framework.

**Strengths:**

* Originality: TSGym is the first automated system that generates models for time-series forecasting based on individual components.
* Quality: An ablation study is provided in the appendix, validating the design choices of the proposed framework. TSGym is compared in a fair manner to other approaches. The performed work and presented theory are correct and reasonable.
* Clarity: The paper is well-written, uses concise language and is easy to follow. The research goal, proposed methodology and the conducted experiments are clearly understandable.
* Significance: TSGym beats state-of-the-art forecasting model approaches on commonly used datasets. Additionally, several general research questions in the field, like transformer vs. MLPs, are addressed in a meta-study.

**Weaknesses:**

* The paper does not mention the computational costs of using TSGym. It would be beneficial to provide additional information i.e. about computational time for the experiments conducted. Currently, it is difficult to estimate broader usability, especially on older and/or weaker hardware.
* Chapter 5 (Conclusion, limitations) does not mention any limitation of the methodology. I suspect that one limitation might be computational costs, especially when applying TSGym to a new domain that requires full retraining.
* To my understanding, each experiment was only conducted once per configuration but averaged over 4 predictions lengths per configuration. However, results would be more robust if there were repeated evaluations per configuration per prediction length.

Minor
--------
* The citations Ailing Zeng 2023a and Ailing Zeng 2023b are identical.
* Deep learning is abbreviated as DL in the introduction but not consequently used afterwards (i.e. in 2.1 the abbreviation is not used).
* There is a missing space in chapter 4.1 (baseline): latestapproach
* There is a grammar error in 4.3 (Question 4): Does large time-series models bring significant improvement for TSGym? The subject “models” is plural, so the verb should be “do” instead of “does”  Do large time-series models bring …
* The appendix could be reordered according to the reader’s flow (i.e. start with appendix B as it is mentioned in the text first and move appendix A further back).

**Questions:**

* It is claimed that TSGym has zero-shot capabilities through the trained meta-learner. The paper states that chapter 4.3 shows the effectiveness of this meta-predictor. However, I don’t understand how the results prove the effectiveness of the meta-learner. Is TSGym only applied to the mentioned datasets? If yes, which datasets did you use to train the framework? Please clarify this.
* The usage of meta features is mentioned in chapter 3. However, there is no mention about the number of extracted features and/or usage of an automated approach to feature extraction. While it is thoroughly explained in appendix G, I think it would be easier to understand by mentioning the usage of TSFEL and the number of extracted features in the main part of the paper, i.e. in chapter 4.1.

**Details Of Ethics Concerns:**

No ethics concerns.

---

### Note · Authors · 2025-11-17

**Comment:**

We are sincerely grateful to all the reviewers for their time and effort in evaluating our manuscript. We especially thank Reviewer 4ymY for the encouraging assessment and detailed feedback, and we appreciate all reviewers for their constructive suggestions.

The reviewers consistently and rightfully pointed to the computational efficiency of our method as a primary concern. We fully acknowledge that the up-front cost of conducting extensive experiments to build the meta-learning performance matrix is significant. This was a key critique from Reviewers K9mH, tqV, and 4ymY.

It is worth clarifying that a key advantage of our approach is that once this initial (admittedly high) cost is paid, our method can be transferred to new datasets with minimal overhead. Furthermore, to mitigate this primary challenge, optimizations such as using Optuna to refine the component search have been explored. This optimization significantly reduces the number of initial experiments needed to build a high-quality meta-training dataset, thereby lowering the pre-training cost. This also proves beneficial when migrating to new tasks.

Regarding the NAS comparison, we wish to clarify that our work is fundamentally zero-shot, requiring no iterative training on the target dataset. A direct comparison to iterative NAS methods may not be entirely fair. Furthermore, existing NAS frameworks are not specifically designed for deep time series forecasting. Adapting them would require invasive code development and significant re-componentization, a substantial undertaking. We did, however, conduct preliminary comparisons against other iterative AutoML methods that are more readily reproducible for time series forecasting, and our approach maintained a leading advantage.

We believe there are many exciting extensions to this work, such as a more granular component-level analysis and a more efficient architecture prediction framework, which were also alluded to by the reviewers. However, the limited time frame may not be sufficient to complete the comprehensive comparisons and in-depth analysis required to fully address these important points and satisfy the reviewers. Therefore, after careful consideration, the decision has been made to withdraw our submission.

We remain confident in the core contributions of TSGym: (1) providing the first large-scale, component-level evaluation for MTSF methods, offering critical insights; (2) offering an automated, meta-learning-based model constructor that outperforms SOTA methods; and (3) broadening the discussion to systematically include emerging large time-series models.

We thank all the reviewers once again. Your feedback is invaluable and will be instrumental in guiding the refinement of this research.

**Withdrawal Confirmation:**

I have read and agree with the venue's withdrawal policy on behalf of myself and my co-authors.